# Generation of shape complexity through tissue conflict resolution

**Alexandra B Rebocho[1], Paul Southam[1], J Richard Kennaway[1], J Andrew Bangham[2†], Enrico Coen[1]***

[1]Department of Cell and Developmental Biology, John Innes Centre, Norwich, England; [2]School of Computational Sciences, University of East Anglia, Norwich, England

**Abstract** Out-of-plane tissue deformations are key morphogenetic events during plant and animal development that generate 3D shapes, such as flowers or limbs. However, the mechanisms by which spatiotemporal patterns of gene expression modify cellular behaviours to generate such deformations remain to be established. We use the Snapdragon flower as a model system to address this problem. Combining cellular analysis with tissue-level modelling, we show that an orthogonal pattern of growth orientations plays a key role in generating out-of-plane deformations. This growth pattern is most likely oriented by a polarity field, highlighted by PIN1 protein localisation, and is modulated by dorsoventral gene activity. The orthogonal growth pattern interacts with other patterns of differential growth to create tissue conflicts that shape the flower. Similar shape changes can be generated by contraction as well as growth, suggesting tissue conflict resolution provides a flexible morphogenetic mechanism for generating shape diversity in plants and animals.

*For correspondence: enrico.coen@jic.ac.uk

†Deceased

**Competing interests:** The authors declare that no competing interests exist.

## Introduction

Out-of-plane deformation of tissue sheets plays a fundamental part in morphogenesis. In animals, it underlies processes such as gastrulation, neurulation, convolution of the cerebral cortex, gut folding, development of imaginal discs and dorsal appendages. Several models have been proposed to account for such deformations, involving a variety of processes such as differential surface contractions, oriented tissue tensions and differential growth (*Chen and Brodland, 2008*; *Clausi and Brodland, 1993*; *Conte et al., 2008*; *Hannezo et al., 2014*; *Osterfield et al., 2013*; *Savin et al., 2011*; *Tallinen et al., 2014*; *Wyczalkowski et al., 2012*). In plants, cell rearrangements and contractions play little or no role in morphogenesis, which is largely driven by growth. Nevertheless, out-of-plane tissue sheet deformations lead to the formation of elaborate structures such as floral spurs, the orchid labellum and pitcher-shaped leaves. These observations raise the question of how out-of-plane tissue deformations are generated in such diverse systems and how they relate to underlying patterns of gene activity and cell behaviours. Plants are a good starting point for addressing this question as lack of cell movement simplifies analysis.

A key feature of out-of-plane deformations is that they involve generation of curvature (local rotations out of the plane). Two mechanisms might account for the generation of local rotations for a tissue sheet. The first is that local rotations arise through forces external to the sheet pulling or pushing on particular regions. For example, petals (whorl 2) grow in between sepals (whorl 1) and stamens (whorl 3), and these adjacent organs could apply forces to shape the petal. However, homeotic mutants that change the identity of stamens to petals do not have a major effect on the complexity of whorl 2 petal shape (*Bradley et al., 1993*), making it unlikely that such a mechanism plays a major role in this case. The second mechanism is that regions within the tissue sheet are specified

**eLife digest** Plant and animal organs come in many different shapes, from pitcher-shaped leaves and butterfly wings, to orchid flowers and the convoluted shape of the brain. Unlike pottery or sculpture, no external hand guides the formation of these biological structures; they arise on their own, through sheets of cells developing into particular three-dimensional shapes. But how does this process of self-making operate? We know that patterns of gene activity are important, because mutations that disrupt these patterns change the shape of the organ. But it is not clear how these patterns lead to sheets of cells curving and bending themselves into their characteristic three-dimensional shapes.

Plants are particularly useful tools for studying how three-dimensional organs form because, unlike animals, their cells do not slide relative to each other, which makes the analysis simpler. Rebocho et al. used a combination of computational modelling and cell analysis to study how the intricately shaped flowers of plants known as Snapdragons form. The experiments show that genes control the shape of Snapdragon flowers by causing groups of cells to grow at different rates and in different directions. This pattern of growth creates internal conflicts that are resolved by sheets of cells curving in particular ways, accounting for the three-dimensional shape.

Rebocho et al. propose that the principles of tissue conflict resolution described in this work may also underlie the development and evolution of many other plant and animal organ shapes. A future challenge is to identify the cellular mechanisms that link patterns of gene activity to the generation and resolution of conflicting cell behaviours.

to grow at different rates and/or directions. Local rotations can arise because they reduce or resolve potential conflicts brought about by such differential growth (*Coen and Rebocho, 2016*), as without regions curving or rotating relative to each other, greater levels of stress would be generated. We refer to this second mechanism, in which heterogeneity of specified growth within the tissue leads to local rotations that reduce potential stresses, as *tissue conflict resolution* (for a more mathematical definition of tissue conflict resolution see Materials and methods).

To clarify the notion of tissue conflict resolution we distinguish between two types of growth: specified and resultant (*Kennaway et al., 2011*). Specified growth is how a region of tissue would deform if it was free from the mechanical constraints of its neighbouring regions. Resultant growth is how a region deforms in the context of neighbouring mechanical constraints, and includes anisotropies and local rotations that emerge from such constraints. Specified growth therefore refers to the intrinsic or active properties of a region, which may be influenced by local gene expression, while resultant growth also includes the passive changes that arise through connectivity with other regions. It is usually not possible to infer specified growth patterns directly from observed deformations (which reflects resultant growth). Modelling allows the consequences of particular hypotheses for specified growth to be evaluated and compared to the data on resultant growth, such as clones and shape deformations.

To illustrate how patterns of specified growth may lead to out-of-plane deformations, consider a square sheet of tissue marked with circular spots (virtual clones, *Figure 1A*). If specified growth is equal in all directions (isotropic specified growth) and a growth-promoting transcription factor, GTF (red shading in *Figure 1*), is expressed uniformly, the tissue simply gets larger (*Figure 1B*, *Video 1*). Alternatively, specified growth could also be anisotropic, in which case regions have the intrinsic property of growing preferentially in one orientation. A simple way to establish such orientations in a tissue is through a polarity field (arrows *Figure 1C*). If specified growth is higher parallel to the local polarity, the tissue elongates (*Figure 1D*, *Video 2*). In both of these examples, all regions within the tissue grow in a similar way without constraining each other, so resultant growth is the same as specified growth. There is no tissue conflict and local rotations are not generated.

Local rotations and curvature can result through spatial variation in specified growth, causing buckling or bending of the tissue. We may define three types of conflict leading to local rotations: surface, areal and directional. If GTF promotes isotropic growth and is expressed at higher level in the top compared to the bottom surface (red vs pink shading in *Figure 1E*), the tissue folds as this

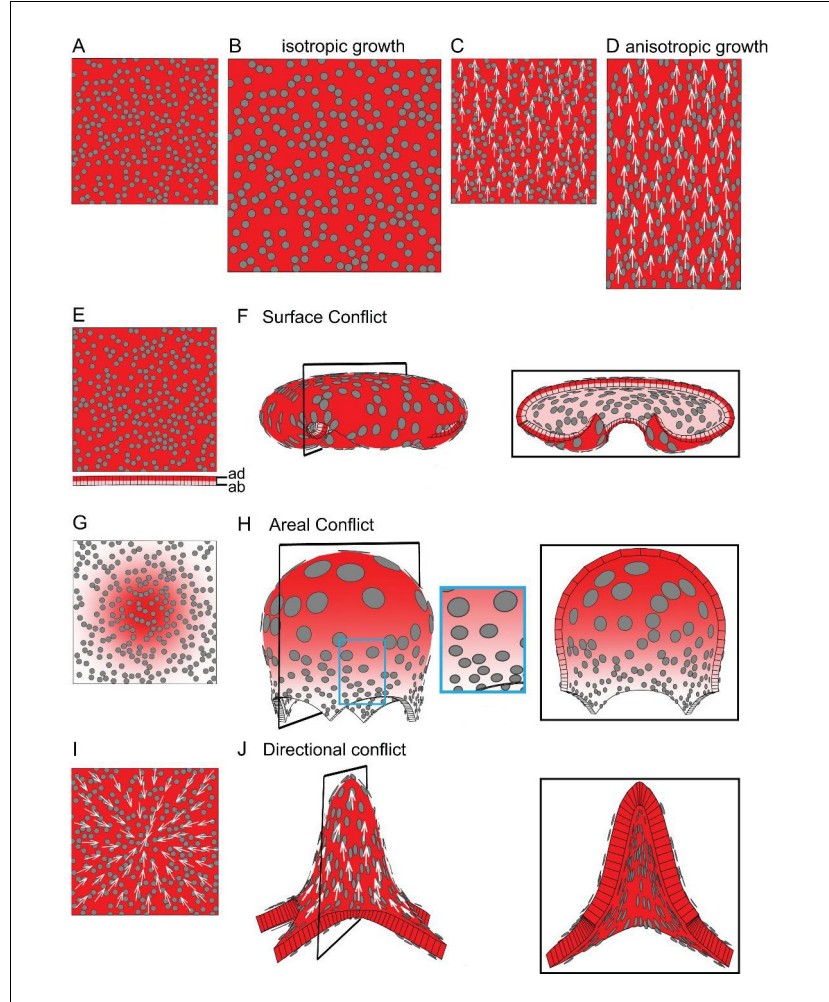

**Figure 1.** Generation of 3D deformations through tissue conflict resolution. (**A–B**) Isotropic specified growth promoted by a uniformly expressed Growth-promoting Transcription Factor (GTF, red shading). The initial square marked with circular clones (**A**) grows into a bigger square with enlarged isometric clones (**B**). (**C–D**) A proximodistal polarity field (white arrows) with uniformly expressed GTF promoting growth parallel to the polarity (specified anisotropic growth). The square (**C**) elongates to from a rectangle (**D**). (**E–F**) *Surface conflict*. GTF promotes specified isotropic growth and is expressed at a higher level in the top surface (red, adaxial, ad) compared to the bottom (pink, abaxial, ab) (**E**). The canvas deforms into a dome with downwards curled edges (**F**). (**G–H**) *Areal conflict*. GTF promotes specified isotropic growth and is more highly expressed in the centre of the canvas (**G**). The canvas deforms into a rounded dome with circular clones bigger at the apex (**H**, side view in left panel and clipped view in right panel) and slightly elliptical at the periphery of the dome (blue square in **H**). (**I–J**) *Directional conflict* with a convergent polarity field (white arrows) and GTF promoting growth parallel to the polarity. The square deforms into an elongated dome with clones elongated parallel to the polarity field (**J**, side view in left panel, clipped view in right panel). For each model the position of the clipping plane is indicated by black line in the side view.

The following figure supplement is available for figure 1:

**Figure supplement 1.** Areal and directional conflicts with flat starting tissue.

reduces the potential conflict in growth between of the two surfaces (*surface conflict*, *Figure 1F*, *Video 3*). If GTF is expressed at a higher level in the centre of the tissue compared to the surround (*Figure 1G*), the *areal conflict* is reduced by the tissue buckling and formation of a round dome (*Figure 1H*, *Video 4*). The direction (up or down) and pattern of buckling may be biased if the sheet has an initial slight curvature generated by surface conflict, or variable if it is initially flat with slight

random perturbations in height (*Figure 1—figure supplement 1A–B*). Even though specified growth is isotropic, anisotropies may result from areal conflict. For example, clones in regions with low specified growth become stretched circumferentially (blue box in *Figure 1H*) by nearby faster growing regions. These anisotropies are a passive result of residual stresses generated by differential growth, rather than being directly specified locally. Residual stresses arise because local rotations only partially resolve the areal conflict (for more details, see description of tissue conflict resolution in Materials and methods). Examples of buckling arising through areal conflict have been described previously (*Conte et al., 2008*; *Green, 1992*; *Nath et al., 2003*; *Shi et al., 2014*).

A third type of conflict is generated when the orientations of specified growth vary. For example, if the polarity field converges at the centre of the tissue (*Figure 1I*) and specified growth rate is higher parallel to the polarity, the **directional conflict** is partially resolved by buckling and formation an elongated dome (*Figure 1J*, *Video 5*). Virtual clones are elongated along the radial axis of the dome (*Figure 1J*). In this example, directional conflict involves a non-parallel polarity field, but it could also involve variation in specified growth rates in relation to polarity. As with areal conflict, the direction of buckling can be biased through initial curvature (*Figure 1—figure supplement 1C–D*).

Unlike surface and areal conflicts, directional conflicts have been little explored. They have been proposed to be involved in generating the Snapdragon corolla shape, during a repatterning event (*Green et al., 2010*; *Kennaway et al., 2011*). However, the precise pattern these conflicts take and their operation at the cellular level has not been tested.

Here we address this problem using a combination of cellular and tissue analysis, together with modelling at different levels of complexity. We show that an orthogonal pattern of directional conflict contributes to the out-of-plane deformation of the lower corolla. This pattern is likely established by a combination of orthogonal gene activity with a polarity pattern visualised by PIN1 localisation. The directional conflict interacts with surface and areal conflicts to generate the corolla shape and curvature. Dorsoventral genes modulate all three types of conflict accounting for a range of phenotypes in wild type and mutants. We propose that genetically controlled tissue conflict resolutions may provide a general mechanism for generating out-of-plane deformations in both plants and animals, despite different underlying cell behaviours.

## Results

### Morphogenesis of wild type and *div*

The bilaterally symmetric Snapdragon (*Antirrhinum majus*) corolla has five petals united for part of their length to form a tube with five separate lobes, which diverge at their sinuses (*Figure 2A–B*). The upper part of the corolla comprises two dorsal petals, while the lower part consists of a pair of

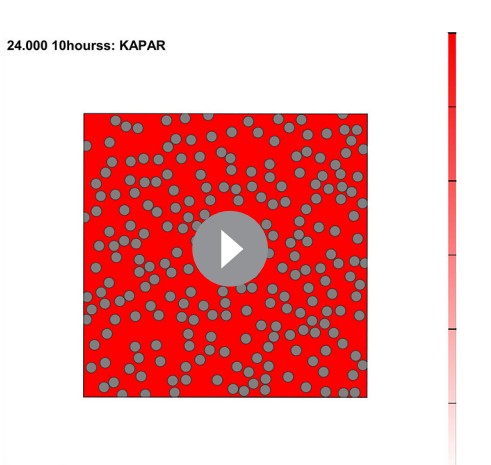

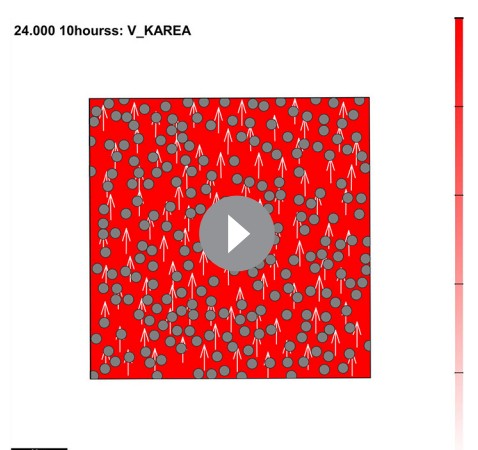

**Video 1.** Isotropic growth model as in *Figure 1B*. The size of the canvas is not rescale to better show the increase in canvas size.

**Video 2.** Anisotropic growth model as in *Figure 1D*. The size of the canvas is not rescale to better show the increase in canvas size.

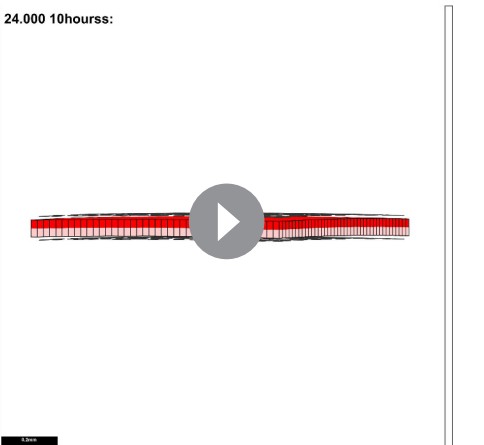

24.000 10hourss:

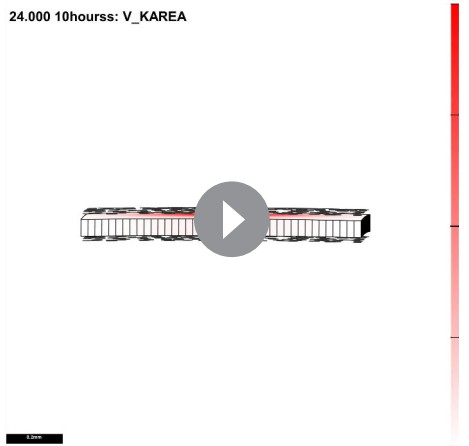

24.000 10hourss: V_KAREA

**Video 3.** Surface conflict model as in *Figure 1F*. Size of canvas is rescaled to better visualise the deformation of the canvas.

**Video 4.** Areal conflict model as in *Figure 1H*. Size of canvas is rescaled to better visualise the deformation of the canvas.

lateral petals flanking a ventral petal (*Figure 2A*). The corolla functions like a mouth, with the lower corolla articulated at a hinge to open or shut (hinge, *Figure 2B*). Several genes controlling dorsoventral asymmetry and flower shape in Snapdragon have been characterised. *CYCLOIDEA* (*CYC*), *DICHOTOMA* (*DICH*) and *RADIALIS* (*RAD*) encode dorsal-specific transcription factors that repress the ventral identity gene *DIVARICATA* (*DIV*) (*Almeida et al., 1997*; *Corley et al., 2005*; *Luo et al., 1999*, *1996*).

The most striking feature of the wild-type lower corolla is a wedge-shaped fold (*Figure 2C–D*), represented in a simplified form in *Figure 2E*. The wedge comprises a platform (palate) that slopes up towards a ridge (rim) and a descending lip (*Figure 2C–E*). The rim rises to form two small peaks at the junctions between lateral and ventral petals (foci, *Figure 2A and C–E*). *DIV* is required for the development of the wedge-shaped fold. In *div* mutants, the wedge-shape of the lower corolla is lost and two small domes form at the foci (*Figure 2F–J*).

To determine the regional origins of the wedge of wild type, we created a fate map (*Figure 3*) using morphological landmarks such as sinuses between petals (blue arrows), main veins (yellow and orange lines) and trichomes (purple ellipse, from 14 DAI = <u>D</u>ays <u>A</u>fter petal <u>I</u>nitiation) (*Figure 3A–G*).

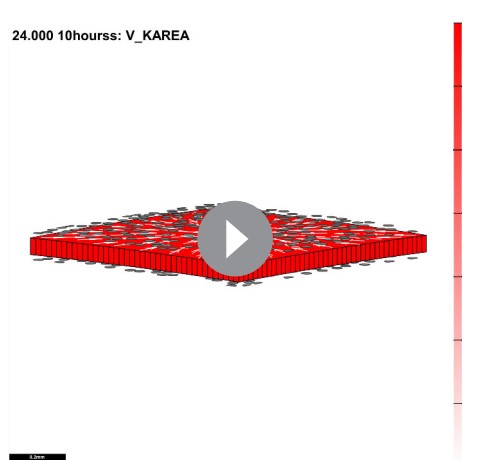

24.000 10hourss: V_KAREA

**Video 5.** Directional conflict as in *Figure 1J*. Size of canvas is rescaled to better visualise the deformation of the canvas.

The first visible sign of wedge formation was a shallow furrow proximal to the petal sinuses at 12 DAI (yellow bracket, *Figure 3B*). Prior to this, the lower corolla showed no out-of-plane deformation (*Figure 3A*). During the next 5 days, the furrow became more pronounced and the regions that will form the palate and lip could be identified as two crescents by 14 DAI (light and dark pink, *Figure 3C*). These crescents spanned the ventral petal together with half the lateral petal on each side, referred to as the flanks (boundary marked by the lateral midvein, *Figure 3A–G*). The other halves of the lateral petals grew less in width and formed the hinge (green shading, *Figure 3C–G*). Over the next seven days the crescent bulged further to form the wedge with a slope and steep lip (*Figures 3E–G* and *2C–D*). Thus, the wedge initiates from a narrow strip of tissue at the tube-lobe boundary that

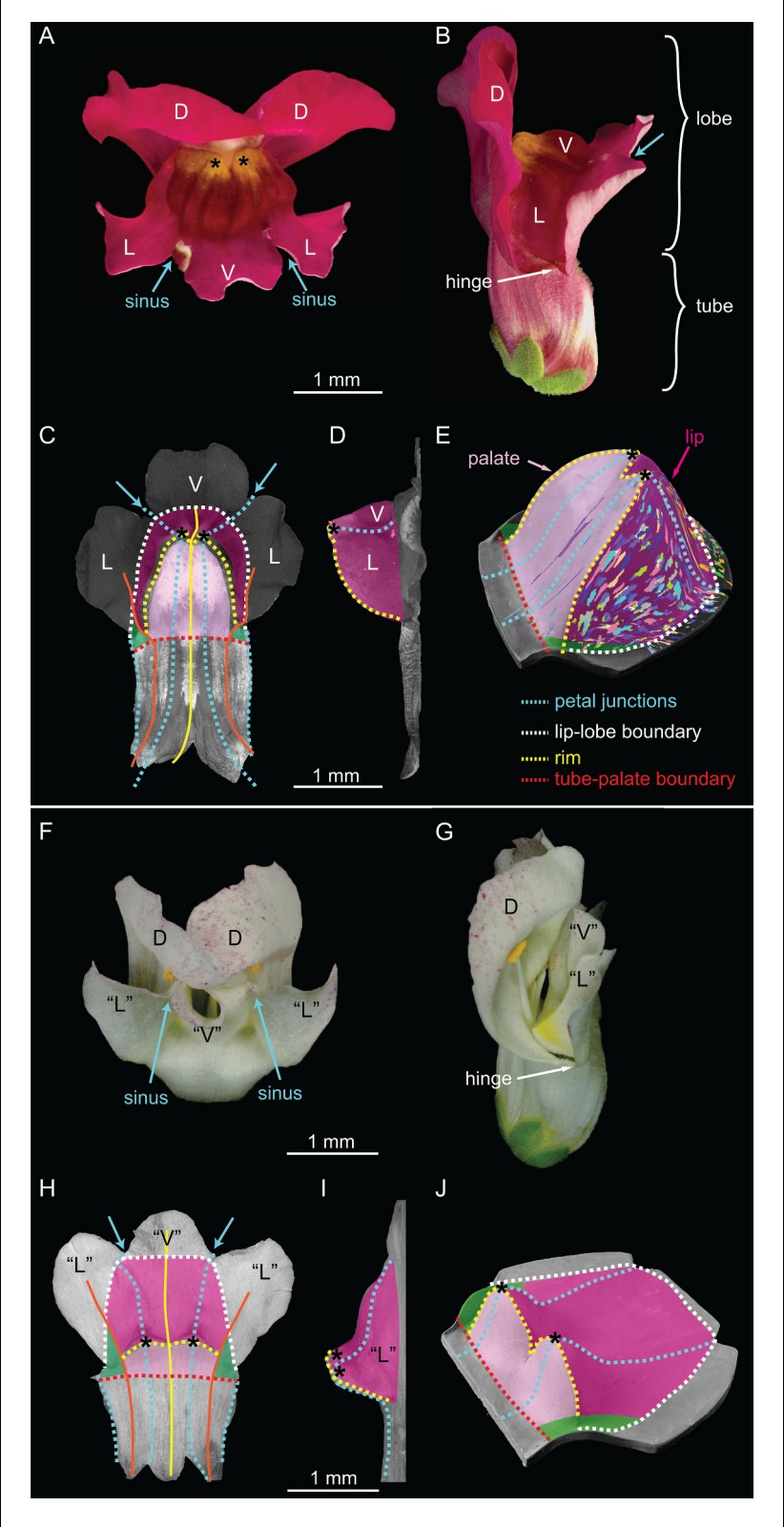

**Figure 2.** Wild-type and *div* Snapdragon corolla morphology. (**A–E**) Wild-type bilaterally symmetric Snapdragon flower, with dorsal (D), lateral (L) and ventral (V) petals (**A**). The flower has a closed mouth hinged at the dorsal to lateral sinuses (hinge). The corolla is divided into a proximal region (tube) and distal region (lobes) (**B**). Dissected and partially flattened lower petals, imaged from above (**C**, adaxial surface, i.e. inside the flower) and side (**D**)
*Figure 2 continued on next page*

*Figure 2 continued*

(black and white images are used to allow labelling of petal regions). Along the proximodistal axis, the distal wedge limit is the boundary between the lip and the distal lobe (white dashed line), and the proximal wedge limit is the boundary between the palate and the proximal tube (red dashed line). Along the mediolateral axis, the lateral petal midveins (orange lines) are the lateral wedge limits. Ventral petal midvein (yellow line), petal junctions (blue dashed lines), foci (asterisks) and petal sinuses (blue arrows) are also labelled. A simplified clay model of the wedge shape (E) illustrates the slope (palate, light pink), ridge (rim, yellow dashed line) and cliff (lip, dark pink). Experimentally induced clonal sectors (**Green et al., 2010**) are superimposed on the 3D shape (multi-coloured regions). The wedge spans the ventral petal and the ventral half of the lateral petal (flanks) while the dorsal half of the lateral petal forms a narrow region either side (hinge, green shading in C and E). (F–J) *div* mutant flower, with normal dorsal petals (D), modified lateral and ventral petals ('L' and 'V') (F), and an open flower (G). To compare the shape of the *div* mutant with wild type, the *div* mutant lower petals were dissected, flattened, imaged from above (H, adaxial side) and from the side (I) and labelled as for wild type. The *div* mutant has two domes at the foci (asterisks in H–J). A simplified clay model (J) highlights the reduction of palate (light pink shading) and the flat lip (dark pink shading).

undergoes an out-of-plane deformation through a defined series of morphogenetic events.

In contrast to wild type, the mature *div* flower has an open mouth and lacks an extended palate (*Figure 2F–G*). The ventral corolla has two small domes where the tube-lobe boundary (yellow dashed line) intersects the ventral-lateral petal junctions (blue dashed line) (*Figure 2H–J*). The centres of intersection correspond to the foci (asterisks in *Figure 2H–J*). Initially, the *div* lower petals showed no out-of-plane deformation (*Figure 3H*).The domes first became evident as small out-of-plane adaxial bulges at 14 DAI, and were clearly visible by 17 DAI (*Figure 3I–J*). At these stages, we mapped the lip region to a crescent-shaped region distal to the domes (*Figure 3I–J*).

To determine how these out-of-plane deformations might relate to cellular behaviours, we analysed the pattern of cell files in *div* and wild type, at different stages of development by staining the petal tissue with calcofluor white. Calcofluor stains older walls less brightly than younger walls, and reveals a range of different cell patterns (*Figure 4*). For example, old walls (highlighted in blue) may surround a region of cells subdivided by randomly oriented recent walls (highlighted in green or white, *Figure 4A*). Or old walls may surround a region subdivided by a ladder of perpendicular recent walls (*Figure 4B*), or a region subdivided by recent perpendicular walls (green), which are in turn further subdivided by more recent walls (white) perpendicular to these (*Figure 4C*).

To understand the origin of these patterns, we modelled cell divisions using the Growing Polarised Tissue (GPT framework), in which the tissue is treated as a connected continuous material, termed the canvas (*Kennaway et al., 2011*). This is the same framework used to model the deformations illustrated in *Figure 1*. Within the GPT framework, specified anisotropic growth is oriented by a polarity field that propagates through the canvas. Regional gene activities modify local specified growth rates parallel ($K_{par}$) or perpendicular ($K_{per}$) to the polarity field. Plant cells can be readily imposed on this framework because cell topology is preserved: once a vertex is created by division, its relation to other vertices is maintained through subsequent growth (plant cells do not move relative to each other). We assume that when the area of a cell exceeds a threshold, it is divided along the shortest path through its centroid (*Errera, 1886*; *Besson and Dumais, 2011*). The new wall is then shortened slightly to give more realistic angles (*Nakielski, 2000*). Cell wall colour and thickness reflect wall age (*Figure 4D–F*).

If specified growth was isotropic and uniform, the resulting pattern of walls resembled that of *Figure 4A*, with the blue outline highlighting the clone derived from the original cell (*Figure 4D*). Within the clone, cell walls were oriented randomly (*Figure 4D*). If specified growth was uniformly anisotropic, with growth oriented proximodistally (vertically in *Figure 4*), a pattern similar to that seen in *Figure 4B* was generated, where division walls were largely perpendicular to the main orientation of growth (*Figure 4E*). If specified growth was initially oriented proximodistally and then switches to mediolateral, a pattern similar to that in *Figure 4C* was generated (*Figure 4F*). Similar patterns of division occurred irrespective of initial cell geometries, although the shape of the final clone varied (*Figure 4—figure supplement 1*). Thus, the calcofluor pattern gives an indication of the history of cell divisions, and orientations of growth. Lines perpendicular to the cell walls reflect

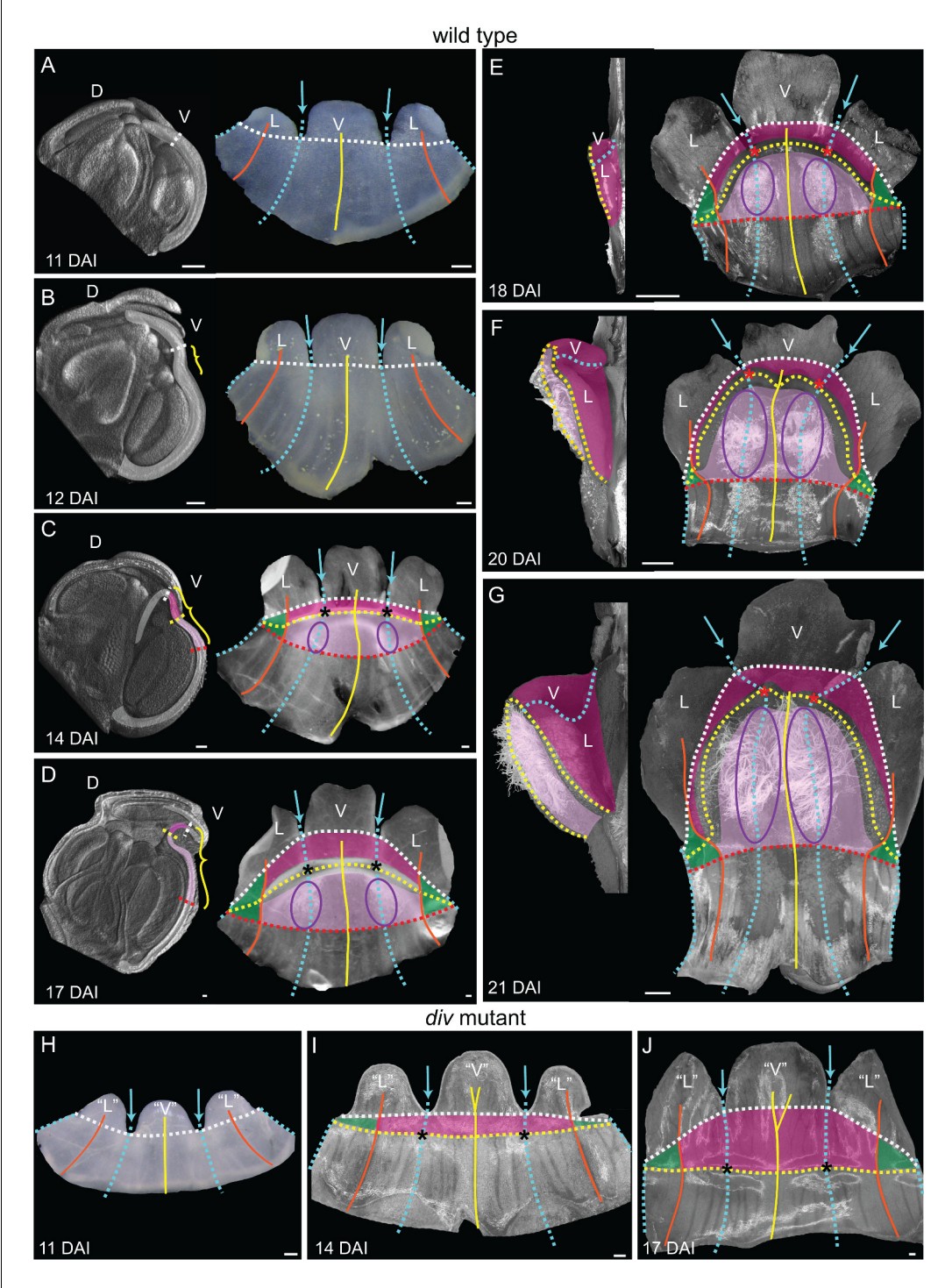

**Figure 3.** Morphogenesis and fate map of *div* and wild-type lower corollas. (A–G) Fate map of wedge emergence in the wild-type lower corolla. The 3D deformation of the lower petals was visualised by OPT (*Lee et al., 2006*) with a longitudinal midsection across the ventral petal (highlighted with white shading in left panels of A-D) and by photographing the partially flattened lower corolla (right panels), labelled with various morphological landmarks as *Figure 2*. The boundary between the palate and the proximal tube (red dashed line) can only be determined after palate trichome emergence at 14 DAI (purple ellipses). At early stages of development, 11 DAI (A) and 12 DAI (B), the lower petals develop a furrow at the rim (yellow bracket in B) although they still appear relatively flat. During the next five days (C and D), the furrow gets more pronounced and can be clearly seen at 17 DAI (yellow bracket in D). From 18 DAI, the out-of-plane deformation is visible from side (left panel in E-G) and top (right panel in E-G) views of the dissected corolla. The size of the wedge increases over the next four days (compare E to G). Scale bars, 100 μm (A–D) and 1 mm (E–G). (H–J) Fate map of dome

*Figure 3 continued on next page*

*Figure 3 continued*

emergence in the *div* lower corolla. The *div* lower corolla was dissected and partially flattened at 11 DAI (**H**), 14 DAI (**I**) and 17 DAI (**J**), and labelled with various morphological landmarks as for wild type, except for the boundary between the palate and the proximal tube, which is difficult to determine at these stages due to the lack of palate trichomes. Initially, petals are relatively flat (**H**). At 14 DAI small adaxial bulges, with foci at their tips (asterisks in **I**), can be seen at the intersection between the petal junctions and the rim. This is also the first timepoint when the lip region can be mapped (dark pink). By 17 DAI, the domes extend half way into the lateral petals (orange line in **J**). Scale bars, 100 μm.

growth oriented parallel (red lines) or perpendicular (yellow lines) to the proximodistal axis of the tissue (*Figure 4G–H*).

Calcofluor staining of developing *div* and wild-type petals showed that at 10 DAI, growth was largely oriented parallel to the proximodistal axis (red lines in *Figure 5A–B*). In the next three days, cells in regions flanking the presumptive foci showed a switch to mediolateral growth (yellow lines in *Figure 5C–D*, *Figure 5—figure supplement 1A–B and F–G*). This growth pattern continued so that by 15 DAI a clear orthogonal pattern of cell files was observed in the *div* mutant, with cell files being elongated medially at the tube-lobe boundary flanking the foci, proximodistally at the petal junctions and mostly isotropic at the intersection (*Figure 5E*). In wild type, the orthogonal pattern of cell files at 15 DAI was less clear than in *div* as the rim flanking the ventral-lateral petal showed a mixture of cell wall orientations, indicating greater proximodistal growth (compare *Figure 5F* to *Figure 5E*). There was also a region of enhanced mediolateral growth in the lateral (lilac box, *Figure 5F*) and ventral lip (green box in *Figure 5—figure supplement 1H*). In addition, diagonally oriented cell files were observed near the sinus (orange lines in blue box in *Figure 5F*). The observed pattern of growth at 15 DAI was maintained through later stages of *div* and wild-type development (*Figure 5—figure supplement 1C–D and I*) and correlated with the regions of the petal that formed out-of-plane deformations (*Figure 5—figure supplement 1E and I*). Thus, there was a switch to mediolateral growth in regions flanking the foci prior to and during the out-of-plane deformation, which was modulated by the activity of *DIV*. The timing of this switch was in line with the repatterning of growth predicted previously (*Green et al., 2010*).

To determine whether this growth repatterning was specific to the ventral-lateral junctions, we analysed cell files at the lateral-dorsal junction (*Figure 5—figure supplement 2*). An orthogonal pattern of cell files was again observed at a similar stage, but in contrast to the ventral-lateral junction, the distal arm of the orthogonal pattern did not grow. Thus, repatterning occurred at all petal junctions of the lower corolla but with varying growth dynamics.

## Orthogonal directional conflicts

The orthogonal patterns of anisotropic growth observed in wild type and *div* suggests that an orthogonal directional conflict may be involved in generating these out-of-plane deformations. To explore this idea further, we simulated orthogonal conflicts using a square initial canvas, similar to that used in *Figure 1*. We introduced several transcription factors with different expression domains into the initial canvas: a factor expressed in a vertical strip (JUN, *Figure 6A*), a horizontal strip (RIM, *Figure 6B*), the right half (HALFSIDE, *Figure 6C*) and upper half (DISTALSIDE, *Figure 6D*). We first assumed a polarity field converging at the centre, similar to that employed in *Figure 1I*. To generate an orthogonal pattern of specified anisotropy, while keeping areal growth rate uniform across the canvas (*Figure 6E*), we assumed both RIM and JUN promoted specified growth parallel to the polarity ($K_{par}$) and inhibited specified growth perpendicular to the polarity ($K_{per}$) (the ratio of $K_{par}$ to $K_{per}$ is indicated by Kaniso in *Figure 6*). Growth was specified to be isotropic where RIM and JUN overlap. This set of assumptions created an orthogonal pattern of specified anisotropic growth (red and yellow regions *Figure 6F*) next to quadrants of specified isotropic growth (green regions, *Figure 6F*). Running a model based on these assumptions led to the formation of a dome (*Figure 6G*, *Video 6*), with clones more elongated within the orthogonal domains of high specified anisotropy than in adjacent quadrants (red ellipse, *Figure 6G*). The anisotropy exhibited by the quadrants was the result of passive deformation rather than being specified directly.

The contribution of the orthogonal directional conflict to shape could be further dissected by removing components. Removing one of the regions of high specified anisotropy by introducing dependence of RIM activity on HALFSIDE, gave an asymmetric dome (*Figure 6H–I*, *Video 7*).

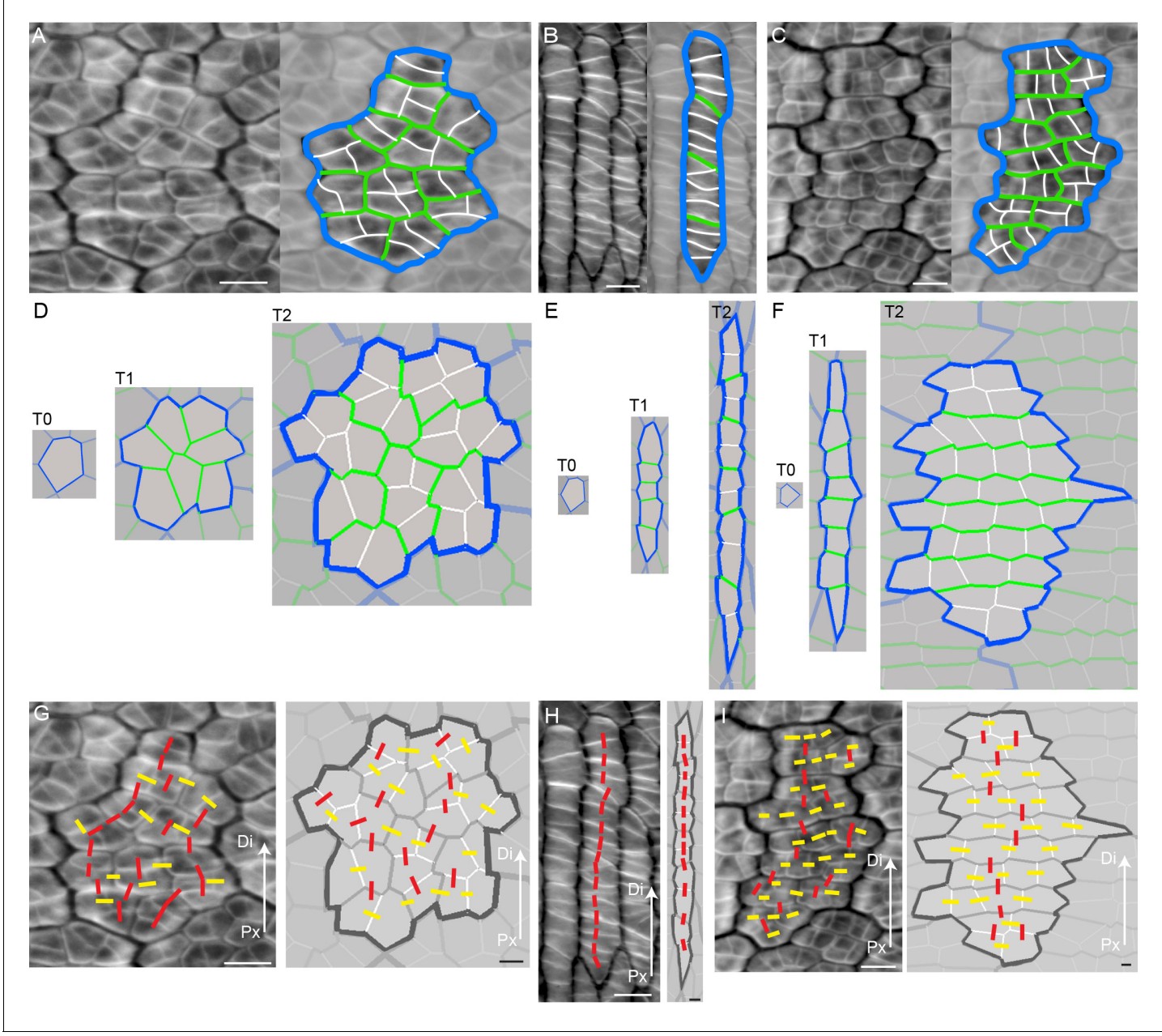

**Figure 4.** Inferring growth orientations from cell division patterns. (**A–C**) Confocal images of petal tissue showing different shapes of cell files (or clones). The cell walls were stained with calcofluor white (left panels). The most recent cell walls stain the brightest and are the thinnest (indicated by the white lines), intermediate cell walls stain less and are thicker (green lines), and the oldest cell walls are the thickest and hardly stain (blue lines). The three examples show different patterns observed on developing petals. Scale bars, 10 μm. (**D–F**) Simulated growth patterns that generate the shape of cell files (or clones) and pattern of cell divisions depicted in A–C. Original cell wall (T0) blue, cell walls formed during T1 in green (thinner than the T0 cell walls), and cell walls formed at T2 in white (the thinnest cell walls). Specified growth isotropic (**D**), oriented vertically (**E**), or oriented vertically and then switching to horizontal (**F**). Scale bars, 10 μm. (**G–I**) Inferring growth orientations from the patterns of cell division in the calcofluor data using lines perpendicular to cell division walls. For each panel, biological data is shown on left, simulation output on right. Lines were classified as roughly parallel (red) or perpendicular (yellow) to the proximodistal axis (Px-Di for data) or to the orientation of specified growth (simulations). The simulations show output for isotropic growth (**G**) vertically oriented specified growth (**H**) or vertical followed by horizontally oriented specified growth (**I**). Scale bars, 10 μm.

The following figure supplement is available for figure 4:

**Figure supplement 1.** Model output of cell division patterns generated from different specified growth patterns with various initial cell geometries.

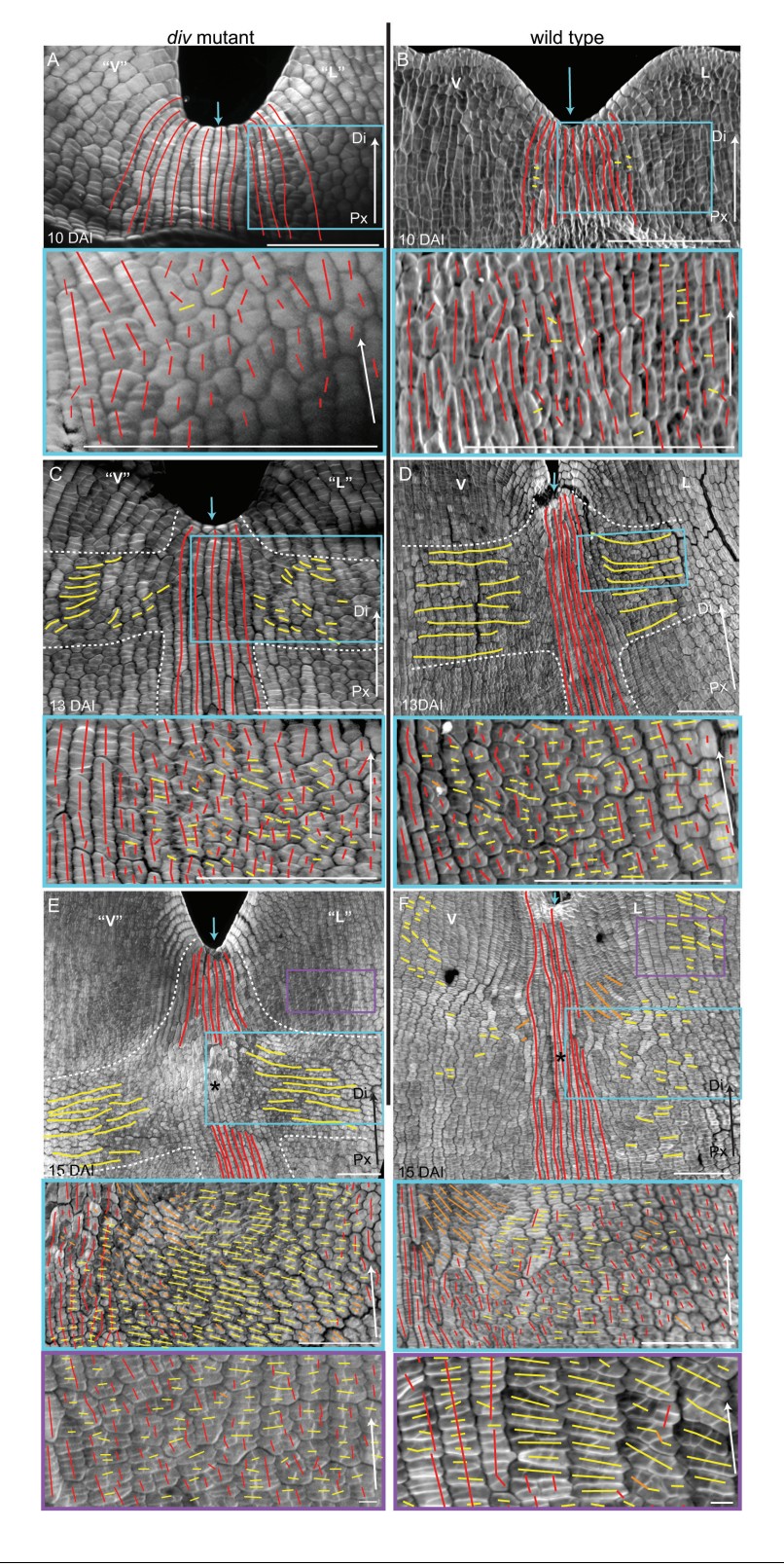

**Figure 5.** Pattern of cell file orientations at the ventral-lateral junctions during *div* and wild-type development. (A–F) Confocal images of *div* and wild-type ventral-lateral (V–L) junctions at 10 DAI (A and B), 13 DAI (C and D) and 15 DAI (E and F). The tissue was stained with calcofluor white to visualise the patterns of cell division and infer growth orientations as described in *Figure 4G–H*: proximodistal growth (red lines), mediolateral growth (yellow lines) and

*Figure 5 continued on next page*

*Figure 5 continued*

diagonal growth (orange lines). Each stage is shown in two magnifications: an overview of the junction region and a zoomed-in region (coloured boxes in **A**-**F**). Only regions showing the clearest cell files and oriented patterns of division are shown in the overview images. The patterns of cell files at the *div* and wild-type V-L junction are mainly proximodistal at 10 DAI (**A** and **B**) and become increasingly mediolateral in the rim regions flanking the junction forming an orthogonal pattern of cell files by 13 DAI (white dashed lines in **C** and **D**). At 15 DAI, the mediolateral region expands and together with the proximodistal files at the junction form a clear orthogonal pattern of growth orientations in *div* (white dashed lines in **E**). In wild type, the orthogonal pattern is not as clear as the rim region flanking the foci and shows a mix of mediolateral and proximodistal growth (blue box in **F**) but extends to the lateral lip, where predominantly mediolateral growth is observed (purple box in **F**) in contrast to the mixed proximodistal and mediolateral growth in the *div* lateral lip (purple box in **E**). Arrow at the lower right corner of each. panel indicates Proximal (Px) and Distal (Di) axis. Scale bar 100 μm, except for purple boxes of **E** and **F** where scale bar is 10 μm.
The following figure supplements are available for figure 5:

**Figure supplement 1.** Pattern of cell file orientations at the ventral-lateral junctions during *div* and wild-type development.
**Figure supplement 2.** Pattern of cell file orientations at the lateral-dorsal junctions during *div* and wild-type development.

Removing two regions of anisotropy, by making JUN activity dependent on DISTALHALF, and RIM activity dependent on HALFSIDE (*Figure 6J*), produced a less pronounced asymmetric dome (*Figure 6K*, *Video 8*). Removing both horizontal regions of high anisotropy by removing RIM activity, gave a ridge (*Figure 6L–M*, *Video 9*). Removing all regions but one, by making specified anisotropy dependent on the combination of JUN with DISTALHALF, resulted in a slightly arched and symmetric ridge (*Figure 6N–O*, *Video 10*). Thus, various types of out-of-plane deformation may be generated by varying the pattern and extent of directional conflict.

To determine other ways of generating orthogonal directional conflicts, we considered a parallel rather than convergent polarity field. An orthogonal conflict could be generated by JUN promoting $K_{par}$ (and inhibits $K_{per}$), while RIM promoted $K_{per}$ (and inhibits $K_{par}$) (*Figure 6P*). A dome was again generated, with the polarity field being deformed by growth so that it passed over and around the dome (*Figure 6Q*, *Video 11*). Virtual clones were oriented parallel to the polarity in the JUN domain (red ellipse, *Figure 6Q*) and perpendicular to the polarity in the RIM domain (black ellipse in *Figure 6Q*).

In all the above cases, the axial information provided by the polarity field was critical for the resulting shape. Boosting isotropic growth in an orthogonal pattern (areal conflict) generated a series of bulges rather than a dome (*Figure 6R–S*, *Video 12*).

Orthogonal directional conflict resolution thus provides a mechanism for generating out-of-plane deformations, and can be specified in several ways through a combination of orthogonal regional identities and polarity fields (convergent or parallel). Further dome shapes can be generated by combining the orthogonal directional conflict with areal and surface conflicts (*Figure 6—figure supplement 1A–C*), or by combining surface and areal conflicts (*Figure 6—figure supplement 1D*).

In the above examples, we modelled deformations as arising through differential growth. In animals, contractions as well as growth may be involved in generating out-of-plane deformations. In early animal embryogenesis it is common for deformations to occur without overall change in size. To determine how principles of tissue conflict resolution could apply in these contexts, we modelled deformations that preserve overall size. For these purposes, contraction could be considered as equivalent to negative growth.

For surface conflict, high specified isotropic growth of the upper surface was counterbalanced by an equal negative specified growth rate (contraction) for the lower surface, and gave a folded shape similar to that based on differential growth (compare *Figure 7A–B* to *Figure 1E–F*). For areal conflict, negative specified isotropic growth at the periphery was counterbalanced by high specified growth in the centre, and gave a shape again similar to that based on increased growth alone (compare *Figure 7C–D* to *Figure 1G–H*). For directional conflict, enhanced growth parallel to polarity

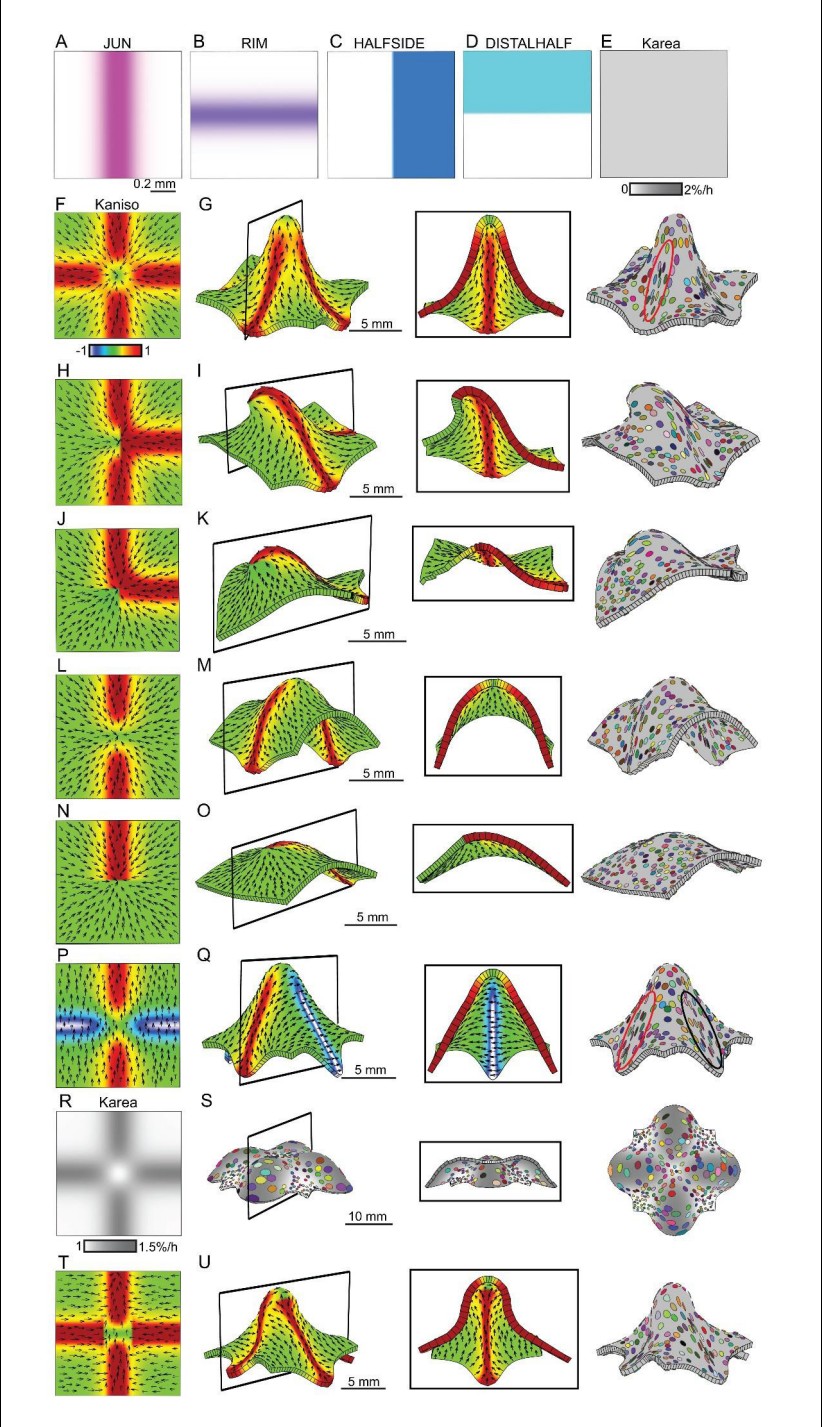

**Figure 6.** Generation of domes through orthogonal tissue conflicts. (A–D) Expression pattern of four growth regulatory factors. JUN is expressed as a vertical domain in the middle of the canvas (A), RIM as a horizontal line in the middle of the canvas (B), HALFSIDE in the right side of the canvas (C) and DISTALHALF in the upper half of the canvas (D). In all models the $K_{area} = (K_{par} + K_{per})$ was maintained uniform throughout the canvas (E). (F–O) Orthogonal directional conflict models. In the first example $K_{par}$ is enhanced in an orthogonal domain established by JUN and RIM but not at the intersection of these factors (F) to generate a dome with clones more elongated in the arms of the orthogonal domains (red ellipse) than in the neighbouring quadrants (G). Modifications to this orthogonal pattern generated variously shaped domes (H–O). A T-shaped pattern (H) generated an asymmetric dome (J). An L-shaped pattern (J) gave a less pronounced asymmetric dome (K), while removing both side arms of high anisotropy (L), gave a ridge with clones more elongated along it (M). Removing all arms but one gave an

*Figure 6 continued on next page*

*Figure 6 continued*

asymmetric ridge (N–O). (P–U) An orthogonal pattern of directional conflict in a parallel polarity field, generated by boosting $K_{par}$ by JUN while boosting $K_{per}$ by RIM (P). This specified growth pattern generated an elongated dome with clones elongated parallel to the polarity along the regions of high $K_{par}$ (red ellipse in Q) and perpendicular to the polarity along the regions of high $K_{per}$ (black ellipse in Q). Between these regions clones are more isodiametric (Q). Boosting isotropic growth with JUN and RIM ($K_{area}$ in R) resulted on the formation of four bulges but no dome (S). An orthogonal pattern of directional conflict can also be generated with a channel of proximodistal polarity in the JUN domain, and mediolateral polarity in flanking regions (T). An orthogonal domain of high $K_{par}$ generates a wide dome (U). Kaniso = ln ($K_{par}$ /$K_{per}$). The colour scale for Kaniso is –1 to +1.

The following figure supplement is available for figure 6:

**Figure supplement 1.** Combining tissue conflicts.

(positive $K_{par}$) was counterbalanced by negative specified growth perpendicular to the polarity (negative $K_{per}$), and gave a similar result to overall positive growth (compare *Figure 7E–H* to *Figure 1I–J* and *Figure 6P–Q*). Such a balance between growth in one orientation and contraction in the perpendicular orientation is equivalent to the process of convergent-extension (*Keller et al., 2008*). Surface, areal and directional conflict resolutions, thus provide possible mechanisms for generating out-of-plane deformations with or without overall growth.

## *DIV* modulates PIN1 activity

Given the cell file data, we hypothesise that orthogonal directional conflict is involved in generating the out-of-plane deformation in *div* and wild type. This hypothesis requires a polarity field for orienting the growth in each region. To test whether a polarity field is present and determine its pattern of orientations, we analysed PIN1 auxin transport protein distribution as this is a possible readout of cell polarity (*Steinmann et al., 1999*). The Snapdragon genome has 11 *PIN* genes, three of which are PIN1 orthologues (*PIN1a*, *PIN1b* and *PIN1c*). RNA in situ hybridisation revealed that the three *PIN1* genes showed detectable expression in developing wild-type petals, with all PIN1 orthologues strongly expressed in the vascular tissue and epidermis in the palate region of the lower petals (*Figure 8—figure supplement 1*). To determine protein localisation, we raised peptide antibodies against each of the three proteins, as well as a generic antibody against a conserved domain in all three. The antibody raised to the PIN1a-based peptide gave the clearest signal. As this peptide shares some amino acids with the other PIN1s, we cannot be sure that the signal derives entirely from PIN1a and therefore refer to the signal obtained as PIN1 signal. Immunolocalisations on sec-

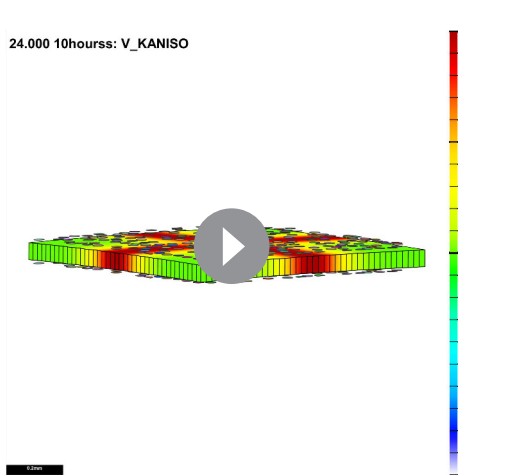

**Video 6.** Orthogonal directional conflict model as in *Figure 6G*. Size of canvas is rescaled to better visualise the deformation of the canvas.

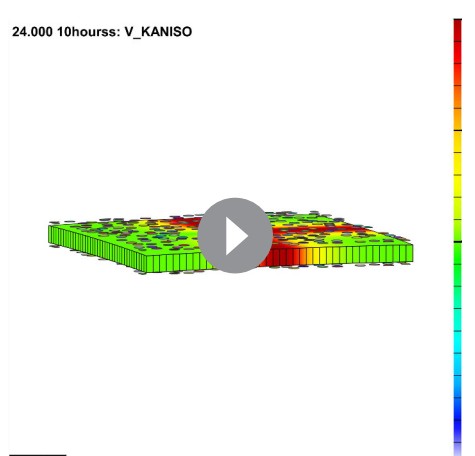

**Video 7.** T-shaped orthogonal directional conflict model as in *Figure 6I*. Size of canvas is rescaled to better visualise the deformation of the canvas.

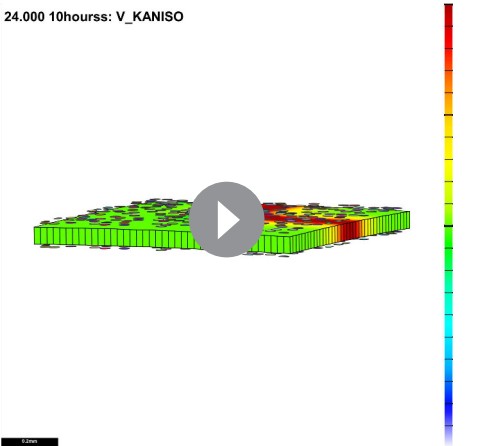

**Video 8.** L-shaped orthogonal directional conflict model as in *Figure 6K*. Size of canvas is rescaled to better visualise the deformation of the canvas.

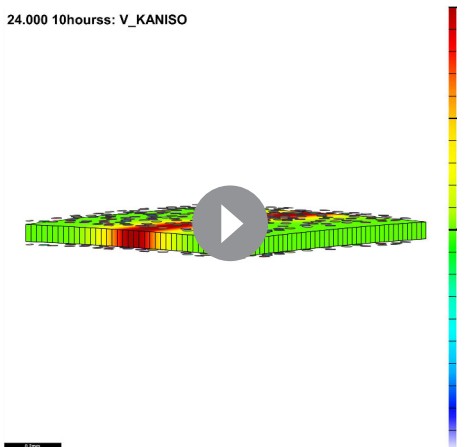

**Video 9.** I-shaped directional conflict model as in *Figure 6M*. Size of canvas is rescaled to better visualise the deformation of the canvas.

tioned tissue showed that PIN1 signal is localised at the distal end of epidermal cells in early wild-type petal primordia, with abaxial and adaxial polarity converging at the tip of the petal primordia (day 9, *Figure 8—figure supplement 2A–C*). This is in agreement with PIN1 patterns observed in emerging primordia in other species (*O'Connor et al., 2014*; *Reinhardt et al., 2003*).

To analyse PIN1 polarity in regions undergoing out-of-plane deformations, we developed a whole-mount immunolocalisation protocol for plant tissue sheets, and software to quantify the distribution of PIN1 signal for each cell (PinPoint software). Prior to the repatterning of corolla growth in *div* and wild type, PIN1 is strongly expressed near the petal sinuses and margins, and is oriented proximodistally (*Figure 8A–D* and *Figure 8—figure supplement 2D–E*). This pattern extends over more cells proximally when the orthogonal pattern of cell files begins to form (13 DAI, *Figure 8E–H* and *Figure 8—figure supplement 2F–G*), overlapping with the distal region showing proximodistal cell files (*Figure 8I–J*). PIN1 is more extended in wild type than in *div*. Within the extended region, polarity is oriented proximodistally along the petal junction (3–4 central files within green dashed line in *Figure 8G*, *Figure 8—figure supplement 2G*). Either side of this, polarity points diagonally, towards the central files and nearby margins (*Figure 8G*, *Figure 8—figure supplement 2G*). In both

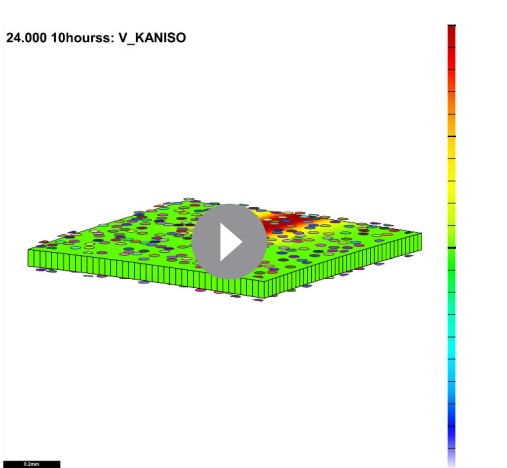

**Video 10.** One arm directional conflict model as in *Figure 6O*. Size of canvas is rescaled to better visualise the deformation of the canvas.

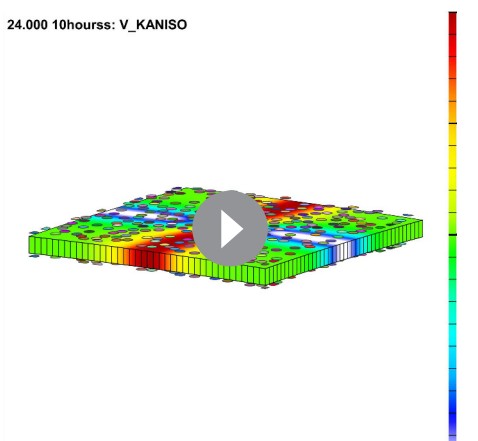

**Video 11.** Orthogonal directional conflict model as in *Figure 6Q*. Size of canvas is rescaled to better visualise the deformation of the canvas.

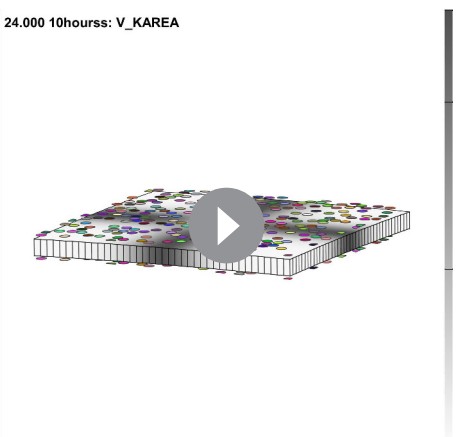

**Video 12.** Orthogonal areal conflict model as in *Figure 6S*. Size of canvas is rescaled to better visualise the deformation of the canvas.

*div* and wild type, the region with diagonal PIN1 polarity corresponds to that showing diagonal cell files (yellow boxes, *Figure 8I–J*). Strong PIN1 signal was not observed in the regions showing mediolateral cell files (orange boxes in *Figure 8I–J*). By 15 DAI, PIN1 expression domain became restricted to the petal margins (*Figure 8—figure supplement 2H–I*). The upregulation of PIN1 in wild type was not observed at the lateral-dorsal sinuses (*Figure 8—figure supplement 2J–K*), where *DIV* was not strongly expressed at this stage (*Galego and Almeida, 2002*). Thus, PIN1 reveals an early proximodistal polarity pattern near the sinus which is modulated by *DIV* to become extended and deflected during repatterning.

## Orthogonal directional conflict plays a key role in morphogenesis of *div*

In light of the data above on fate mapping, cell files and PIN1 localisation, we modelled corolla morphogenesis, beginning with the simpler deformation observed in *div*. Rather than modelling the entire *div* corolla, we first modelled the region forming the out-of-plane deformation.

Based on the fate map experiments, the *div* and wild-type out-of-plane deformations develop from a curved strip of tissue at the tube-lobe boundary, encompassing the presumptive palate and lip (*Figure 3*). We therefore used a strip of canvas as the initial configuration for our minimal model (*Figure 9A*). The strip was convex on the abaxial and concave on the adaxial surface, as observed for this region of the corolla before out-of-plane deformation. Growth rates were modulated by regional identity factors distributed along three axes – proximodistal, mediolateral and dorsoventral – similar to those employed previously (*Green et al., 2010*). The dorsoventral factor was a diffusible signal derived from the dorsal region, sRAD, and specified the hinge region of the lateral petal (red shading in *Figure 9A*). The proximodistal factors included PALATE (PLT), RIM and LIP (*Figure 9A*), which corresponded to distinct regions and cell types in the flower (*Keck et al., 2003*). RIM was a diffusible signal that activated the expression of BRIM to subdivide the regions of PLT and LIP expression (*Figure 9A*). The mediolateral factors LAT and MED corresponded to the petal junctions and midvein regions, respectively (*Figure 9A*). All of these factors interacted combinatorially to control the specified rates of growth parallel ($K_{par}$) and perpendicular ($K_{per}$) to the polarity field (details of interactions are given in Supplementary Material and methods). As PIN1 cellular localisation was only observed around the sinus and was mainly proximodistal, we made the simplest assumption that polarity is maintained as proximodistal throughout the canvas (black arrows in *Figure 9B*). Alternatively, polarity outside the PIN1 upregulation domain may reorient to point towards the petal junction to create a region of mediolateral polarity flanking a channel of proximodistal polarity (*Figure 6T–U*, *Video 13*).

For our minimal model of *div*, growth of the initial strip was specified to be higher parallel than perpendicular to the polarity (ratio of $K_{par}$ to $K_{per}$ is indicated by Kaniso in *Figure 9B*). $K_{par}$ was also inhibited on the lower (abaxial) surface in the rim to promote out-of-plane bending (surface conflict, *Figure 9C*). Specified growth was further inhibited at the edges of the strip (hinge) through the activity of a dorsoventral gene (sRAD) (areal conflict, *Figure 9D*). $K_{par}$ was also promoted in the distal lip (areal and directional conflict, *Figure 9D*). These assumptions led to the generation of a bilaterally symmetric canvas with short lateral edges by 11.5 DAI (*Figure 9E*). Orthogonal directional conflict was introduced at 12 DAI through combinatorial interactions that enhanced $K_{par}$ relative to $K_{per}$ at the petal junctions, and enhanced $K_{per}$ relative to $K_{par}$ along the rim (Kaniso panel in *Figure 9E* showing orthogonal pattern marked by dashed white lines). At the hinge, growth in the distal part of the orthogonal domain was inhibited by sRAD (red dashed lines in *Figure 9E*). In

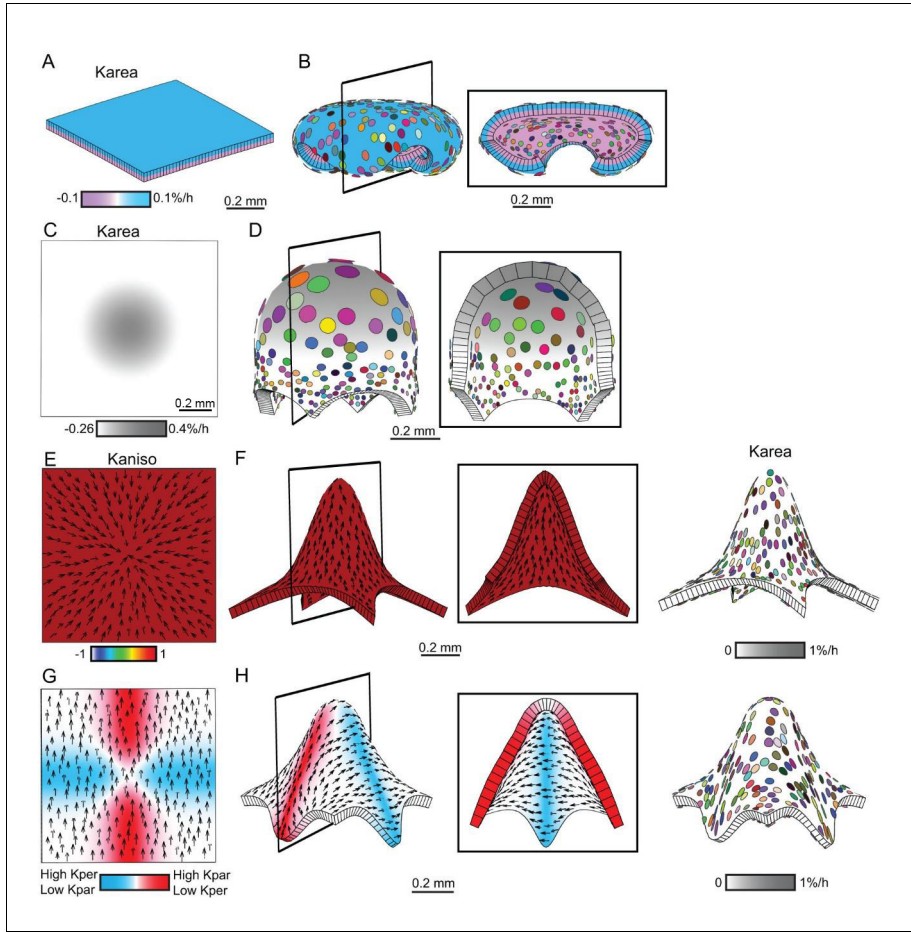

**Figure 7.** Generation of domes by tissue conflicts through contraction and growth. Similar shape domes to *Figure 1* can be generated using a combination of contraction and growth, with overall canvas size remaining the same. (A–B) Surface conflict. Uniform isotropic specified growth for one surface (blue) and isotropic specified contraction (negative growth) for the other surface (lilac), causes an initial square (A) to deform into a dome with downward curled edges (B). (C–D) Areal conflict. Isotropic specified growth in the centre counterbalanced by contraction at the edges causes the square canvas (C) to deform into a rounded dome with circular clones which are bigger at the apex (D). (E–F) Directional conflict with a convergent polarity field. Uniformly high specified growth parallel to the polarity is counterbalanced by uniformly high specified contraction perpendicular to the polarity. The canvas deforms from an initial square (E) into a dome with elongated clones parallel to the polarity field (F). (G–H) Orthogonal directional conflict with a parallel polarity field. High specified growth parallel to the polarity in the vertical domain (red) and perpendicular to the polarity in the horizontal domain (blue) are counterbalanced by contraction perpendicular to the polarity in the vertical domain and parallel in the horizontal domain (G). A dome is generated with clones elongated parallel to the polarity in the red domain and perpendicular to the polarity in the blue domain (H). Kaniso = $\ln(K_{par}/K_{per})$. The colour scale for Kaniso is –1 to +1. $K_{area} = K_{par} + K_{per}$.

addition, $K_{per}$ was inhibited by MED (*Figure 9E*). Running this model generated two domes centred on the foci, and an extended lip, matching the shape observed in *div* (*Figures 9F* and *2J*, *Video 14*).

This model involved three types of conflict: areal, directional and surface. To evaluate the contribution of each type of conflict to the final shape, we removed each individually. Removing surface conflict (by equalising growth rates on the two surfaces) led to the out-of-plane deformations protruding abaxially instead of adaxially (*Figure 9G* and *Video 15*). This was because the direction of bulging reflected the curvature of the initial strip of tissue, which was abaxially convex (*Figure 9A*). Removing areal conflicts, by normalising the growth rates across the canvas, resulted in a crumpled

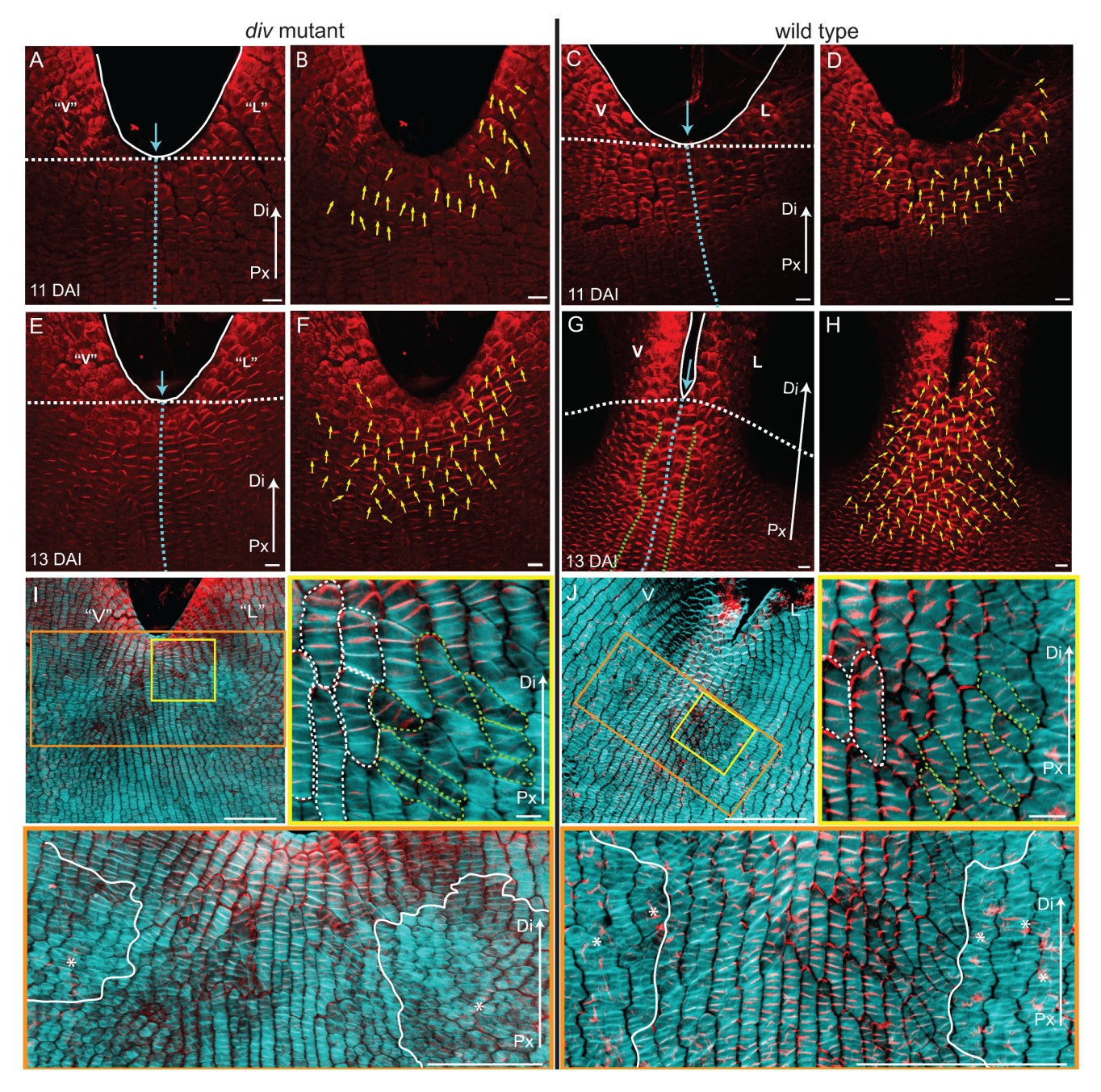

**Figure 8.** PIN1 polarity and correlation with cell file orientation. (A–H) Whole-mount immunolocalisation in *div* and wild-type ventral-lateral junctions using the antibody raised to the PIN1a peptide at 11 DAI (A–D) and 13 DAI (E–H). Left panels show confocal images of PIN1 immunolocalisation (red signal). Right panels depict the average PIN1 polarity (yellow arrows) calculated using the PinPoint software. At 11 DAI, PIN1 is expressed at the ventral-lateral petal junction (blue dashed line) just below the sinus (blue arrow) and points proximodistally towards the petal margin (white outline) in both *div* (A and B) and wild type (C and D). At 13 DAI, PIN1 expression extends more proximally in *div* (E and F) and to an ever greater extent in wild type (G and H). The PIN1 polarity in wild type has a central region of 2–3 proximodistal files (within green dashed line in G) and flanking regions with diagonal polarity deflected towards the central files (G and H). Scale bar, 10 μm. Px, proximal, Di, distal. (I and J) Confocal images of *div* (I) and wild-type (J) ventral-lateral junctions at 13 DAI combining the immunohybridised PIN1a antibody signal (red) with the calcofluor white cell wall signal (cyan) at different magnifications (yellow and orange boxes). When overlapping, the orientation of the cell files correlates with the pattern of cellular PIN1 polarity: proximodistal cell files have proximodistal PIN1 polarity (e.g. files marked with white dashed lines in yellow boxes) while diagonal cell files have diagonally oriented PIN1 polarity (e.g. files marked with green dashed line in yellow boxes). The region of PIN1 expression does not overlap with the

*Figure 8 continued on next page*

*Figure 8 continued*

region where the mediolateral growth is observed (white outlines in oranges boxes). Asterisks refer to subepidermal PIN1 signal in the vascular tissue. Scale bar (**E** and **F**), 100 μm and (**G**) 10 μm.

The following figure supplements are available for figure 8:

**Figure supplement 1.** Expression patterns of *AmPIN1* genes.

**Figure supplement 2.** PIN1 polarity patterns in *div* and wild type.

strip with long hinge regions (*Figure 9H*, *Video 16*). Removing all directional conflict by eliminating polarity, while maintaining the patterns of areal and surface conflicts, generated a curved strip (*Figure 9I*, *Video 17*). Removing orthogonal directional conflict by removing the enhanced mediolateral growth at the rim generated a relatively flat shape with a bend at the rim (*Figure 9J*, *Video 18*). Thus, the out-of-plane deformation of *div* depends on all three types of conflict, with orthogonal directional conflict playing a key role in creating the domes.

The interactions from the above minimal model were incorporated into a model of the entire Snapdragon corolla to see if they could account for the overall final shape. We first modified the published full corolla model (*Green et al., 2010*) to create a ground state for the lower petals (see Supplementary Material and methods). This produced a corolla with a wild-type looking upper corolla, but a lower corolla lacking out-of-plane deformations (*Figure 9—figure supplement 1A*). Incorporating the interactions of the minimal *div* model above into this ground state produced a corolla with an open mouth, a reduced palate, two domes tipped by the foci and a flat extended lip, similar to the *div* phenotype (*Figure 9K* 2F, *Video 19*). As with the minimal model, removing orthogonal directional conflict generated a corolla with a bend at the rim for the lower corolla rather than two domes (*Figure 9—figure supplement 1B*).

### *DIV* modulates directional conflict

To explore the role of DIV, we incorporated its activity within the minimal model. In accordance with previous experimental data, we assumed that *DIV* expression was graded, from low levels at the lateral-dorsal junctions to high levels in the ventral petals (*Figure 10A*) (*Galego and Almeida, 2002*). Unlike the *div* mutant, the wild-type corolla has an extended palate and lip (forming the slope of the wedge, *Figure 2E*). We therefore assumed that *DIV* promotes $K_{par}$ in the palate and lip regions from an early stage (Kaniso, *Figure 10B*; $K_{area}$, *Figure 10D*). *DIV* also enhanced surface conflict by increasing the extent of differential growth parallel to the polarity between the two surfaces near the rim, consistent with differential expression of cell division genes in this region (*Figure 10D*; (*Gaudin et al., 2000*)). These assumptions generated a canvas similar to the div model at 11.5 DAI but with an extended palate (compare 11.5 DAI, *Figure 10E* with *Figure 9E*). To simulate the observed deflection of polarity towards the wild-type sinus at 12 DAI, we assumed *DIV* activates expression of a new polariser sink at the sinus (blue arrows, *Figure 10E*). To account for the observed patterns of orthogonal cell files in the wild type, we assumed that at 12 DAI *DIV* extends the region of high $K_{per}$ flanking the foci into the lip region (white dashed lines, *Figure 10E*), enhancing directional conflict. $K_{per}$ was also inhibited at the distal and proximal boundaries as a proxy for the restraining effect of adjacent tissue.

With these assumptions, a wedge shape was generated that resembled that observed in wild type (*Figure 10F*, *Video 20*). The pattern of growth could be compared with that of a wild-type flower by generating virtual clones, induced at 12 DAI. The pattern obtained was similar to that seen by mapping experimentally induced clones on the final wedge shape (*Figure 2E*). In both cases, clones were predominantly proximodistal throughout the palate and distal lip regions, but mediolateral in the flanking lip regions (compare *Figure 10F* to *Figure 2E*).

As with the minimal model of *div,* we evaluated the contribution of each type of conflict to the final shape. Removing surface conflict led to the out-of-plane deformation protruding abaxially instead of adaxially (*Figure 10G*, *Video 21*). Removing areal conflicts resulted in a crumpled strip with long hinge regions and protruding domes with a narrow palate (*Figure 10H*, *Video 22*).

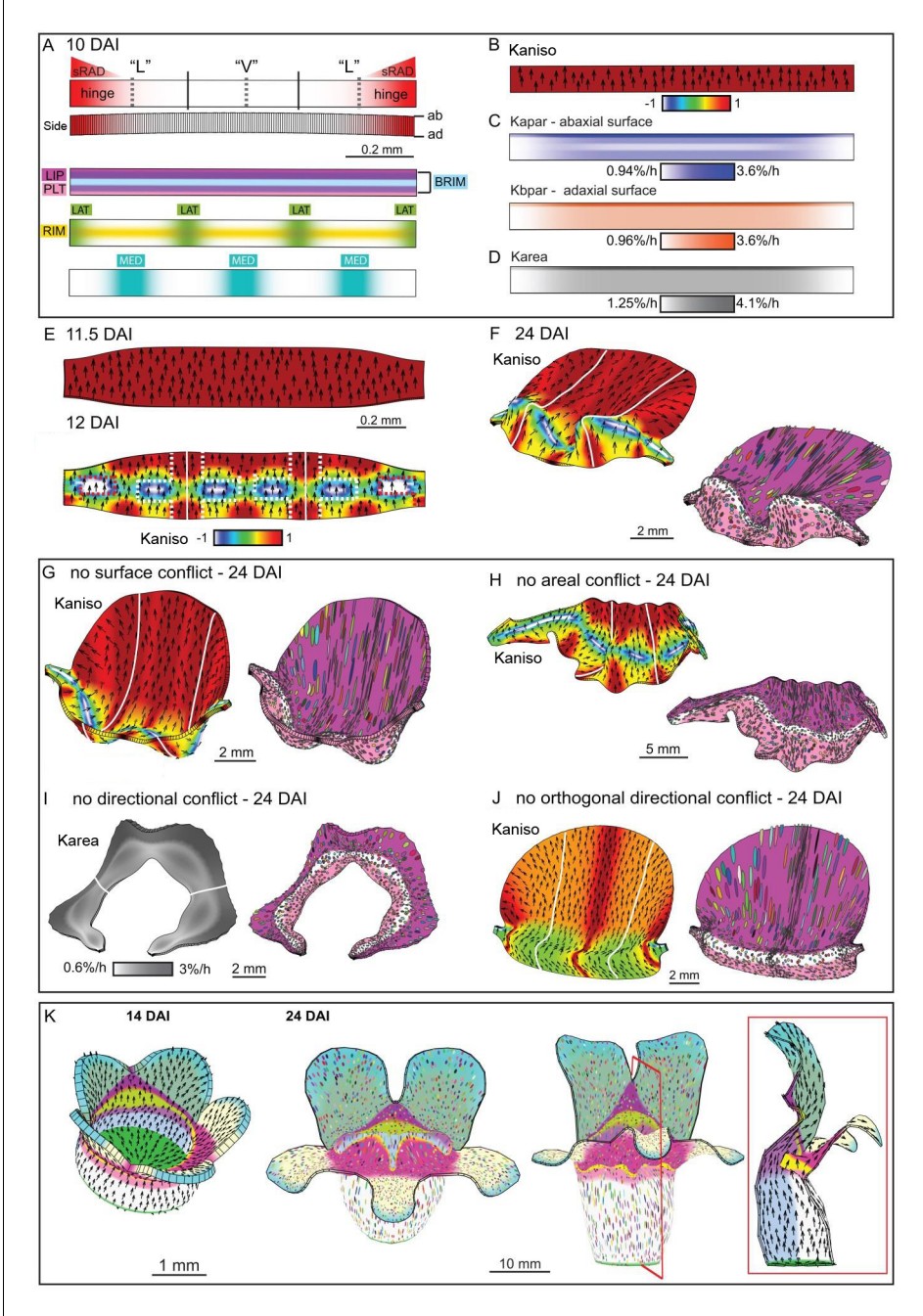

**Figure 9.** Tissue-level model of *div* corolla development. (A–F) Modelling formation of *div* domes. (A) The initial shape represents a strip of tissue at the tube to lobe boundary, from which the domes develop (*Figure 3H–J*). The strip is slightly curved to represent the initial petal shape, as shown in the side view of the canvas thickness (Side). During the setup phase, several identity factors are established in domains along three different axes: (1) dorsoventral – sRAD which grades into the lateral petals from the dorsal petals, (2) proximodistal - LIP, PLT, RIM and BRIM and (3) mediolateral - LAT at petal junctions, and MED in the midvein region. (B) These factors interact combinatorially to control the specified rates of growth parallel ($K_{par}$) and perpendicular ($K_{per}$) to the proximodistal polarity field (black arrows). Growth is initially higher in $K_{par}$ than in $K_{per}$ (Kaniso). (C) Surface conflict introduced by differential growth at the rim. Kapar is specified $K_{par}$ on the abaxial surface of the canvas, whereas Kbpar is specified $K_{par}$ on the adaxial surface. The lower value of Kapar relative to Kbpar in rim region will lead to bending of the canvas at the rim. (D) Growth rate is higher in the LIPDISTAL region (see darker grey in the distal region of the canvas), and inhibited at the hinge region by sRAD (lighter shade of grey at the lateral edges of canvas). (E) The above pattern of growth produces a strip with short hinge region at 11.5 DAI just before the time of

*Figure 9 continued on next page*

*Figure 9 continued*

repatterning. At 12 DAI, an orthogonal pattern of directional conflict is introduced. $K_{par}$ is enhanced by LAT around the petal junctions (white lines) and inhibited by RIM, while $K_{per}$ is enhanced by RIM and inhibited by MED and LAT (the orthogonal pattern of anisotropy is marked by dashed white lines for the lateral-ventral junctions, and red dashed lines for the lateral-dorsal junction). (F) Growth following repatterning leads to a shape at 24 DAI similar to the *div* lower lip, with two domes at the foci region and an extended asymmetric lip (left panel shows Kaniso, while right panel shows the LIP and PLT regional factors and virtual clones). Virtual clones are predominantly elongated along the proximodistal domain except in the RIM region where clones are mediolateral oriented. (G–J) Evaluating the contribution of growth conflicts to the shape of the *div* model. (G) Removal of surface conflict produced a similar shape as in F but the out-of-plane deformations protrude towards the abaxial side instead of the adaxial side (left panel shows Kaniso, while right panel shows the LIP and PLT regional factors and virtual clones). (H) Removal of areal conflict, by normalising the areal growth across the canvas while maintaining the anisotropy, generated a curved canvas with long hinge region and abaxial protruding foci. (I) Removing all directional conflict by setting polariser to zero, so that all growth is isotropic, resulted in a curved narrow strip with slight bulges at the region of the high growth in the *div* (dark grey areas around the petals junctions. (J) Removing only the orthogonal directional conflict, resulted in a canvas with a broad flat lip and a uniform bend at the RIM without domes. (K) Incorporating the above *div* patterns of areal, surface and directional conflict into the full Snapdragon model produced a *div* mutant corolla with an extended lip (dark pink), short palate (pink) and two domes at the RIM (yellow) centred on the foci. Canvas shown at 14 DAI and maturity (24 DAI). Kaniso = ln($K_{par}/K_{per}$). The colour scale for Kaniso is –1 to +1. $K_{area} = K_{par} + K_{per}$.

The following figure supplement is available for figure 9:

**Figure supplement 1.** Virtual mutants in the Snapdragon model.

---

Removing all directional conflict by eliminating polarity generated a curled canvas with two large domes in the lateral petal (*Figure 10I*, *Video 23*). Reducing orthogonal directional conflict by removing the enhanced mediolateral growth at the rim generated a sharp bend at the rim and a narrow lip region (*Figure 10J*, *Video 24*). Thus, as for *div*, morphogenesis of wild type depends on all three types of conflict, with directional conflict playing an essential role.

Integrating the activity of *DIV* into the full corolla model of *div* described above generated a corolla with a typical Snapdragon shape (*Figures 10K* and *2A*, *Video 25*). Removing dorsoventral gene activities (i.e. *DIV*, *CYC*, *DICH*) from the model generated phenotypes resembling those of the respective mutants (*Figure 11A–D*, *Videos 26* and *27*). These phenotypes were distinct from virtual mutants obtained by removing each tissue conflict from the wild-type model (*Figure 9—figure supplement 1C and E–F*). Removing the deflection of polarity at the sinus resulted in an overarching

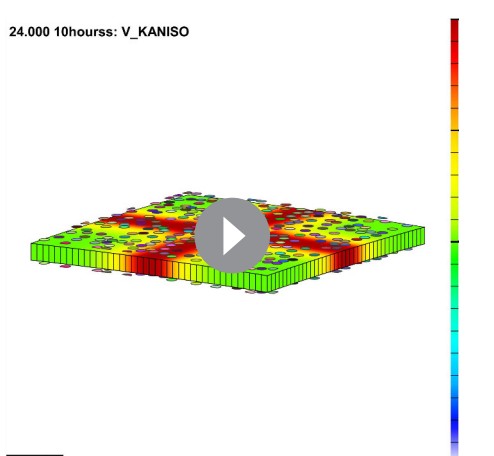

**Video 13.** Orthogonal directional conflict model as in *Figure 6U*. Size of canvas is rescaled to better visualise the deformation of the canvas.

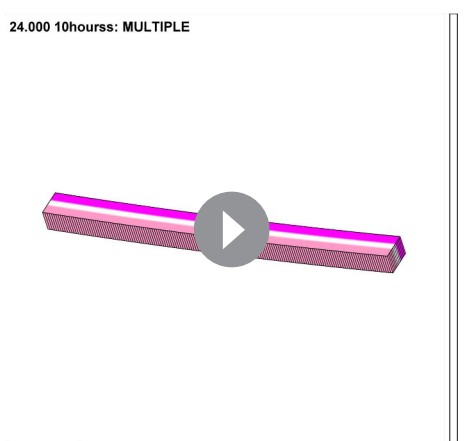

**Video 14.** *div* domes model as in *Figure 9F*. Size of canvas is rescaled to better visualise the deformation of the canvas.

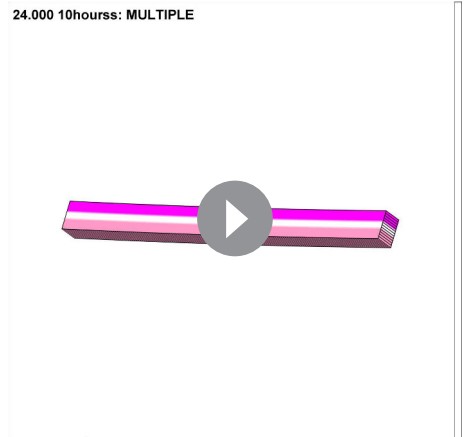

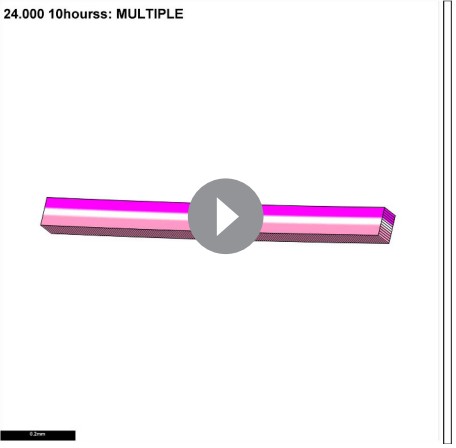

**Video 15.** *div* domes model without surface conflict as in *Figure 9G*. Size of canvas is rescaled to better visualise the deformation of the canvas.

**Video 16.** *div* domes model without areal conflict as in *Figure 9H*. Size of canvas is rescaled to better visualise the deformation of the canvas.

lower corolla (*Figure 9—figure supplement 1D*), suggesting that the deflection of polarity could play a role in refining corolla shape.

## Discussion

Through analysis of cell files, polarity patterns and computational modelling, we show that an orthogonal directional conflict plays a key role in generating out-of-plane deformations underlying the 3D Snapdragon corolla shapes of wild-type and dorsoventral mutants. This directional conflict interacts with areal and surface conflicts to generate the observed shape. We propose these conflicts are generated through combinatorial patterns of gene activity along different axes of the flower, which modulate local rates and orientations of growth, and hence the pattern of conflicts. Dorsoventral genes, such as *DIV*, affect all three types of conflicts to various degrees, which resolve to generate flowers with altered shapes in different genotypes.

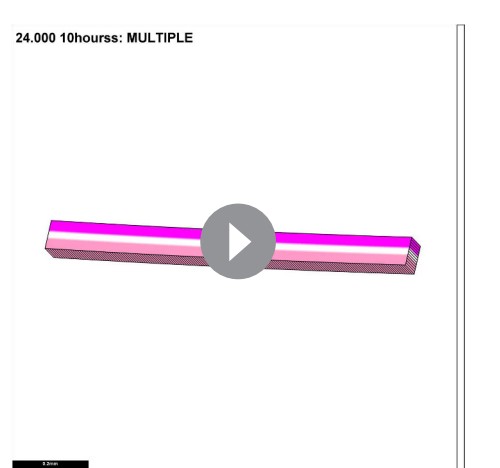

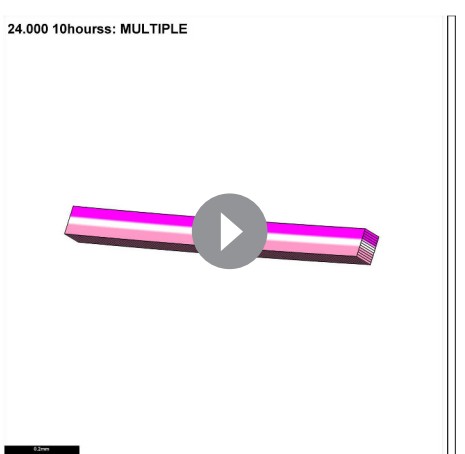

**Video 17.** *div* domes model without directional conflict as in *Figure 9I*. Size of canvas is rescaled to better visualise the deformation of the canvas.

**Video 18.** *div* domes model without orthogonal directional conflict as in *Figure 9J*. Size of canvas is rescaled to better visualise the deformation of the canvas.

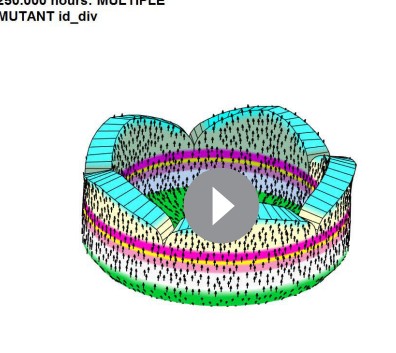

250.000 hours: MULTIPLE
MUTANT id_div

0.2mm

**Video 19.** *div* full Snapdragon model as in *Figure 9K*. Size of canvas is rescaled to better visualise the deformation of the canvas.

Computational modelling allows the role of each conflict to be revealed by simulations in which each conflict is removed, while preserving the others, producing virtual mutants with altered shapes. These phenotypes are distinct from virtual mutants obtained by removing dorsoventral gene activity from the model (*Figures 9G–J* and *10G–J* and *Figure 11A–D*). The difference may reflect evolutionary history through which genes modulate conflicts in multiple ways to modify flower shape. The end result is that each gene, such as *DIV*, has an effect on multiple conflicts.

Each conflict depends on combinatorial interaction of multiple genes, such as those specifying orthogonal domains needed for directional conflict. These expression domains are established along the mediolateral and proximodistal axes. Orthogonal gene expression domains are prevalent in development (*Wolpert et al., 2015*), although the genes establishing the orthogonal domains in the Snapdragon corolla remains to be identified.

In addition to orthogonal domains, directional conflict also requires a system for providing axial information within each region that allows specified growth to be oriented. Axial information cannot be specified directly by gene expression levels which provide only scalar information (*Lawrence et al., 2007*). Axiality could be provided by a polarity field (*Abley et al., 2013*; *Bosveld et al., 2012*; *Goodrich and Strutt, 2011*) or through mechanical cues such as stresses (*Aigouy et al., 2010*; *Heisler et al., 2010*); (*Hervieux et al., 2016*).

We hypothesise that the axial information is primarily provided by a polarity field, with growth rates specified parallel or perpendicular to it. PIN1 localisation reveals a tissue cell polarity pattern that it is largely proximodistal at early stages. At the time of repatterning, PIN1 is upregulated at the petal junctions, where polarity remains largely proximodistal, rather than being inverted distal to the foci as had been previously hypothesised (*Green et al., 2010*). Nevertheless, the growth orientations specified by these two polarity patterns would be similar, as specified anisotropic growth depends on axiality, which is the same whether the polarity is inverted or not. Thus, PIN1 localisation helps narrow down the possibilities for potential polarity fields guiding specified growth by providing data that cannot be inferred from growth analysis alone.

Although remaining proximodistal along the petal junctions, PIN1 polarity is deflected during repatterning towards the sinus. Growth orientations follow this line of deflection, consistent with polarity orienting growth (*Figure 8I–J*). The direction of polarity in the flanking regions which undergo mediolaterally oriented growth is unknown, as PIN1 signal is not observed in these regions. For simplicity we assume the polarity is maintained as proximodistal in these regions, and growth is enhanced perpendicular to the polarity. These assumptions can generate a Snapdragon corolla shape similar to that of a previous model (*Green et al., 2010*).

An alternative to polarity fields for orienting anisotropic specified growth is to employ the residual stresses generated by areal conflicts. This idea has been explored in the case of *Arabidopsis* sepal development (*Hervieux et al., 2016*), where it was proposed that stresses generated by differential isotropic specified growth could feed back to reinforce tissue regions in the direction of the local stress. Regions of tissue would then no longer grow isotropically if mechanically isolated (i.e. specified anistropic growth). However, simulations show that using local stresses to orient specified anisotropy does not allow a coordinated pattern of orientations to be specified (*Hervieux et al., 2016*). This is because growth feeds back to modify the stresses and destabilises their orientations. To get around this problem, it was proposed that average stress orientation across the sepal orient specified anisotropic growth (*Hervieux et al., 2016*). It is unclear, however, how such a sensing mechanism might operate and how it would allow an orthogonal pattern of orientations to be specified in the case of the Snapdragon flower.

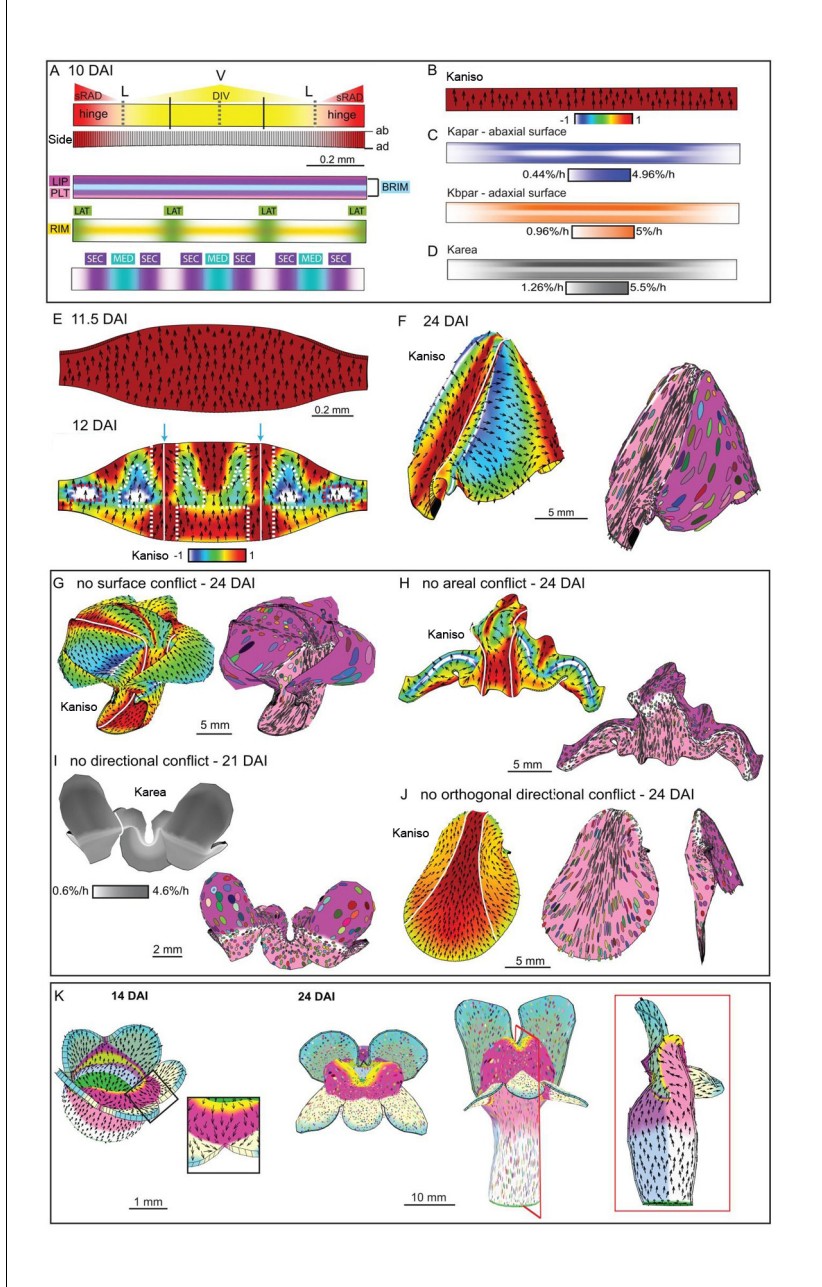

**Figure 10.** Tissue-level model of wild-type corolla development. (**A–J**) Modelling of the wild-type wedge. (**A**) The regional identities are similar to those in *div* (**Figure 9A**) with the addition of the graded DIV expression (yellow shading) and a region, SEC, between the midvein and petal junctions (bottom panel). (**B**) Growth is initially higher in $K_{par}$ than in $K_{per}$ (Kaniso). (**C**) DIV inhibits Kapar at the ventral-lateral junctions (whiter domain in Kapar), while the RIM region is kept short by DIV inhibiting $K_{par}$ slightly on both surfaces. (**D**) Growth is boosted in the region of the proximal lip and the palate (see darker grey regions in $K_{area}$). (**E**) The above pattern of growth results in a taller strip, compared to *div*, at 11.5 DAI (compare with **Figure 9E**). At 12 DAI (repatterning stage), DIV promotes $K_{per}$ in combination with SEC and LIP, extending the region of high mediolateral growth (Kaniso, modified orthogonal pattern indicated with white dashed lines). DIV also activates expression of a minus organiser at the sinus (blue arrows), deflecting the polarity field towards it. (**F**) The above specified pattern of growth leads to a shape and pattern of clone orientations similar to that of **Figure 2E** (left panel shows Kaniso while right panel shows the LIP and PLT regional factors and virtual clones). (**G–J**) Evaluating the contribution of growth conflicts to the shape of the wild-type wedge models. (**G**) Removal of surface conflict produced a similar shape as in F but the out-of-plane deformations protrude towards the abaxial side instead of the adaxial side. (**H**) Removing areal conflict generated a small wedge with protruding foci and a long hinge region. (**I**) Removing all directional conflict, resulted in a

*Figure 10 continued on next page*

*Figure 10 continued*

narrow strip with big bulges at the lateral petal which would normally form the flanks of the wedge. (J) Removing only the orthogonal conflict resulted in a shape with a long palate, sharp bend at the rim (left and middle panel) and a narrow lip (side view in right panel). (K) Incorporating the above growth patterns of areal, surface and directional conflict into the full Snapdragon model produced a closed mouth wild-type corolla, with an extended palate, a ridge at the rim and a steep lip. Canvas shown at 14 DAI and maturity (24 DAI). The region of deflection of polarity induced at the repatterning stage is shown enlarged at 14 DAI. Regions highlighted are lip (dark pink), palate (pink) and rim (yellow). Kaniso = $\ln(K_{par}/K_{per})$. The colour scale for Kaniso is −1 to +1. $K_{area} = K_{par} + K_{per}$.

Whatever the mechanism for orienting specified growth, it needs be connected with cellular properties. A growing plant tissue can be considered as a deforming mesh of cell walls which yields continuously to cellular turgor pressure (*Lockhart, 1965*; *Moulia and Fournier, 2009*). This continuous process of mesh deformation is coordinated with introduction of new walls through cell division, keeping cell sizes within certain bounds and allowing mesh strength to be maintained as it grows. Specified growth depends on how genes control the degree of wall extensibility and cell turgor. Turgor pressure acts isotropically, while cell wall extensibility can be anisotropic because of the orientation and cross-linking of wall fibres such as cellulose. If a sheet of cellular tissue has uniform turgor and its walls have isotropic mechanical properties and yield to the pressure, the tissue gets uniformly larger. This situation corresponds to uniform isotropic specified growth (*Figure 1B*). Conflicts arise through spatial variation in turgor and/or wall extensibility. For example, the areal conflict illustrated in *Figure 1H* could arise if walls in the central region of a square tissue yield more readily in the plane to turgor pressure. Similarly, the surface conflict in *Figure 1F* could reflect walls yielding more readily in the upper layer of cells of the tissue.

Specified anisotropic growth depends on orientation and/or cross-linking of cellulose fibres in the cell wall, giving it anisotropic yielding properties. Tissue cell polarity could influence growth orientations by biasing alignments of microtubules (*Hashimoto, 2015*), or other processes influencing cell wall anisotropy (*Cosgrove, 2016a*, *2016b*). Directional conflicts would then arise through variation in tissue cell polarity orientations and/or how cells modify wall extensibility in response to these orientations (e.g. wall stiffness parallel or perpendicular to the polarity, (*Coen and Rebocho, 2016*)).

We show that tissue conflict resolutions can generate similar deformations with or without overall growth, suggesting they provide a flexible morphogenetic mechanism applicable to many systems (*Figures 6* and *7*). For example, pollen wall folding has been proposed to arise through differential shrinkage following water loss, which would correspond to areal and directional conflicts in shrink (negative growth) rates (*Katifori et al., 2010*). Gastrulation has been proposed to involve differential

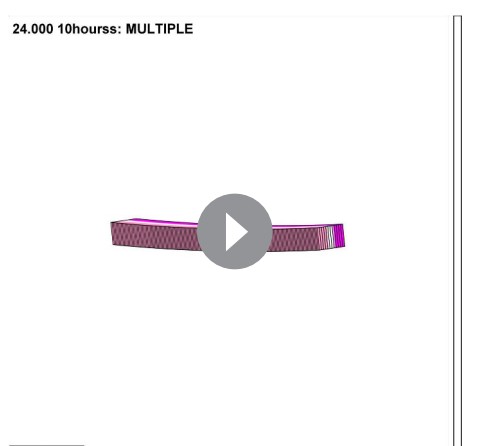

**Video 20.** Wild-type wedge model as in *Figure 10F*. Size of canvas is rescaled to better visualise the deformation of the canvas.

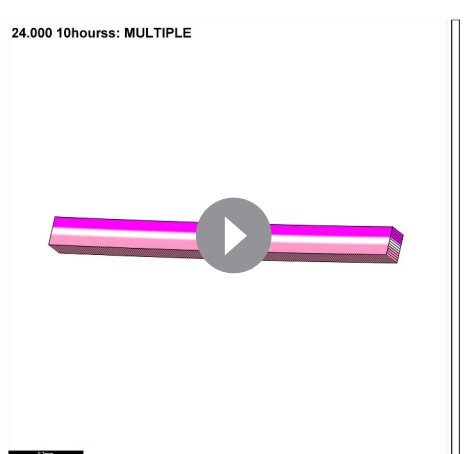

**Video 21.** Wild-type wedge model without surface conflict as in *Figure 10G*. Size of canvas is rescaled to better visualise the deformation of the canvas.

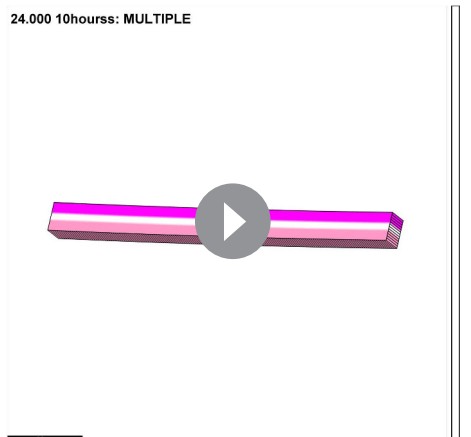

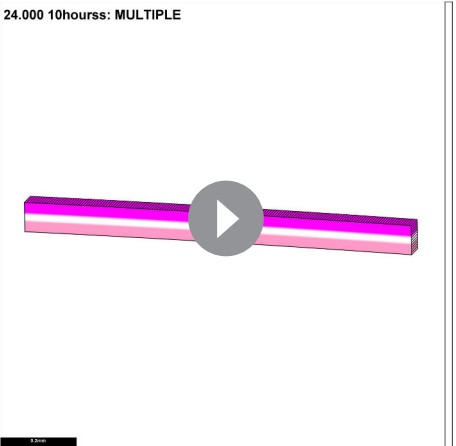

**Video 22.** Wild-type wedge model without areal conflict as in *Figure 10H*. Size of canvas is rescaled to better visualise the deformation of the canvas.

**Video 23.** Wild-type wedge model without directional conflict as in *Figure 10I*. Size of canvas is rescaled to better visualise the deformation of the canvas.

contraction of the apical and basal ends of cells, corresponding to a surface conflict (*Conte et al., 2008*). The formation of the Drosophila wing involves deformation and out-folding of an epithelial sheet in the imaginal disc. Cell divisions within the dorsoventral boundary region of the disc tend to be oriented orthogonally to divisions in the nearby wing disc (*Baena-López et al., 2005*; *Mao et al., 2011*), mirrored by orthogonal expression domains (e.g. Dpp, Wingless) (*Neto-Silva et al., 2009*). Formation of the wing may therefore involve a directional conflict. Animal epithelial deformations can also be generated in the absence of cell proliferation. The dorsal appendage in *Drosophila* is proposed to arise through differential tension oriented along orthogonal expression domains, corresponding to directional conflict (*Osterfield et al., 2013*). Thus, similar types of tissue conflict resolution may underlie deformations for both plant and animal tissues, even though the underlying cell behaviours can be very different.

According to the view presented here, tissue deformations arise through a dynamic interplay between gene activity and mechanical connectivity leading to local rotations through tissue conflict resolution. We show how genes can influence shape by changing the spatiotemporal pattern of tissue conflicts. This requires that genes modify local cell properties, such as wall extensibility in plants, actomyosin contractibility in animals, as well as tissue cell polarity (*Coen and Rebocho, 2016*). Outstanding questions are how such cellular properties are genetically controlled to produce the various types of tissue conflict, and how they are modulated during evolution to generate diverse shapes such as floral spurs and pitcher-shaped leaves.

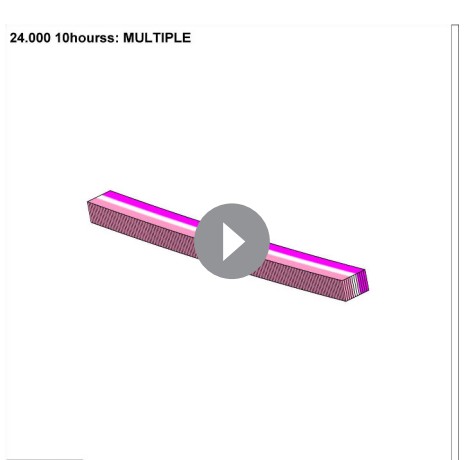

**Video 24.** Wild-type wedge model without orthogonal directional conflict as in *Figure 10J*. Size of canvas is rescaled to better visualise the deformation of the canvas.

## Material and methods

### Plant material

*Antirrhinum majus* (snapdragon) wild type (JI7 and JI2) and *div-13* (*Almeida et al., 1997*) were grown in the greenhouse at the John Innes Centre.

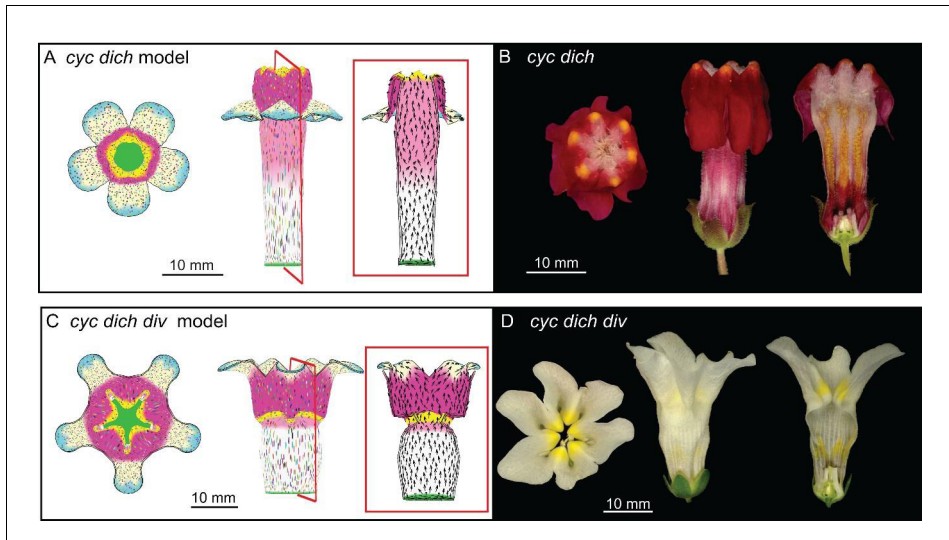

**Figure 11.** Tissue-level models of dorsoventral mutants. Comparison between corolla shapes of virtual mutants generated by removing gene activities from the tissue-level model (**A, C**) and corresponding phenotypes of real corollas (**B, D**). Double mutant *cyc dich* (**A,B**) and triple mutant *cyc dich div* (**C,D**) are shown. For each panel, corollas are viewed from top (left), side (middle) and in longitudinal mid-section (right, position of section for the virtual corollas indicated by red line in middle panel). For the virtual mutants, regions highlighted are lip (dark pink), palate (pink) and rim (yellow).

## Staging of Stock7 flower development

Using the published staging reference (*Vincent and Coen, 2004*) for JI2 flower development, we adapted it to JI7. Although these stocks are related, bud emergence and plastochron counts are slightly different in JI7. Therefore, we developed a JI7 staging reference by dissecting and photographing several flower bud series and relating bud features such as carpel, stamen and petal morphology between stocks as well as measurements such as petal width and lobe length. Imaging of buds and flattened petals was done using a stereomicroscope (Leica M205C). Bud age is referred to as Days After petal Initiation (DAI) of the flower meristem, and corresponds to the published reference system (*Vincent and Coen, 2004*). We used the JI7 reference system to stage the mutants. As

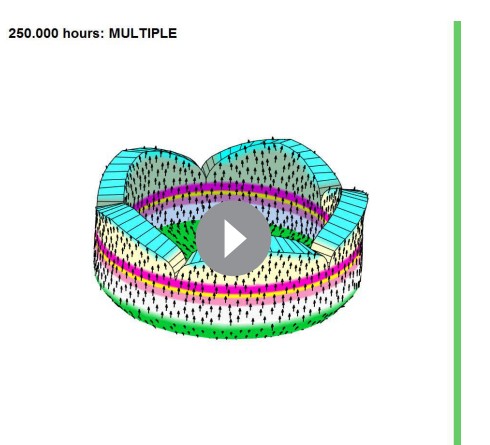

**Video 25.** Wild-type full Snapdragon model as in *Figure 10K*. Size of canvas is rescale to better visualise the deformation of the canvas.

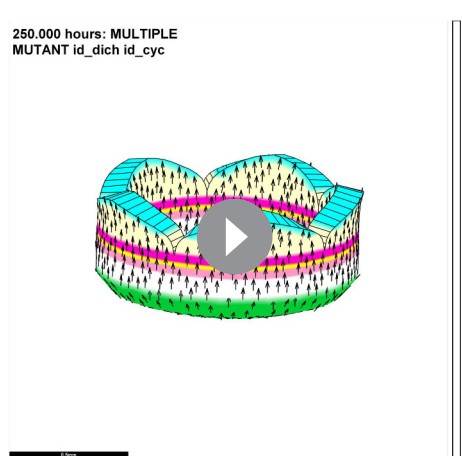

**Video 26.** *cyc dich* double mutant model as in *Figure 11A*. Size of canvas is rescale to better visualise the deformation of the canvas.

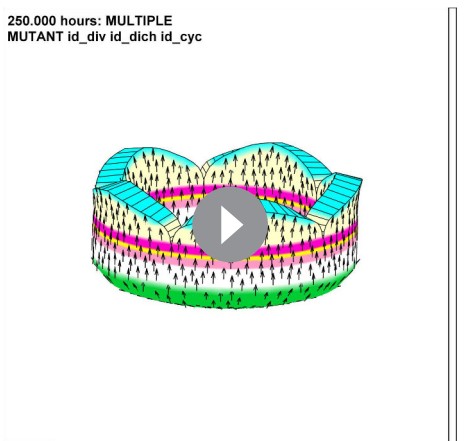

**250.000 hours: MULTIPLE MUTANT id_div id_dich id_cyc**

**Video 27.** *cyc dich div* triple mutant model as in *Figure 11C*. Size of canvas is rescale to better visualise the deformation of the canvas.

the *div*-13 phenotypes have modified lower petals, we used the length and width of the dorsal petals to stage the mutant flowers at different stages of development.

## Dissection and flattening of lower petals

Snapdragon buds were dissected from inflorescences, sepals removed and the corolla opened and flattened on a drop of prostatic adhesive (technovent) placed onto a glass slide. At early stages the whole petal could be flattened onto the glue. At later stages, when wedge formation was deforming the lower petals, we tried to keep the 3D shape of the petals by only flattening the distal lobe and palate-tube regions.

## Fate map of wedge emergence

To follow the development of the wedge region from emergence until maturity in *div-13* and wild-type, we used dissected and flattened material from staging experiments, as well as the OPT bud series, to mark different morphological landmarks that delimit the wedge, and made measurements on how this region grew. The measurements were done in imageJ. We used the same

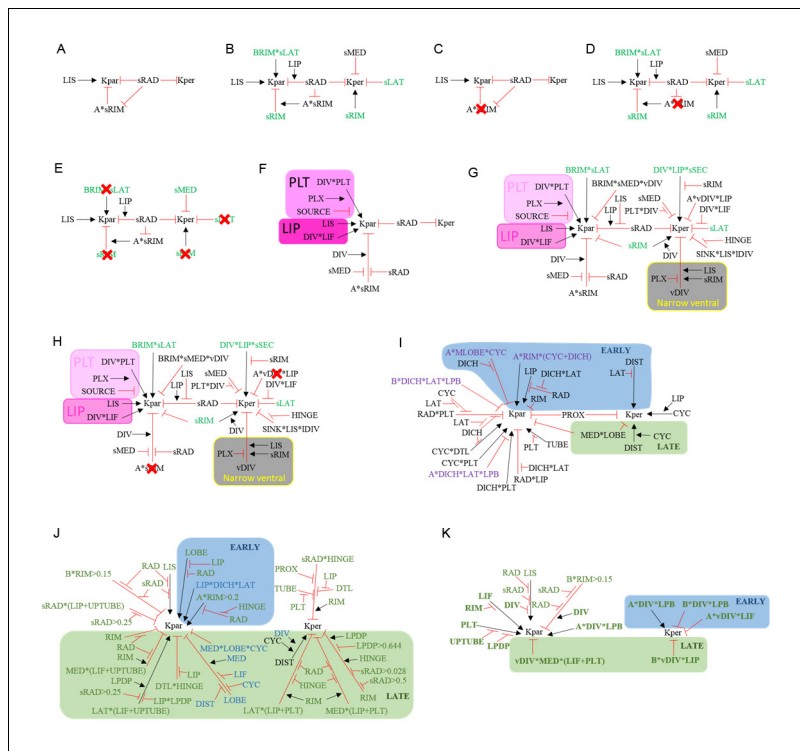

**Figure 12.** Summaries of growth regulatory networks (KRNs) used in models See materials and methods for full explanation. (**A**) Phase I of div domes. (**B**) Phase II of div domes. (**C**) Phase I of div domes with no surface conflict. (**D**) Phase II of div domes with no surface conflict. (**E**) Phase II of div domes with no orthogonal conflict. (**F**) Phase I of wild-type wedge. (**G**) Phase II of wild-type wedge. (**H**) Phase II of wild-type wedge with no surface conflict. (**I**) Ground state of full corolla model. (**J**) div mutant full corolla model showing new interactions (green) and modified interactions (blue). (**K**) Wild-type full corolla model showing only DIV-dependent interactions (green and bold).

landmarks for *div-13* and wild-type (exceptions indicated below). We used the lateral petal midvein as the mediolateral limit of the wedge, which was visible at early stages just after petal emergence at 8 DAI. To mark the upper and lower limits of the wedge along the proximodistal axis, we used the lip-distal lobe boundary and the palate-distal tube boundary, respectively. The petal-distal lobe boundary ran along the sinus of the petals, so to mark this boundary we drew lines from the D-L petal sinus to the V-L sinus as well as the two V-L sinuses. This morphological landmark was easily marked from the time of petal emergence. The proximal wedge limit could only be marked after 14 DAI, when palate trichomes emerged. This limit could not be marked in *div*-13 as the lower petals did not develop palate trichomes (due to absence of ventral identity). The rim position was visible from 14 DAI, as a furrow in flattened petals.

## Optical tomography projection (OPT)

Snapdragon flower buds were dissected from inflorescences, sepals removed and the buds prepared for OPT as previously described (*Lee et al., 2006*). Scanning and reconstructions were performed using a prototype OPT Scanner at the John Innes Centre. The 3-D images were visualised and processed using VolViewer software (http://cmpdartsvr3.cmp.uea.ac.uk/wiki/BanghamLab/index.php/Main_Page).

## *In situ* hybridisation

Snapdragon inflorescences were harvested and hand cut to 1 cm thick, before fixing in phosphate-buffered saline (PBS) pH 7.5 containing 4% paraformaldehyde (RT-15710, EMS), 0.1% Triton-X100 and 0.1% Tween-20, overnight at 4℃. After dehydration (50% and 70% ethanol), samples were embedded in paraffin (wax paramat-361148C, VWR) using a Tissue-TEK VIP processor (Sakura). 8 μm sections were cut, mounted on polysine slides (VWR) and dried overnight at 42℃. Slides were dewaxed in histoclear (2x for 10 min) (HS-200, National Diagnostics) and hydrated through a decreasing ethanol series (made in 1x saline solution: 0.85% (w/v) NaCl). Slides were washed in PBS solution before digesting the tissue with Pronase (0.125 mg/ml, P6911 Sigma-Aldrich) for 12 min. A 0.2% (v/v) Glycine-PBS solution was used to block the Pronase activity (3 min, RT), followed by a PBS wash and another fixation step with 4% paraformaldehyde in PBS for 10 min. After a 2 × 10 min PBS washes, the tissue charges were neutralised using a 0.5% (v/v) acetic anhydride solution made in 0.1M triethanolamine pH 8.0, to minimise unspecific binding and background. Slides were washed in PBS and dehydrated though an increasing ethanol series (made in 1x saline solution) and left to dry for at least 15 min. To generate specific digoxigenin-labelled riboprobes probes, the ORF of AmPIN1a (F, 5'ATGATAACTTTATCTGATTTTTACCTG-3' and R, 5'-TCATAGTCCCAGCAAAATATAGTAG-3'), AmPIN1b (F, 5'-AAAAATGATATCTTTATCTGATTTTTACC-3' and R, 5'TCAAGTCCCAA-CAAAATATAGTAGATG-3') and AmPIN1c (F, 5'ATGATCACTTTAACAGACTTATATCACG-3' and R, 5'-CTAGAGTCCAAGCAAGATGTAGTAG-3') were amplified using TAQ DNA polymerase PCR (201205, Qiagen) and cloned into pCR4-TOPO TA vector (K4575-01, Life Technologies) following manufacturer's instructions. The cloned fragments were amplified using forward specific primers (AmPIN1a, 5'-GAGAAAGTGAAAGTGATGCCTCC-3', AmPIN1b, 5'-CAATGAAGATGGTTACTTGGAG-3' and AmPIN1c, 5'CCAAACCAACTGCAATGCCACC −3') and the M13F primer (pCR4-TOPO kit), column purified (28104, Qiaquick PCR purification) followed by a phenol/chloroform extraction. Antisense probes were obtained by RNA transcription using the T7 promoter (10881767001, Roche) and DIG-UTP (11209256910, Roche), according to manufacture instructions. The AmPIN1a, AmPIN1b and AmPIN1c probes were hydrolysed at 60℃ using 200 mM carbonate buffer pH 10.2 solution, for 80, 8 and 60 min, respectively. The hydrolysed probes were boiled in 50% formamide (4311320, Life Technologies) solution for 2 min and left on ice to cool, before adding the hybridisation buffer (per 5 ml: 625 μl 10x Salts (3M NaCl, 0.1M Tris-HCl pH6.8, 0.1M NaPO4, 50 mM EDTA), 2500 μl Deionized formamide (4311320, Life Technology Ltd), 62.5 μl tRNA (R4251, Sigma-Aldrich), 125 μl Denhardt's(30915, Sigma-Aldrich), 437.5 μl RNase-free water, 1250 μl 50% Dextran Sulphate (D8906, Sigma-Aldrich)). Two microliters of hydrolysed probe plus 100 μl of hybridisation buffer were used per slide, and covered with Hybrislip hybridisation cover (HS6024-CS, Grace biolabs). Slides were hybridised in a humid box overnight at 50℃. Next day, slides were washed 3 times in 0.2x SSC solution at 55℃ for 25 min. Single stranded RNA was digested with RNaseA (0.02 mg/ml, R4875 Sigma-Aldrich) in NTE buffer (10 mM Tris-HCl pH 7.5, 1 mM EDTA and 500 mM NaCl)

at 37°C for 30 min. Slides were washed in 100 mM Tris.NaCl pH7.5 solution and then blocked using a freshly made 0.5% (w/v) blocking reagent (11096176001, Roche) in the same buffer, for 1 hr at room-temperature (RT). After a 30 min wash in 1% (w/v) BSA (A7906, Sigma-Aldrich) (made in 100 mM Tris.NaCl pH7.5 with 0.3% Tritonx100), slides were hybridised with a 1:3000 dilution of Anti-digoxigenin-AP antibody (11093274910, Roche) for 90 min. Slides were washed in 100 mM Tris.NaCl pH7.5 with 0.3% (v/v) tritonx100, 4 times for 25 min. Triton-x100 was removed from slides by washing them in 100 mM Tris.NaCl pH7.5 for 5 min. Before detection, the slides were washed in a 100 mM Tris.NaCl pH9.5 to equilibrate the pH. Detection was performed overnight using a NBT/BCIP (Promega) in 100 mM Tris.NaCl pH9.5 plus 50 mM MgCL2. The in situ experiments were performed three times.

## Immunocytochemistry

Protocol adapted from Conti and Bradley (*Conti and Bradley, 2007*). For section immunolocalisation, snapdragon inflorescences were harvested and hand cut to 2.5 mm thickness. Tissue was fixed in a 3.7% (v/v) formaldehyde (F8775, Sigma-Aldrich)-ethanol-acetic acid solution (FAA) with 1% (v/v) DMSO and 0.5% (v/v) Triton-x100, overnight at 4°C. After dehydration (50% and 70% ethanol), samples were embedded in paraffin (wax paramat-361148C, VWR) using a Tissue-TEK VIP processor (Sakura). 8 µm sections were cut, mounted on polysine slides (VWR) and dried overnight at 42°C. Slides were dewaxed in histoclear (2x for 10 min) (HS-200, National Diagnostics) and hydrated through a decreasing ethanol series. To improve antigen retrieval, slides were boiled in 10 mM citrate (pH 6) for 15 min and then left to cool for 30 min at room temperature. Slides were rinsed in deionized water and blocked in BB (5% (w/v) non-fat dry milk in PBS pH 7.5) at RT for 3 hr. Immuno-hybridisation of AmPIN1 was carried out overnight at 4°C using 1:500 primary anti-AmPIN1 antibody in BB. After washing three times (10 min) with PBS containing 0.3% (v/v) Triton-x100, the slides were hybridised with Alexa594-conjugated secondary antibody (Jackson ImmunoResearch, 1:500) for 3 hr. After incubation, slides were washed as after primary antibody incubation. Finally, slides were counterstained with 0.1% (w/v) calcofluor (F3543, Sigma-Aldrich) and mounted and imaged in 1% (w/v) DABCO (D27802, Sigma-Aldrich)/PBS/50% glycerol solution. For whole-mount immunolocalisation, buds were dissected and fixed flat onto a glass slide by glue as described in the dissection and flattening of lower petals section. The tissue was then fixed in a 3.7% (v/v) formaldehyde (F8775, Sigma-Aldrich)-ethanol-acetic acid solution (FAA) with 1% (v/v) DMSO and 0.5% Triton-x100, overnight at 4°C, after a short vacuum treatment. After two washes with 50% ethanol, the slides were washed with deionized water and boiled in 10 mM citrate (pH 6) for 15 min and then left to cool for 30 min at room temperature. Slides were rinsed in deionized water and blocked in BB (3% (w/v) BSA in PBS pH 7.5) at RT for 3 hr. Primary and secondary antibody hybridisation, washes and calcofluor staining were performed as described above. Tissue was mounted and imaged in 1% (w/v) DABCO/PBS solution. Number of replicas for calcofluor staining and whole mount PIN1 immuno of *div* mutant lower petals at various stages of development = 6. Number of replicas for calcofluor staining and whole mount PIN1 immuno of wild-type mutant lower petals at various stages of development = 11. Replicas gave similar results for similar developmental stages.

## Antibody production

The AmPIN1a, AmPIN1b, AmPIN1c and AmPIN1 (conserved peptide between all three PIN1 proteins) antibodies were produced in rabbit by Cambridge Research Discovery (CRB) using the following predicted antigenic peptide sequences present in the cytosolic loop domain. Peptide sequence for AmPIN1a -SRGPTPRPSNFEEE, AmPIN1b – HRGNNEDGYLERDEL, AmPIN1c – FSPASTKKKGE NGKD and AmPIN1 -IYSMQSSRNPTPRGS. The clearest signal was obtained with the AmPIN1a antibody. The AmPIN1 antibody gave similar results but with more background.

## Microscopy

Scanning electron microscopy was carried out as described in Vincent and Coen (*Vincent and Coen, 2004*). In situ hybridised material was imaged using a Leica DM6000 and Nomarski settings, x10 dry lens, and a DFC420 digital camera was used to photograph in situ sections. All measurements made from in situ images were calculated using ImageJ software (http://imagej.nih.gov/ij/). Fluorescent immunolocalisation results were imaged using a Leica SP5 II confocal microscope and x25 water

dipping lens. Images were acquired in a format of 1024 × 1024, with a line average of 4, scan speed of 600 Hz and pinhole of 1 airy unit. Excitation for Alexa 594 on laser 20% was with a 561 laser line, with a detection band of 570–630 nm. Calcofluor was imaged with a 405 diode laser and detection at 450–500 nm. Z stack images were at 1 µm/section. Image reconstruction was performed using the ImageJ software.

## Quantifying PIN1 polarity

PIN1 polarity was quantified using the PinPoint software, developed in MATLAB to analyse gene expression in cell boundaries. A composite stack of Anti-PIN1-Alexa594 and calcofluor staining was loaded into the program. For each cell segmentation, we determined which cell the PIN signal belonged to by manually scrolling through the z-stack. In ambiguous cases this determination was aided by PIN signal following the curvature of cell walls (*Figure 8—figure supplement 2*). A representative segmentation plane was chosen near to the cell's midplane and the cell boundary was manually segmented. The segmentation was used to create an image mask of the cell. A second image mask was created for each cell by morphologically eroding the cell boundary with a disk-shaped structuring element of radius five pixels ($\approx 1.5$ µm). A difference image was created by subtracting the eroded mask from the original cell mask. For each pixel in the difference image, a vector was calculated using the PIN1 pixel intensity as magnitude and angle measured from the cell centroid to the pixel, relative to the x-axis as orientation. The sum of these vectors was used to represent the polarity of each cell. Average vector fields were calculated by summing vectors for each cell within a 30 × 30 pixel window.

## Definition of tissue conflict resolution

Mathematically, growth of a region is described by a growth tensor which can have both a strain (symmetric) and rotational (skew-symmetric) component (*Hejnowicz and Romberger, 1984*). A specified growth tensor has a strain component but no rotational component, as we assume that each tissue region exerts no intrinsic rotational force. However, a resultant growth tensor may have both strain and rotational components. If all regions are specified to grow in a similar manner and no external forces act on the tissue, specified and resultant growth are the same. This is a conflict-free situation and no local rotations are generated (the resultant tensor has no rotational component). By contrast, if specified growth varies across a tissue it may create situations of potential conflict which are resolved or reduced through local rotation (i.e. the resultant growth tensor has a rotational component). Local rotations provide degrees of freedom not part of the specified growth which can allow potential stresses to be reduced. We refer to this mechanism for generating local tissue rotations as *tissue conflict resolution*.

In some cases, the potential conflicts in growth may be fully resolved through rotations, in which case the only difference between specified and resultant growth tensors is in the rotational component and there is no residual strain. For example, a conformal map is a planar deformation in which local rotations are generated through a gradient in isotropic specified growth without producing residual strain (*Alim et al., 2016*; *Mitchison, 2016*). If the tissue did not rotate locally to accommodate the gradient in specified growth, residual stresses would be generated. Thus, local rotations fully resolve the potential tissue conflict. In most cases, however, local rotations only partially resolve the conflicts and the specified and resultant growth tensors differ in the strain component (residual strain) as well as the rotational component. Some residual strain is inevitable in cases of tissue buckling (curvature through areal or directional conflicts), as the two surfaces of the tissue grow to a different extent, even though their specified strains are identical.

## Computational modelling

See Supplementary Material and methods for a full description of tissue-level modelling. All models used can be downloaded from:

http://cmpdartsvr1.cmp.uea.ac.uk/downloads/software/OpenSourceDownload_Elife_Rebocho_2016/GPT_TissueConflicts.zip

Models are based on the Growing Polarised Tissue framework (GPT-framework) (*Green et al., 2010*; *Kennaway et al., 2011*). Tissue, is considered as a continuous sheet, termed a canvas, with two surfaces (A and B). Identity and signalling factors can be specified throughout the canvas and

interact through a gene regulatory network (GRN). A growth regulatory network (KRN) controls the specified growth parallel ($K_{par}$) and perpendicular ($K_{per}$) to the local polarity, established by taking the gradient of a diffusible factor POLARISER (POL). The production and degradation of POL depends on a polarity regulatory network (PRN). To visualise specified anisotropy, we calculate Kaniso = ln ($K_{par}$ /$K_{per}$) and display its local value using a colour scale. If growth is isotropic ($K_{par}$ = $K_{per}$), Kaniso = 0. Specified areal growth rate is given by $K_{area}$ = $K_{par}$ + $K_{per}$.

## Cell file simulations

### Figure 4 and Figure 4—figure supplement 1. Cell file

To explore growth patterns that could give rise to the observed cell wall staining with calcofluor, cell division is modelled using the GPT framework. Cells are drawn on the canvas and their vertices displaced with the canvas as it grows. New walls and vertices introduced when the cell area reached a threshold value. Division walls take the shortest path through the cell centroid. The model has three stages: (T0), when the canvas is established and the initial cells are introduced; (T1), a first phase of growth, and (T2) a second phase of growth. To distinguish between the age of cells walls formed at each stage, we specify that walls at an initial time point (T0) are marked with thick blue lines, cell walls that arise during T1 are shown with thinner green lines and cell walls that arise during T2 are marked with very thinner white lines. A proximodistal polarity field is established at T0 using the GUI. We explore three growth patterns:

1. Uniform isotropic growth,

$$\begin{aligned} K_{par} &= 0.05 \\ K_{per} &= 0.05 \end{aligned}$$

2. High $K_{par}$,

$$\begin{aligned} K_{par} &= 0.08 \\ K_{per} &= 0 \end{aligned}$$

3. High $K_{par}$ during T1 switching to high $K_{per}$ at start of T2

$$\begin{aligned} \text{During T1} \quad K_{par} &= 0.1 \\ K_{per} &= 0 \\ \text{During T2} \quad K_{par} &= 0 \\ K_{per} &= 0.1 \end{aligned}$$

## Generation of domes through tissue conflicts

To illustrate how various types of tissue conflict resolution lead to curvature, we use an initial square with very slight curvature and marked with circular clones.

### Figure 1A, B. Uniform isotropic growth

To model uniform specified isotropic growth the KRN equations are:

$$\begin{aligned} K_{par} &= 0.03 \\ K_{per} &= 0.03 \\ K_{nor} &= 0.044 \end{aligned}$$

where Knor is specified growth rate in canvas thickness.

### Figure 1C, D. Uniform anisotropic growth

To model uniform specified anisotropic growth, a polarity field is established by producing POL at the bottom of the canvas (through factor BOTTOM) and fixing it to a low concentration at the top (through factor SINK). The gradient of POL defines the local polarity. For uniform anisotropic specified growth, the KRN equations are:

$$K_{par} = 0.03$$
$$K_{per} = 0.02$$
$$K_{nor} = 0.044$$

## Figure 1E, F. Surface conflict

To model surface conflict with growth, the KRN equations are:

$$K_{par}^{a} = K_{per}^{a} = 0.06$$
$$K_{par}^{b} = K_{per}^{b} = 0.05$$
$$K_{nor} = 0.044$$

## Figure 1G, H. Areal conflict

To model areal conflict with growth, the KRN equations are:

$$K_{par} = 0.05 + 0.05 \bullet i_{centre}$$
$$K_{per} = 0.05 + 0.05 \bullet i_{centre}$$
$$K_{nor} = 0.044$$

where $i_{centre}$ is an identity factor with a level of 1 at the centre of the canvas from where it gradually declines radially.

## Figure 1i, J. Convergent directional conflict

To model directional conflict, a convergent polarity is established by producing POL at the centre of the canvas (by factor FOCI) and degrading at the periphery (by factor EDGE). The KRN equations are:

$$K_{par} = 0.05$$
$$K_{per} = 0.02$$
$$K_{nor} = 0.044$$

## Figure 1—figure supplement 1A,B. Areal conflict flat start

As in *Figure 1G,H* but starting with a flat canvas with a small amount of random perturbation in the z-plane.

## Figure 1—figure supplement 1C,D. Convergent directional conflict flat start

As in *Figure 1I,J* but starting with a flat canvas with a small amount of random perturbation in the z-plane.

## Figure 6F, G. Orthogonal directional conflict

To model orthogonal directional conflict in the context of a convergent polarity field, the same polarity field as above was used, with KRN equations:

$$K_{par} = 0.05 + 0.05 \bullet S_{jun} + 0.05 \bullet S_{rim} - 0.2 \bullet S_{jun} \bullet S_{rim}$$
$$K_{per} = 0.05 - 0.05 \bullet S_{jun} - 0.05 \bullet S_{rim} + 0.2 \bullet S_{jun} \bullet S_{rim}$$
$$K_{nor} = 0.044$$

where the distribution of $s_{jun}$ and $s_{rim}$ are as in *Figure 6A and B* respectively. Note that $K_{area} = K_{par} + K_{per} = 0.1$, and is therefore uniform throughout the canvas (ensuring no areal conflict).

## Figure 6H, I. T-shaped directional conflict

To model a T-shaped directional conflict in the context of a convergent polarity field, the KRN equations are:

$$K_{par} = 0.05 + 0.05 \bullet S_{jun} + 0.05 \bullet S_{rim} \bullet i_{halfside} - 0.1 \bullet S_{jun} \bullet S_{rim}$$
$$K_{per} = 0.05 - 0.05 \bullet S_{jun} - 0.05 \bullet S_{rim} \bullet i_{halfside} + 0.1 \bullet S_{jun} \bullet S_{rim}$$
$$K_{nor} = 0.044$$

where $i_{halfside}$ is a factor expressed in the left half of the canvas (*Figure 6C*).

## Figure 6J, K. L-shaped directional conflict

To model a L-shaped directional conflict in the context of a convergent polarity field, the KRN equations are:

$$\begin{aligned}
K_{par} &= 0.05 + 0.05 \bullet S_{jun} \bullet i_{distalhalf} + 0.05 \bullet S_{rim} \bullet i_{halfside} - 0.02 \bullet S_{jun} \bullet S_{rim} \\
K_{per} &= 0.05 - 0.05 \bullet S_{jun} \bullet i_{distalhalf} - 0.05 \bullet S_{rim} \bullet i_{halfside} + 0.02 \bullet S_{jun} \bullet S_{rim} \\
K_{nor} &= 0.044
\end{aligned}$$

where $i_{distalhalf}$ has the distribution shown for DISTALHALF in *Figure 6D*.

## Figure 6L, M. directional conflict for vertical domain

To model a directional conflict for a vertical domain in the context of a convergent polarity field, the KRN equations are:

$$\begin{aligned}
K_{par} &= 0.05 + 0.05 \bullet S_{jun} - 0.1 \bullet S_{jun} \bullet S_{rim} \\
K_{per} &= 0.05 - 0.05 \bullet S_{jun} + 0.1 \bullet S_{jun} \bullet S_{rim} \\
K_{nor} &= 0.044
\end{aligned}$$

## Figure 6N, O. Directional conflict for upper half of vertical domain

To model a directional conflict for a vertical domain in the context of a convergent polarity field, the KRN equations are:

$$\begin{aligned}
K_{par} &= 0.05 + 0.05 \bullet S_{jun} \bullet i_{distalhalf} \\
K_{per} &= 0.05 - 0.05 \bullet S_{jun} \bullet i_{distalhalf} \\
K_{nor} &= 0.044
\end{aligned}$$

## Figure 6P, Q. orthogonal directional conflict in a parallel polarity field

To model a directional conflict in the context of a parallel polarity field, the polarity field was established as for *Figure 1C*. The KRN equations are:

$$\begin{aligned}
K_{par} &= 0.05 + 0.05 \bullet S_{jun} - 0.05 \bullet S_{rim} \\
K_{per} &= 0.05 - 0.05 \bullet S_{jun} + 0.05 \bullet S_{rim} \\
K_{nor} &= 0.044
\end{aligned}$$

## Figure 6R, S orthogonal areal conflict

To model areal conflict for an orthogonal domain, the KRN equations are:

$$\begin{aligned}
K_{par} &= 0.05 + 0.05 \bullet S_{jun} + 0.05 \bullet S_{rim} - 0.02 \bullet S_{jun} \bullet S_{rim} \\
K_{per} &= 0.05 + 0.05 \bullet S_{jun} + 0.05 \bullet S_{rim} - 0.02 \bullet S_{jun} \bullet S_{rim} \\
K_{nor} &= 0.044
\end{aligned}$$

## Figure 6T, U. orthogonal directional conflict in an orthogonal polarity field

To model directional conflict in the context of an orthogonal polarity field, POL is initially produced at the bottom of the canvas (SOURCE) and degraded at the midpoint along the top of the canvas (SINK). Polarity along the vertical midline region is locked to create a polarity channel. while also acting as a weak sink for the SIDE regions where POL is now produced. The KRN equations are:

$$\begin{aligned}
K_{par} &= 0.05 + 0.05 \bullet S_{jun} + 0.05 \bullet S_{rim} \bullet i_{margins} - 0.075 \bullet S_{jun} \bullet S_{rim} \\
K_{per} &= 0.05 - 0.05 \bullet S_{rim} \bullet i_{margins} - 0.05 \bullet S_{jun} + 0.075 \bullet S_{jun} \bullet S_{rim} \\
K_{nor} &= 0.044
\end{aligned}$$

where $i_{margins}$ defines regions flanking the channel.

## Figure 6—figure supplement 1A,B. Combining directional and areal conflicts.

To model a combination of directional conflict with enhanced areal growth rates (areal conflict) in the orthogonal domains we used the polarity field as in *Figure 1C* and KRN equations:

$$
\begin{aligned}
K_{par} &= 0.01 + 0.06 \bullet S_{jun} - 0.12 \bullet S_{jun} \bullet S_{rim} \\
K_{per} &= 0.01 + 0.06 \bullet S_{rim} - 0.12 \bullet S_{jun} \bullet S_{rim} \\
K_{nor} &= 0.01
\end{aligned}
$$

### Figure 6—figure supplement 1C. Combining surface and directional conflicts

To model a combination of surface conflict and orthogonal directional conflict in the context of a convergent polarity field (generated as in *Figure 1I*), the KRN equations are:

$$
\begin{aligned}
K_{par}^{a} &= 0.055 + 0.05 \bullet S_{jun} + 0.05 \bullet S_{rim} - 0.2 \bullet S_{jun} \bullet S_{rim} \\
K_{per}^{a} &= 0.055 - 0.05 \bullet S_{jun} - 0.05 \bullet S_{rim} + 0.2 \bullet S_{jun} \bullet S_{rim} \\
K_{par}^{b} &= 0.05 + 0.05 \bullet S_{jun} + 0.05 \bullet S_{rim} - 0.2 \bullet S_{jun} \bullet S_{rim} \\
K_{par}^{b} &= 0.05 - 0.05 \bullet S_{jun} - 0.05 \bullet S_{rim} + 0.2 \bullet S_{jun} \bullet S_{rim} \\
K_{nor} &= 0.044
\end{aligned}
$$

### Figure 6—figure supplement 1D. Combining surface and areal conflicts

To model a combination of surface and areal conflict, we removed the slight curvature from our starting square so that it is flat with some random noise in the z-plane and with a specified a uniform surface differential of 10% higher growth on the A surface. The KRN equations are:

$$
\begin{aligned}
K_{par}^{a} &= K_{per}^{a} = 0.03 + 0.025 \bullet i_{centre} \\
K_{par}^{b} &= K_{per}^{b} = 0.025 + 0.025 \bullet i_{centre} \\
K_{nor} &= 0.044
\end{aligned}
$$

### Figure 7A, B. Surface conflict without overall growth

To model surface conflict while preserving overall canvas size, the KRN equations are:

$$
\begin{aligned}
K_{par}^{a} &= K_{per}^{a} = 0.005 \\
K_{par}^{b} &= K_{per}^{b} = -0.005 \\
K_{nor} &= 0
\end{aligned}
$$

### Figure 7C, D. Areal conflict without overall growth

To model areal conflict while preserving overall canvas size, the KRN equations are:

$$
\begin{aligned}
K_{par} &= 0.019 - 0.032 \bullet i_{edge} \\
K_{per} &= 0.019 - 0.032 \bullet i_{edge} \\
K_{nor} &= 0
\end{aligned}
$$

where $i_{edge}$ has a complementary distribution to $i_{centre}$. Parameters were chosen such that the final canvas size remained roughly constant.

### Figure 7E, F. Convergent directional conflict without overall growth

To model directional conflict with growth and contraction. The same polarity field as in *Figure 1I* was used, with KRN equations:

$$
\begin{aligned}
K_{par} &= 0.01 \\
K_{per} &= -0.01 \\
K_{nor} &= 0
\end{aligned}
$$

### Figure 7G, H. Orthogonal directional conflict in a parallel polarity field without overall growth

To model orthogonal directional conflict in the context of a parallel polarity field without overall growth, the KRN equations are:

$$
\begin{aligned}
K_{par} &= 0.05 \bullet S_{jun} - 0.05 \bullet S_{rim} \\
K_{per} &= 0.05 \bullet S_{rim} - 0.05 \bullet S_{jun} \\
K_{nor} &= 0
\end{aligned}
$$

Note that $K_{area} = K_{par} + K_{per} = 0$.

## Modelling the *div* domes and wild-type wedge

### Figures 9A and 10A. Set up

The initial rectangular canvas is 1060 µm by 120 µm, based on average width and depth measurements of the incipient wedge at 10 DAI. The KRN has two phases: 10–12 DAI (phase I) and 12–24 DAI (phase II). POL production is promoted by SOURCE, expressed at the most proximal region of the rectangle, and degraded at SINK, expressed at the most distal region of the canvas, generating a proximodistal polarity field. By 10 DAI, the polariser gradient, the expression pattern of the identity factors and diffusible factors (prefixed with 's') are established, and then fixed to the canvas for the rest of the simulation. Regional factors are established along three axes:

Dorsoventral axis: the dorsally expressed gene RAD is expressed at the most lateral petal edges and produces a signal (sRAD) that diffuses into the lateral petals (hinge region) and inhibits the ventrally expressed gene DIV.

Proximodistal axis: PALATE (PLT) is expressed in the proximal region. PALATEPROX (PLX) is expressed in the most proximal part of the palate. RIM is expressed at the distal limit of PLT. RIM promotes production of sRIM which diffuses and activates BRIM when the levels of sRIM are higher than 0.01. LIP is distal to RIM and is subdivided in the proximal region, LIPCLIFF (LIF), and a distal region, LIPDISTAL (LIS).

Mediolateral axis: graded factors are sMED, which is centred on the petal midvein, sSEC, which is centred on the secondary veins, and sLAT which is centred on the junction between petals.

Sectors (circular regions) were induced at around 12.5 DAI, with initial sector size scaled to the initial size of a cell (~10 µm). Growth in canvas thickness was maintained at 0.44 %/h in all models.

### Figure 9 A-E. Phase I of *div* domes

Growth is higher in $K_{par}$ than in $K_{per}$. Both $K_{par}$ and $K_{per}$ are inhibited at the hinge by the action of sRAD, to keep the hinge region short and narrow. $K_{par}$ is promoted at the distal lip (LIS) region to generate the asymmetric lip. The KRN phase I equations are:

$$\begin{aligned}
K_{par}^{b} &= 0.02 \bullet \text{inh}(5, \text{S}_{rad}) \bullet \text{pro}(0.8, i_{lis}) \\
K_{par}^{a} &= K_{par}^{b} \bullet \text{inh}(0.5, (\text{S}_{rim} > 0.4). \bullet \text{inh}(100, \text{S}_{rad})) \\
K_{per}^{b} &= 0.005 \bullet \text{inh}(3, \text{S}_{rad}) \\
K_{per}^{a} &= K_{per}^{b}
\end{aligned}$$

where pro $(x, y_z)$ promotes factor $y_z$ by an amount x, and inh $(x, y_z)$ inhibits factor $y_z$ by an amount x (*Green et al., 2010*). To promote the bending of the canvas at rim region, $K_{par}$ on the A surface was inhibited by sRIM where it exceeded at threshold value, except for near the hinge, where sRAD is high.

The KRN for Phase I is summarised in *Figure 12A* (activation as black arrows and inhibition as red lines).

### Figure 9F. Phase II of *div* domes

In addition to the interactions of phase I, the inhibition of $K_{par}$ with sRAD is boosted in by LIP, MED inhibits $K_{per}$ to keep the midvein regions narrow. Directional conflict was introduced by promoting $K_{par}$ with the combination of sLAT and BRIM region, inhibiting $K_{par}$ by sRIM, promoting $K_{per}$ by sRIM, and inhibiting $K_{per}$ with sLAT. The overall KRN for phase II is:

$$\begin{aligned}
K_{par}^{b} &= 0.012 \bullet \text{inh}(5, \text{s}_{rad} \bullet \text{pro}(3, i_{lip})) \bullet \text{inh}(0.8, i_{lis}) \bullet \text{inh}(2, \text{s}_{rim}) \bullet \text{pro}(0.8, \text{s}_{lat} \bullet i_{brim}) \\
K_{par}^{a} &= K_{par}^{b} \bullet \text{inh}(0.5, (\text{s}_{rim} > 0.4) \bullet \text{inh}(100, \text{s}_{rad})) \\
K_{per}^{b} &= 0.012 \bullet \text{inh}(3, \text{s}_{rad}) \bullet \text{pro}(2.5, \text{s}_{rim}) \bullet \text{inh}(3, \text{s}_{lat}) \bullet \text{inh}(3, \text{s}_{med}) \\
K_{per}^{a} &= K_{per}^{b}
\end{aligned}$$

These interactions are summarised in *Figure 12B*, with orthogonal directional conflict interactions in green.

### Figure 9G. No surface conflict

The above KRN is modified, with removed interactions marked with a red cross as shown in *Figure 12C* (Phase I) and *Figure 12D* (Phase II).

## Figure 9H. No areal conflict

KRN as in Phase I and II of *div* domes except specified areal growth rates were normalised across the canvas through the following equations:

$$
\begin{aligned}
K_{areab} &= K_{per}^{b} + K_{par}^{b} \\
K_{areaa} &= K_{per}^{a} + K_{par}^{a} \\
K_{par}^{b} &= 0.025 \bullet K_{par}^{b}/K_{areab} \\
K_{par}^{a} &= 0.025 \bullet K_{par}^{a}/K_{areaa} \\
K_{per}^{b} &= 0.025 \bullet K_{per}^{b}/K_{areab} \\
K_{par}^{a} &= 0.025 \bullet K_{per}^{a}/K_{areaa}
\end{aligned}
$$

The multiplying factor 0.025 was chosen to prevent the canvas becoming too big.

## Figure 9i. No directional conflict and specified anisotropy

POL is not produced and therefore no polarity field is established.

## Figure 9j. No orthogonal conflict

The modified KRN with removed interactions marked with a red cross is shown in *Figure 12E*.

## Figure 10A-E. Phase I of wild-type wedge

The KRN is similar to that for *div* phase I in *Figure 9A–E*, with the following modifications. DIV promotes $K_{par}$ in combination with PLT, with a slight boost by PLX, and inhibition at the proximal edge of the canvas by SOURCE. DIV promotes $K_{par}$ in combination with LIF. Furrow formation is enhanced through sRIM, except for the hinge (sRAD) and midvein regions (sMED). The full KRN is:

$$
\begin{aligned}
K_{par}^{b} &= 0.02 \bullet \text{inh}\,(5, s_{rad}) \bullet \text{pro}\,(0.8, i_{lis}) \bullet \text{pro}\,(1.5, i_{div} \bullet i_{lif}) \bullet \text{pro}\,(0.8, i_{div} \bullet i_{plt} \bullet \text{pro}\,(0.3, i_{plx}) \bullet \text{inh}\,(5, i_{source})) \\
K_{par}^{a} &= K_{par}^{b} \bullet \text{inh}\,(3, s_{rim}) \bullet \text{inh}\,(10, s_{med}) \bullet \text{inh}\,(100, s_{rad})) \\
K_{per}^{b} &= 0.005 \bullet \text{inh}\,(3, s_{rad}) \\
K_{per}^{a} &= K_{per}^{b}
\end{aligned}
$$

*Figure 12 F* shows a summary of KRN for Phase I, with the network promoting PLT and LIP growth highlighted.

## Figure 10F. Phase II of wild-type wedge

At the start of phase II polarity is deflected by establishing a new site of POL degradation at the SINUS (intersection between the petal junctions and distal lip). The deflection of polarity was made specific to the adaxial (B) surface of the canvas by freezing polarity on the A surface, preventing it from reorienting.

The boost of $K_{par}$ by LIS and the inhibition of $K_{par}$ and $K_{per}$ by sRAD are the same as in *div* model for phase II. Relative to phase I, the promotion of $K_{par}$ by LIF and DIV is reduced. The orthogonal directional conflict is similar to that in *div* phase II, except that $K_{per}$ promotion by sRIM is boosted by DIV and extended to the LIP regions. The extended region was kept narrow by SEC. Inhibition of $K_{per}$ by sLAT is reduced in the LIF region to allow for more $K_{per}$ and the formation of the protruding lip in the ventral petal. The inhibition of $K_{per}$ by sMED is reduced by PLT in combination with DIV to prevent the palate from becoming too narrow. The full KRN for phase II is:

$$K_{par}^b = \quad 0.012 \bullet \mathrm{inh}(5, S_{rad} \bullet \mathrm{pro}(3, i_{lip}))$$
$$\bullet \mathrm{pro}(0.8, i_{lis})$$
$$\bullet \mathrm{pro}(0.5, i_{div} \bullet i_{lif})$$
$$\bullet \mathrm{pro}(0.8, i_{div} \bullet i_{plt} \bullet \mathrm{pro}(0.3, i_{plx}) \bullet \mathrm{inh}(5, i_{source}))$$
$$\bullet \mathrm{inh}(2, S_{rim})$$
$$\bullet \mathrm{inh}(0.2, i_{div>0.95} \bullet S_{med} \bullet i_{brim})$$
$$\bullet \mathrm{inh}(50, S_{rad} \bullet i_{lis} \bullet \mathrm{pro}(100, i_{hinge}))$$
$$\bullet \mathrm{pro}(0.8, S_{lat} \bullet i_{brim})$$

$$K_{par}^a \quad = K_{par}^b \bullet \mathrm{inh}(3, i_{div} \bullet S_{rim} \bullet \mathrm{inh}\,(10, S_{med}) \bullet \mathrm{inh}\,(100, S_{rad}))$$

$$K_{per}^b \quad = 0.012 \bullet \mathrm{inh}(3, S_{rad})$$
$$\bullet \mathrm{pro}(2.5, S_{rim} \bullet \mathrm{pro}(0.5, i_{div}))$$
$$\bullet \mathrm{inh}(3, s_{lat} \bullet \mathrm{inh}(1, i_{div>0.83}) \bullet i_{lif})$$
$$\bullet \mathrm{inh}(0.5, i_{div>0.95} \bullet \mathrm{pro}(1, 4 \bullet S_{rim} + 1.5 \bullet i_{lis}) \bullet \mathrm{inh}(100, i_{plx}))$$
$$\bullet \mathrm{inh}(3, S_{med} \bullet \mathrm{inh}(10, i_{plt} \bullet i_{div}))$$
$$\bullet \mathrm{inh}(2, (1 - i_{div}) \bullet i_{lis} \bullet \mathrm{pro}(2, i_{sink}) \bullet \mathrm{inh}(100, i_{hinge}))$$
$$\bullet \mathrm{pro}(4.8, i_{div} \bullet i_{lip} \bullet S_{sec} \mathrm{inh}(8, S_{rim}))$$

$$K_{per}^a \quad = K_{per}^b \bullet \mathrm{inh}(0.5, i_{div>0.95} \bullet i_{lip})$$

A summary of the KRN is given in *Figure 12G* with orthogonal conflict interactions in green and the interactions responsible for keeping the ventral petal narrow in a yellow box, (vDIV represents the ventral DIV expression domain, where DIV level exceeds 0.95).

## Figure 10G. No surface conflict
Differential growth between canvas surfaces in $K_{par}$ at sRIM and $K_{per}$ at ventral lip removed (*Figure 12H*).

## Figure 10H. No areal conflict
Areal growth rates normalised as described for *div* in *Figure 9H*.

## Figure 10I. No directional conflict
POL is not produced.

## Figure 10J. No orthogonal directional conflict
Interactions in shown in green in the KRN (*Figure 12G*) removed.

### Full corolla models
Figure 9K Figure 9—figure supplement 1 Figure 10K, Figure 11

The models have three stages: (1) Step-up phase (9 DAI), when the polariser gradient, the expression pattern of most identity and signalling factors are established (patterns are maintained throughout simulation unless otherwise stated) (2) 10–13 DAI, during which EARLY factor is expressed and (3) 14–24 DAI, during which LATE factor is expressed. The PRN is as described in the set-up phase for the previously published model (*Green et al., 2010*), and the polarity field maintained throughout the entire corolla development, except in the wild-type model (see below). The POL gradient is frozen on both surfaces in the tube region during EARLY and in the lip region during LATE. Virtual sectors are induced around the repatterning stage (14 DAI). Growth in canvas thickness is 0.3 %/h. To facilitate comparisons, equations have different colours depending whether they are common between the published and the current model (black), present in published model but modified in the current model (blue), present in the published model but removed from the current model (red) or new to the current model (purple). Equation numbering is as in *Green et al. (2010)*: $K_{par}$ equations are K1, $K_{per}$ equations are K2, surface differential equations $K_{par}^a$ are K3, $K_{par}^b$ are K4, $K_{per}^a$ are K5 and $K_{per}^b$ are K6.

## Figure 9—figure supplement 1A. Ground state

All genetic interactions in the published model for wild type under the control of DIV or LTS (LAT-ERAL PETALS) which modulate specified growth rates in lower petals (see red equations below) are removed to produce a corolla without out-of-plane deformations in the lower petals.

KRN equations for the ground state are:

$K_{par} =$ (K.1)

$0.013$ (K.1.1)

$\bullet \mathrm{inh}(0.2, i_{prox})$ (K.1.2)

$\bullet \mathrm{pro}(0.4, \mathrm{i}_{tube} \bullet \mathrm{inh}(100, \mathrm{i}_{plt}))$ (K.1.3)

$\bullet \mathrm{pro}(1.4, \mathrm{i}_{early} \bullet \mathrm{i}_{lip} \bullet \mathrm{inh}(100, \mathrm{i}_{rad} \bullet \mathrm{inh}(100, \mathrm{i}_{dich} \bullet \mathrm{i}_{lat}))$ (K.1.4a)

$\bullet \mathrm{inh}(10, \mathrm{i}_{early} \bullet \mathrm{i}_{rim})$ (K.1.5)

$\bullet \mathrm{inh}(0.5, \mathrm{i}_{late} \bullet \mathrm{i}_{med} \bullet \mathrm{i}_{lobe})$ (K.1.6)

$\bullet \mathrm{inh}(1, \mathrm{i}_{rad} \bullet \mathrm{i}_{lip} \bullet \mathrm{inh}(6, \mathrm{i}_{dich} \bullet \mathrm{i}_{lat}))$ (K.1.8)

$\bullet \mathrm{inh}(1, \mathrm{i}_{rad} \bullet \mathrm{i}_{plt} \bullet \mathrm{inh}(15, \mathrm{i}_{dich} \bullet inh(5, \mathrm{i}_{lat})) \bullet \mathrm{inh}(30, \mathrm{i}_{cyc} \bullet \mathrm{inh}(40, \mathrm{i}_{lat}^2)))$ (K.1.9)

$\bullet \mathrm{pro}(0.3, \mathrm{i}_{cyc} \bullet \mathrm{i}_{dtl} \bullet \mathrm{inh}(0.5, \mathrm{i}_{dich}))$ (k.1.10)

$\bullet \mathrm{pro}(0.45, (\mathrm{i}_{cyc} + 0.2 \bullet \mathrm{i}_{dich}) \bullet \mathrm{i}_{plt})$ (K.1.11)

$\textcolor{red}{\bullet \mathrm{pro}(2.4, \mathrm{i}_{early} \bullet \mathrm{i}_{div} \bullet \mathrm{i}_{plt})}$ (K.1.7)

$\textcolor{red}{\bullet \mathrm{pro}(2.2, \mathrm{i}_{late} \bullet \mathrm{i}_{lts} \bullet \mathrm{i}_{med} \mathrm{inh}(0.5, \mathrm{i}_{lpb}) \bullet \mathrm{inh}(4, \mathrm{i}_{lat}) \bullet (\mathrm{i}_{lip} + 0.3 \bullet \mathrm{i}_{plt}))}$ (K.1.12)

$K_{per} =$ (K.2)

$0.0075$ (K.2.1)

$\bullet \mathrm{inh}(0.2, \mathrm{i}_{prox})$ (K.2.2)

$\bullet \mathrm{pro}(2, \mathrm{i}_{early} \bullet \mathrm{i}_{dist} \bullet \mathrm{inh}(20, \mathrm{i}_{lat}))$ (K.2.3)

$\bullet \mathrm{inh}(0.3, \mathrm{i}_{late} \bullet \mathrm{i}_{lobe} \bullet \mathrm{i}_{med})$ (K.2.5)

$\textcolor{red}{\bullet \mathrm{pro}(1, \mathrm{i}_{late} \bullet \mathrm{i}_{dist} \bullet \mathrm{pro}(1.2, (\mathrm{i}_{cyc} + \mathrm{i}_{div})))}$ (K.2.4b)

$\bullet \mathrm{pro}(0.1, \mathrm{i}_{cyc} \bullet \mathrm{pro}(1.5, \mathrm{i}_{lip}))$ (K.2.7)

$\textcolor{red}{\bullet \mathrm{inh}(1.3, \mathrm{i}_{late} \bullet \mathrm{i}_{div} \bullet \mathrm{inh}(2.5, \mathrm{i}_{lobe.} \bullet \mathrm{inh}(2, \mathrm{i}_{lpb})) \bullet \mathrm{pro}(1, 1\mathrm{i}_{plt}))}$ (K.2.6a)

$\textcolor{red}{\bullet \mathrm{inh}(6, \mathrm{i}_{early} \mathrm{i}_{lts} \bullet \mathrm{i}_{med})}$ (K.2.8)

$\textcolor{red}{\bullet \mathrm{pro}(0.2, \mathrm{i}_{late} \bullet \mathrm{i}_{lts} \bullet \mathrm{i}_{med} \bullet \mathrm{i}_{plt})}$ (K.2.9)

$K_{par}^a = K_{par}$ (K.3)

$\bullet \mathrm{inh}(1, \mathrm{i}_{dich} \bullet \mathrm{i}_{lat} \bullet \mathrm{i}_{lpb})$ (K.3.3)

$\textcolor{red}{\bullet \mathrm{inh}(1, \mathrm{i}_{early} \bullet (\mathrm{i}_{cyc} + \mathrm{i}_{div}) \bullet \mathrm{i}_{mlobe} \bullet \mathrm{inh}(5, \mathrm{i}_{dich}))}$ (K.3.4)

$\textcolor{red}{\bullet \mathrm{inh}(0.2, \mathrm{i}_{early} \bullet \mathrm{i}_{div} \bullet \mathrm{i}_{lpb} \bullet \mathrm{inh}(5, \mathrm{i}_{med}))}$ (K.3.5)

$\textcolor{red}{+ 0.05 \bullet \mathrm{i}_{early} \bullet \mathrm{i}_{rim} \bullet (1.2 \bullet \mathrm{i}_{div} + 0.3 \bullet \mathrm{i}_{lts} \bullet \mathrm{i}_{med} + 0.5 \bullet (\mathrm{i}_{cyc} + \mathrm{i}_{dich}))}$ (K.3.2b)

$K_{par}^b = K_{par}$ (K.4)

$\bullet \mathrm{pro}(2, \mathrm{i}_{dich} \bullet \mathrm{i}_{lat} \bullet \mathrm{i}_{lpb})$ (K.4.2)

$K_{par}^a = K_{per}$ (K.5)

$\textcolor{red}{\bullet \mathrm{pro}(1, \mathrm{i}_{early} \bullet \mathrm{i}_{div} \bullet \mathrm{i}_{lpb} \bullet \mathrm{inh}(3, \mathrm{i}_{med}))}$ (K.5.2)

$K_{per}^b = K_{per}$ (K.6)

$\textcolor{red}{\bullet \mathrm{inh}(1, \mathrm{i}_{early} \bullet \mathrm{i}_{div} \bullet \mathrm{i}_{lpb} \bullet \mathrm{inh}(3, \mathrm{i}_{med}))}$ (K.6.2)

Summary of KRN for ground state is given in *Figure 12I*.

Several new regions are created to incorporate genetic interactions defined in the wedge models described in *Figures 9* and *10*: SECVEIN (SEC), expressed at the secondary vein position of each petal and producing signal, sSEC; LIPCLIFF (LIF), expressed at the proximal region of the LIP; and LIPDISTAL (LIS), at the distal region of the LIP. In the published model, DIV is initially expressed uniformly across the lower petals and becomes graded at 14 DAI, due to inhibition by LTS (expressed at the hinge region of the laterals). In the current model, DIV is graded from early stages through inhibition by sRAD. The sRAD signal is produced in the dorsal petals and diffuses into lateral petals,

where it activates the HINGE identity factor (instead of LTS) and LPDP identity factor at the lateral-dorsal petal junction.

## Figure 9K. *div* mutant

Same as the ground stage with the following modifications

- $K_{par}$ in the dorsal and lower petal lobes modified (modified K.1.6a and K.1.6b and new K.1.20) and growth of the LIP in the DICH region boosted (modified K.1.4b).
- $K_{par}$ boosted by LIPDISTAL (LIS) in lower petals while inhibited by HINGE. This activation is reduced by LATE (K.1.16 and K1.17).
- $K_{par}$ and $K_{per}$ both inhibited in the hinge region by sRAD, particularly in the LIP and PLT (K.1.18 and K.2.10, respectively).
- The differential growth in $K_{par}$ beween A (adaxial) and B (abaxial) surfaces at the sRIM introduced, to promote petal bending (promotion in $K_{par}^a$, K.3.2c) and furrow formation (inhibition in $K_{par}^b$, K.4.3)
- $K_{par}$ in DISTAL LOBE (DTL) promoted at the hinge region (K.1.19).
- $K_{par}$ inhibited by MED in combination with LIF and UPTUBE, particularly at the rim of the lower petals (K.1.21).
- The proximodistal arms of the orthogonal directional conflict are specified by boosting $K_{par}$ at the petal junctions (with LAT, excluding dorsal-dorsal junctions) specifically at the regions of LIPCLIFF (LIF) and distal palate (UPTUBE), with the exception of the lateral-dorsal LIP region where Kpar is inhibited (K.1.22). Additionally, $K_{par}$ at the rim is inhibited (K.1.23).
- The mediolateral arms of the orthogonal directional conflict are established by promoting $K_{per}$ with sRIM flanking the petal junctions, excluding the dorsal petal junction, and inhibiting $K_{per}$ in with LAT and MED in combination with LIP and PLT (K.2.11-K.2.13).
- $K_{per}$ in distal petal regions promoted by removing K.2.5 and modifying K.2.4c)

The KRN equations are:

$$K_{par} = \hspace{8cm} (K.7)$$
$$(\text{as K.1.1} - \text{K.1.3}, \text{K.1.8}, \text{and K.1.10} - \text{K.1.11})$$

$$\bullet \text{pro}\left(1.4,\ i_{early}\bullet i_{lip}\bullet i_{dich}\bullet i_{lat}\right) \hspace{4cm} (K.1.4b)$$

$$\bullet \text{inh}\left(10,\ i_{early}\bullet i_{rim}\right) \hspace{5cm} (K.1.5)$$

$$\bullet \text{inh}\left(0.5,\ i_{late}\bullet i_{med}\bullet i_{lobe}\bullet i_{cyc}\right) \hspace{3.5cm} (K.1.6a)$$

$$\bullet \text{inh}\left(0.3,\ i_{late}\bullet i_{lobe}\bullet \text{pro}(0.5,\ i_{med})\bullet \text{inh}(5,\ i_{dist})\bullet \text{inh}\left(5,\ i_{lif}\right)\bullet \text{inh}\left(100,\ i_{cyc}\right)\right) \hspace{1cm} (K.1.6b)$$

$$\bullet \text{inh}\left(1,\ i_{rad}\bullet i_{plt}\bullet \text{inh}(15,\ i_{dich}\bullet \text{inh}(5,\ i_{lat}))\bullet \text{inh}\left(30,\ i_{cyc}\bullet \text{inh}(40,\ i_{lat}^2)\right)\right) \hspace{1cm} (K.1.9a)$$

$$\bullet \text{pro}\left(2,\ i_{early}\bullet i_{lis}\bullet \text{inh}(5,\ s_{rad})\bullet \text{inh}(100,\ i_{rad})\right) \hspace{2.5cm} (K.1.16)$$

$$\bullet \text{pro}(0.5,\ i_{late}\bullet i_{lis}\bullet \text{inh}(5,\ s_{rad})\bullet \text{inh}(100,\ i_{rad})) \hspace{2.5cm} (K.1.17)$$

$$\bullet \text{inh}\left(2,\ s_{rad}\bullet \left(i_{lip} + 0.5\bullet i_{uptube}\right)\bullet \text{inh}\bullet(100,\ s_{rad}{>}0.25)\right) \hspace{1.5cm} (K.1.18)$$

$$\bullet \text{inh}\left(0.35,\ i_{late}\bullet i_{dtl}\bullet i_{hinge}\bullet \text{inh}(100,\ i_{lip})\right) \hspace{2.5cm} (K.1.19)$$

$$\bullet \text{pro}\left(0.3,\ i_{early}\bullet i_{lobe}\bullet \text{inh}\left(100,\ i_{lip}\right)\bullet \text{inh}(1,\ i_{rad})\right) \hspace{2cm} (K.1.20)$$

$$\bullet \text{inh}\left(0.5,\ i_{late}\bullet i_{med}\bullet \left(i_{lip} + i_{uptube}\right)\bullet \text{pro}(2,\ s_{rim})\bullet \text{inh}(100,\ i_{rad})\right) \hspace{1cm} (K.1.21)$$

$$\bullet \text{pro}\left(0.8,\ i_{late}\bullet i_{lat}\bullet \left(i_{lip} + i_{uptube}\right)\bullet \text{inh}(100,\ s_{rad{>}0.4})\bullet \text{pro}\left(1,\ i_{lpdp}\right)\bullet \text{inh}\left(100,\ i_{lip}\bullet i_{lpdp}\right)\right) \hspace{0.5cm} (K.1.22)$$

$$\bullet \text{inh}\left(2,\ i_{late}\bullet s_{rim}^2\bullet \text{inh}(100,\ i_{rad})\right) \hspace{3.5cm} (K.1.23)$$

$$K_{per} = \tag{K.8}$$

(as K.2.1 − K.2.3 and K.2.7)

$$\bullet \mathrm{inh}(0.3,\ \mathrm{i}_{late}\bullet\mathrm{i}_{lobe}\bullet\mathrm{i}_{med}) \tag{K.2.5}$$

$$\bullet \mathrm{pro}\big(1,\ \mathrm{i}_{late}\bullet\mathrm{i}_{dist}\bullet\mathrm{pro}\big(1.8,\ (\mathrm{i}_{cyc})\big)\big) \tag{K.2.4c}$$

$$\bullet \mathrm{inh}\big(4,\ \mathrm{S}_{rad}\bullet\mathrm{i}_{hinge}\bullet\mathrm{inh}\big(100,\mathrm{i}_{tube}\bullet\mathrm{inh}(100,\ \mathrm{i}_{plt})\big)\big)$$

$$\bullet \mathrm{pro}(2,\ \mathrm{S}_{rim})\bullet\mathrm{inh}(10,\ \mathrm{i}_{prox})\bullet\mathrm{inh}\big(100,\ \mathrm{i}_{dtl}\bullet\mathrm{inh}(100,\ \mathrm{i}_{lip})\big) \tag{K.2.10}$$

$$\bullet \mathrm{pro}\big(12,\ \mathrm{i}_{late}\bullet\mathrm{S}_{rim}\bullet\mathrm{pro}(5,\mathrm{i}_{lpdp})\bullet\mathrm{inh}(100,\mathrm{i}_{lpdp>0.644})\bullet\mathrm{pro}(5,\ \mathrm{i}_{hinge})$$

$$\bullet \mathrm{inh}(10,\ \mathrm{S}_{rad>0.028})\bullet\mathrm{inh}(100,\ s_{rad>0.5})\bullet\mathrm{inh}(100,\ \mathrm{i}_{dich})\big) \tag{K.1.11}$$

$$\bullet \mathrm{inh}\big(2,\ \mathrm{i}_{late}\bullet\mathrm{i}_{med}\bullet(\mathrm{i}_{lip}+\mathrm{i}_{plt})\bullet\mathrm{pro}(1,\ \mathrm{s}_{rim})\bullet\mathrm{inh}(100,\ \mathrm{i}_{rad})\bullet\mathrm{inh}(100,\ \mathrm{i}_{hinge})\big) \tag{K.2.12}$$

$$\bullet \mathrm{inh}\big(2,\ \mathrm{i}_{late}\bullet\mathrm{i}_{lat}\bullet(\mathrm{i}_{lip}+\mathrm{i}_{plt})\bullet\mathrm{pro}(1,\ \mathrm{s}_{rim})\bullet\mathrm{inh}(100,\ \mathrm{i}_{rad})\bullet\mathrm{inh}(100,\ \mathrm{i}_{hinge})\big) \tag{K.2.13}$$

$$K_{par}^{a} = K_{par} \tag{K.9}$$

(as K.3.3 − K.3.4)

$$+\,0.04\bullet\mathrm{i}_{early}\bullet(\mathrm{S}_{rim>0.2})\bullet\mathrm{nih}\big(2,\mathrm{i}_{cyc}\ \mathrm{inh}(25,\ \mathrm{i}_{dich})\ \mathrm{inh}(1,\mathrm{i}_{hinge})\big) \tag{K.3.2C}$$

$$K_{par}^{b} = K_{par} \tag{K.10}$$

(as K.4.2)

$$\bullet \mathrm{inh}(0.5,\ \mathrm{S}_{rim>0.15}\bullet\mathrm{inh}(100,\ \mathrm{i}_{rad})\bullet\mathrm{inh}(10,\ \mathrm{S}_{rad})) \tag{K.4.3}$$

A summary of KRN for the *div* mutant, showing only new interactions (green), and modified interactions (blue), is given in **Figure 12J**.

## Figure 9—figure supplement 1B. *div* mutant corolla without orthogonal conflict

Equations as in previous section with those establishing the orthogonal growth orientations (K.1.22, K.2.11 and K.2.13) removed.

## Figure 10k wild type corolla

The DIV factor is introduced into the above *div* corolla model. DIV modulated the growth rates and orientations across the corolla as follows:

- DIV decreases $K_{par}$ with LIS (modified K.1.16a and K.1.17a)
- DIV promotes $K_{par}$ with LIF (K.1.24 and K.1.25).
- DIV promotes $K_{par}$ with PLT (K.1.26 and K.1.27).
- DIV inhibits $K_{par}$ with MED in the ventral petal to generate ventral furrow (K.1.28).
- DIV extends the region of high $K_{per}$ into the LIP at the position of the secondary veins (sSEC) (K.2.14).
- DIV inhibits $K_{per}$ when above a certain threshold (narrowing its expression to the ventral petal - vDIV) (K.2.15).
- DIV inhibits $K_{per}$ in the lateral petal lobe, keeping the distal edge of the LIP narrow and allowing the proximal region of the lip to bulge out slightly (K.2.16). This is further enhanced by the differential surface $K_{per}$ at the LIP (K.5.3 and K.6.3).
- DIV promotes surface conflict the lip-lobe boundary (LPB) by inhibiting $K_{par}$ in the A surface. This allows the lower lips to bend upwards (K.3.6). This is further enhanced by the differential surface $K_{per}$ in the same region (K.5.2a and K.6.2a)
- DIV promotes $K_{par}$ with RIM (K.4.3a).

DIV also activated a new region of polariser degradation at the SINUS (region at the intersection between the petal junctions and top lip nodes) at the repatterning stage (14 DAI). Upon the activation of this new polarity sink, the polarity field deflects towards the sinus. The deflection of polarity is kept specific to the adaxial (B) surface of the canvas by freezing the polarity at the A surface, preventing it from reorienting.

KRN equations are:

$$K_{par} = \tag{K.11}$$

$$(\text{as } K.1.1 - K.1.4b, K.1.6a, K.1.6b, K.1.8 - K.1.11, K.1.18 = K.1.23)$$

$$\bullet \text{pro}\big(2,\ i_{early}\bullet i_{lis}\bullet \text{inh}(5,\ S_{rad})\bullet \text{inh}(100,\ i_{rad})\bullet \text{inh}(0.7, i_{div})\big) \tag{K.1.1.6a}$$

$$\bullet \text{pro}\big(0.5,\ i_{late}\bullet i_{lis}\bullet \text{inh}(5,\ S_{rad})\bullet \text{inh}(100,\ i_{rad})\bullet \text{inh}(0.7,\ i_{div})\big) \tag{K.1.17a}$$

$$\bullet \text{pro}\big(2.5,\ i_{early}\bullet i_{div}\bullet i_{lif}\bullet \text{inh}(100,\ i_{rim})\big) \tag{K.1.24}$$

$$\bullet \text{pro}\big(0.2,\ i_{late}\bullet i_{div}\bullet i_{lif}\bullet \text{inh}(100,\ i_{rim})\big) \tag{K.1.25}$$

$$\bullet \text{pro}\big(2,\ i_{early}\bullet i_{div}\bullet i_{plt}\big) \tag{K.1.26}$$

$$\bullet \text{pro}\big(0.45,\ i_{late}\bullet i_{div}\bullet i_{plt}\bullet \text{inh}(2,\ i_{uptube})\bullet \text{inh}(100, i_{lpdp})\big) \tag{K.1.27}$$

$$\bullet \text{inh}\big(0.8,\ i_{late}\bullet i_{med}\bullet (i_{lif} + 0.5\bullet i_{plt})\bullet (i_{div>0.97})\big) \tag{K.1.28}$$

$$K_{per} = \tag{K.12}$$

$$(\text{as } K.2.1 - K.2.4c,\ K.2.7 \text{ and } K.2.10 - K.2.13)$$

$$\bullet \text{pro}\big(6,\ i_{late}\bullet i_{div}\bullet S_{sec}\bullet i_{lip}\bullet \text{inh}(100,\ S_{rim}^2)\big) \tag{K.2.14}$$

$$\bullet \text{inh}\big(1,\ i_{late}\bullet (i_{div>0.97})\bullet \text{pro}(2,\ S_{rim>0.2})\bullet \text{inh}(1,\ i_{plt})\bullet \text{inh}(10,\ i_{lif})\bullet \text{inh}(100,\ i_{dtl}\bullet \text{inh}(100,\ i_{lip}))\big) \tag{K.1.15}$$

$$\bullet \text{inh}\big(1,\ i_{late}\bullet i_{mlobe}\bullet (i_{div>0.995}\& i_{div>0.7})\big) \tag{K.2.16}$$

$$K_{par}^a = K_{par} \tag{K.13}$$

$$(\text{as } K.3.2c,\ K.3.3 - K.3.4)$$

$$\bullet \text{inh}\big(0.2,\ \bullet i_{div}\bullet i\big)_{lpb}\big) \tag{K.3.6}$$

$$K_{par}^b = K_{par} \tag{K.14}$$

$$(\text{as } K.4.2)$$

$$\bullet \text{inh}(0.5,\ S_{rim>0.15}\bullet \text{pro}(1,\bullet i_{div})\bullet \text{inh}(100,\ i_{rad})\bullet \text{inh}(10, s_{rad})) \tag{K.4.3a}$$

$$K_{par}^a = K_{per} \tag{K.15}$$

$$\bullet \text{pro}\big(1,\ i_{early}\bullet i_{div}\bullet i_{lpb}\big) \tag{K.5.2a}$$

$$\bullet \text{inh}\big(0.5,\ i_{early}\bullet (i_{div>0.97})\bullet i_{lif}\big) \tag{K.5.3}$$

$$K_{per}^b = K_{per} \tag{K.16}$$

$$\bullet \text{inh}\big(1,\ i_{early}\bullet i_{div}\bullet i_{lpb}\big) \tag{K.6.2a}$$

$$\bullet \text{inh}\big(2,\ i_{late}\bullet (i_{div>0.97})\bullet i_{lip}\big) \tag{K.6.3}$$

Summary of KRN for wild-type depicting the DIV-dependent interactions alone (green and bold interactions) is given in *Figure 12K*.

To test the contribution of the different types of tissue conflicts to the shape of the wild-type Snapdragon corolla, we removed each of the conflicts independently.

## Figure 9—figure supplement 1C. No orthogonal conflict
Orthogonal conflict equations (K.1.22, K.2.11, K.2.13 and K.2.14) removed.

## Figure 9—figure supplement 1D. No polarity deflection
Region of polariser degradation at the SINUS removed.

## Figure 9—figure supplement 1E. No surface conflict
Surface differential equations K.3-K.6 removed.

## Figure 9—figure supplement 1F No areal conflict
Areal growth rates normalised as in *Figure 9H*.

## Acknowledgements
We thank all the members of the Coen lab for helpful discussions. Thank you Sarah Hake, Graeme Mitchison, Desmond Bradley, Christopher Whitewoods, Manuela Costa, Catherine Mansfield, Beatriz Gonçalves, for comments on the manuscript. A special thanks to Jonathan Keep for moulding the *div* and wild-type clay models. We thank Lucy Copsey and Catherine Taylor for their help with plant

material. We also thank Grant Calder for the confocal microscope support. A.B.R was supported by an EMBO-long term fellowship (ALTF 568–2008) and HFSP long-term fellowship (LT000563). The author's research was supported by the UK Biotechnology and Biological Research Council (P.S, J.A. B, J.R.K. and E.C., BB/J004588/1.). We would like to dedicate this article to the memory of Andrew Bangham, our dear friend and collaborator, for all his enthusiasm, work and support on this project.

## Additional information

### Funding

| Funder | Grant reference number | Author |
| --- | --- | --- |
| European Molecular Biology Organization | ALTF 568-2008 | Alexandra B Rebocho<br>Enrico Coen |
| Human Frontier Science Program | LT000563/2010 | Alexandra B Rebocho<br>Enrico Coen |
| Biotechnology and Biological Sciences Research Council | BB/J004588/1 | Paul Southam<br>J Richard Kennaway<br>Enrico Coen |

The funders had no role in study design, data collection and interpretation, or the decision to submit the work for publication.

### Author contributions

ABR, Conception and design, Acquisition of data, Analysis and interpretation of data, Drafting or revising the article; PS, Designed and performed experiments and computational modelling, Designed PinPoint software and made PIN1 polarity calculations, Analysed data and wrote the paper; JRK, Analysis and interpretation of data, Supported GFtbox modelling; JAB, Conception and design; EC, Conception and design, Analysis and interpretation of data, Drafting or revising the article

### Author ORCIDs

Enrico Coen, http://orcid.org/0000-0001-8454-8767

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
