## [Decision Letter]

Thank you for submitting your article "Generation of shape complexity through tissue conflicts" for consideration by *eLife*. Your article has been reviewed by three peer reviewers, and the evaluation has been overseen by Naama Barkai as the Senior Editor and Reviewing Editor. The following individual involved in review of your submission has agreed to reveal her identity: Ottoline Leyser (Reviewer #2).

The reviewers have discussed the reviews with one another and the Reviewing Editor has drafted this decision to help you prepare a revised submission.

The authors study the process of tissue deformation, focusing on the petal deformation during the development of the bee landing pad in the snapdragon flower. The key question addressed is how such deformation results from individual cell behavior. As main tools, the authors combine experimental methods that allow to quantitatively follow tissue growth with computer simulations. As a main result, the authors propose the framework of 'tissue conflict', the idea that deformation can result from differences in the growth behavior of individual cells, and which result from specific patterns of gene expression at the cellular and tissue level.

All reviewers recommended this paper as an excellent contribution to this field that should be of wide general interest because of the important conceptual advance in provides.

As you will see below, however, there are points raised which require clarification before publication. Most of these points do not require new data but a better explanation and/or exploration of the existing data and its interpretation, and wider consideration of alternative interpretations. We would like to ask you therefore to address all requests.

*Reviewer #1:*

The experiments and modeling aimed at elucidating the role of DIV in shaping snapdragon flowers provide a nice complement to other recent works from the Coen lab (Eldridge et al. 2016 Development, Sauret-Güeto et al. 2013 PLoS biol, Kuchen et al. 2012, Green et al. 2010 PLoS Biology, Kenneway et al. 2011 PLoS Comp Bio) illustrating how genetically specified growth and mechanics can shape organ form. In this framework the concepts of specified and resultant growth have been very valuable.

A systematic examination of growth patterns leading to out-of-plane deformations is interesting and useful for the understanding of the emergence of organ form. The simulation study at the start of the manuscript is a step in this direction, and extending this exploration into something more substantial would represent a significant contribution.

A key difficulty with the manuscript is that the central concept of "tissue conflicts" is not sufficiently well-defined. Specifically, it is not clear how "tissue conflicts" are related to aspects of growth (specified or resultant) or mechanics (stresses, strains and residual stresses). Comments in the manuscript seem to imply that "conflicts" either correspond to situations when specified growth induces residual stresses (e.g. Introduction: "Even though growth is isotropic, anisotropies result from areal conflict"), or growth is non-uniform. Either case seems somewhat problematic.

Residual stresses may in some cases be sufficient to generate out of plane deformations, however it is unclear whether they are from a theoretical perspective necessary. Consequently, it is unclear whether and to what degree residual stresses drive generation of 3D forms, or are a common by-product of growth that sometimes feeds back to geometry.

Similarly, non-uniform growth does not necessarily lead to out of plane deformations. For instance, Coleocetae has very non-uniform growth but none the less remains flat (Dupuy L, Mackenzie J, Haseloff J. 2010. Coordination of plant cell division and expansion in a simple morphogenetic system. Proceedings of the National Academy of Sciences, USA 107: 2711-2716).

In either case, it is unclear what conceptual clarity is gained by introducing the notion "conflicts" or if there can be a proper mathematical conceptualization of a conflict-something which we have for residual stresses and growth. The danger therefore is that precise formulations are replaced with an appealing but imprecise notion.

Specific substantive comments:

1) Initial cell geometry presumably has a strong impact on the cell division patterns in the cell file simulations presented in Figure 3. Without quantifying this effect, how is it possible to accurately interpret the results of these simulations? This seems a key issue to resolve.

2) Cell divisions in cell file simulations are performed by splitting cells along the shortest wall passing through the centroid. As this divides cells according to Errera's rule, it is unclear why the authors have chosen to cite Besson and Dumais (PNAS 2011) instead of Errera's original manuscript (Errera L (1886) Sur une condition fondamentale d'équilibre des cellules vivantes.). The simulated patterns presented in Figure 4 also appear to incorporate shortening of the new cell wall, which is not considered by Besson and Dumais, but instead the cell division method proposed in Nakielski, J. (2000) in Pattern Formation in Biology, Vision, and Dynamics.

3) Examining the longitudinal midsections of WT flowers in Figure 3, the form of the lower corolla appears to closely follow the anatomy of organs enclosed by the petals. This seems relevant but the authors do not seem to comment on it? As similar images are not provided for div mutants it is thus unclear to what degree the observed differences in corolla form between mutant and wild-type plants may relate to differences in the growth of adjacent portions of the flower.

4) The PIN quantification tool is interesting. A major challenge however is how to confidently quantify amount of signal at each side of the wall. The two flanking membranes are very thin and the signal is generally not sufficiently discrete in the magnifications they are looking. I can see how quantification would work for total amount of signal in the two sides flanking a wall but if they can quantify each side they need to explain the method more clearly. How do they deal with problems introduced by many sections being oblique which would skew the relative signal to the wall or slight non-uniformity in staining? How does calcofluor affect the method? Overall it would seem appropriate to provide information on fidelity and accuracy of the method.

*Reviewer #2:*

This paper introduces the idea of tissue conflict as a "general framework for viewing out-of-plane deformations with or without overall growth" during morphogenesis. I prefer the term "framework" as used here, to "mechanism" as used elsewhere in the paper. The value of this framework is illustrated in the context of the analysis of petal deformation during the development of the bee landing pad in the snapdragon flower. The tissue conflict concept extends a conceptual toolbox generated by this team, encapsulated in a computational modelling environment. This work has provided important insights into the process of morphogenesis, which remains a largely enigmatic area of developmental biology because of the difficulty of thinking coherently in 4D. The addition of the tissue conflicts idea seems to me very promising, and this is successfully illustrated by the insights it provides into petal deformation, as regulated by the DIV gene.

The paper has the difficult task of presenting 4D results, both in general and in the specific case of the snapdragon corolla. The authors do a good job of leading the reader through these complexities. I have suggested some additions below that would have helped me further, but I recognize that different people will find different narratives/illustrations helpful.

In the Discussion, the authors consider how their framework for the analysis of morphogenesis compares to others in the field. They also consider the molecular-cellular basis for the properties they propose. These aspects are extremely helpful in allowing the reader to understand the relationship between the various heuristics currently in use.

Overall, I find this paper to be an excellent contribution to this field that should be of wide general interest because of the important conceptual advance in provides.

My specific suggestions for improvement are:

1) For me the OPT sections presented in Figure 3 provided a much easier entry point to understanding the out of plane deformations of interest than the illustrations in Figure 2. Could some sections of this sort be added to Figure 2?

2) I would have liked a more precise definition of exactly how the red, yellow and orange inferred growth orientation lines were drawn. For example, in Figure 5 the growth orientation lines in the zoomed in box from B do not map well onto the lines in the zoomed out image. Presumably the authors generated some scale-based rules for exactly how to draw these lines. These could maybe be included in Figure 4 where the approach is introduced.

3) I would have found it helpful if Figure 10 and Figure 9 were more directly comparable, with equivalent illustrations presented in the same order even where there was no difference. DIV is required to work quite hard and it is important to be able to grasp quickly the extent of its activities.

4) It would be helpful if the Discussion included a section on how resultant growth might be generated at the cellular level.

[Please note I did not review the detailed computational model]

*Reviewer #3:*

This is an elegant and overall well-executed study on morphogenesis in snapdragon petals. In addition, the paper is written enthusiastically and pedagogically. I nevertheless have a few reservations.

1) "Growth conflict", as termed by the authors, is a concept that has a long history including in previous work of the Coen group. It may be called "embedding of surfaces" in mathematics (e.g. work of John Nash), "incompatible strain" in applied mathematics (e.g. Lewicka et al.) or "non-Euclidian metric" in physics (e.g. Sharon et al.), as well in developmental biology (e.g. Paul Green, referenced in the paper). The main contribution of the present study is to characterize a case were spatial differences in growth are directional (instead of differences in areal growth as in previous studies).

2) It is possible to achieve the same 3D shape by various patterns of 2D growth, notably a pattern with no conflict (where resultant and target growth are identical). What is the experimental evidence that leads the authors to exclude an explanation with no conflict?

3) As stated by the authors, the 3D shape arises from residual stresses due to the spatial differences in growth rate or direction. However, they neglect mechanical stresses/forces due to other floral organs. Are other organs really negligible? For instance, is petal shape influenced by homeotic conversions of stamens?

4) It is unclear what are the causal links between directional growth, cell divisions, and/or PIN1a polarity, and the putative underlying polarity field. How far do cell divisions show the main direction of growth? How independent is PIN1a localisation from the orientation of recent divisions?

---

## [Author Response]

*[…] Reviewer #1:*

*[…] A key difficulty with the manuscript is that the central concept of "tissue conflicts" is not sufficiently well-defined. Specifically, it is not clear how "tissue conflicts" are related to aspects of growth (specified or resultant) or mechanics (stresses, strains and residual stresses). Comments in the manuscript seem to imply that "conflicts" either correspond to situations when specified growth induces residual stresses (e.g. Introduction: "Even though growth is isotropic, anisotropies result from areal conflict"), or growth is non-uniform. Either case seems somewhat problematic.*

*Residual stresses may in some cases be sufficient to generate out of plane deformations, however it is unclear whether they are from a theoretical perspective necessary. Consequently, it is unclear whether and to what degree residual stresses drive generation of 3D forms, or are a common by-product of growth that sometimes feeds back to geometry.*

*Similarly, non-uniform growth does not necessarily lead to out of plane deformations. For instance, Coleocetae has very non-uniform growth but none the less remains flat (Dupuy L, Mackenzie J, Haseloff J. 2010. Coordination of plant cell division and expansion in a simple morphogenetic system. Proceedings of the National Academy of Sciences, USA 107: 2711-2716).*

*In either case, it is unclear what conceptual clarity is gained by introducing the notion "conflicts" or if there can be a proper mathematical conceptualization of a conflict-something which we have for residual stresses and growth. The danger therefore is that precise formulations are replaced with an appealing but imprecise notion.*

Based on the comments of this and the other reviewers we realise that the term ‘tissue conflict’ was too readily confused with residual stress. As the key point about tissue conflict is the way it becomes resolved to generate local rotations and curvature, we have changed the name to ‘tissue conflict resolution’ and defined it more clearly in the Introduction (see response to referee #2 below), and have given a more mathematical formulation in the Materials and methods. In particular, we have clarified that tissue conflict resolution refers to a mechanism for generating tissue rotations through differential growth, as distinct from rotations arising through external forces. We also cite a recent review covering planar and well as out-of-plane tissue conflict resolution (Coen & Rebocho, 2016). The mathematical formulation given in the Materials and methods is below:

“Definition of tissue conflict resolution. Mathematically, growth of a region is described by a growth tensor which can have both a strain (symmetric) and rotational (skew-symmetric) component (Hejnowicz and Romberger 1984). […] Some residual strain is inevitable in cases of tissue buckling (curvature through areal or directional conflicts), as the two surfaces of the tissue grow to a different extent, even though their specified strains are identical.”

*Specific substantive comments:*

*1) Initial cell geometry presumably has a strong impact on the cell division patterns in the cell file simulations presented in Figure 3. Without quantifying this effect, how is it possible to accurately interpret the results of these simulations? This seems a key issue to resolve.*

The reviewer raises an interesting point. To address it we did simulations with various initial cell geometries. The resultant division patterns are very similar, although the outlines of the clones have a different shape. We have mentioned this finding in the Results (subsection “Morphogenesis of wild type and *div*”, seventh paragraph) and included a supplementary figure (Figure 4—figure supplement 1).

*2) Cell divisions in cell file simulations are performed by splitting cells along the shortest wall passing through the centroid. As this divides cells according to Errera's rule, it is unclear why the authors have chosen to cite Besson and Dumais (PNAS 2011) instead of Errera's original manuscript (Errera L (1886) Sur une condition fondamentale d'équilibre des cellules vivantes.). The simulated patterns presented in Figure 4 also appear to incorporate shortening of the new cell wall, which is not considered by Besson and Dumais, but instead the cell division method proposed in Nakielski, J. (2000) in Pattern Formation in Biology, Vision, and Dynamics.*

We thank the reviewer for raising these points and have modified the citations and explained the method used more fully:

“We assume that when the area of a cell exceeds a threshold, it is divided along the shortest path through its centroid (Errera 1886, Besson and Dumais 2011). The new wall is then shortened slightly to give more realistic angles (Nakielski 2000).”

*3) Examining the longitudinal midsections of WT flowers in Figure 3, the form of the lower corolla appears to closely follow the anatomy of organs enclosed by the petals. This seems relevant but the authors do not seem to comment on it? As similar images are not provided for div mutants it is thus unclear to what degree the observed differences in corolla form between mutant and wild-type plants may relate to differences in the growth of adjacent portions of the flower.*

The reviewer rightly points out that the corolla tube follows the shape of the anther for some of its length and raises the question of whether the anther may have a role in shaping the petal. One argument against this is that in *plena* mutants, in which stamens are replaced by petals, corolla shape in whorl 2 is similar to wild-type. We now cite this finding in the Introduction (second paragraph, see also response to reviewer #3 below).

*4) The PIN quantification tool is interesting. A major challenge however is how to confidently quantify amount of signal at each side of the wall. The two flanking membranes are very thin and the signal is generally not sufficiently discrete in the magnifications they are looking. I can see how quantification would work for total amount of signal in the two sides flanking a wall but if they can quantify each side they need to explain the method more clearly. How do they deal with problems introduced by many sections being oblique which would skew the relative signal to the wall or slight non-uniformity in staining? How does calcofluor affect the method? Overall it would seem appropriate to provide information on fidelity and accuracy of the method.*

We have added some close-up images of cells in the Supplementary material (Figure 8—figure supplement 2) showing the calcofluor and PIN signal. These images show that PIN signal can usually be localised to one side of the wall. We have also further clarified the description of the method (subsection “Quantifying PIN1 polarity”), explaining that a section near the midplane was chosen for each cell as this reduces the problem of obliqueness and non-uniformity.

*Reviewer #2:*

*This paper introduces the idea of tissue conflict as a "general framework for viewing out-of-plane deformations with or without overall growth" during morphogenesis. I prefer the term "framework" as used here, to "mechanism" as used elsewhere in the paper. The value of this framework is illustrated in the context of the analysis of petal deformation during the development of the bee landing pad in the snapdragon flower. The tissue conflict concept extends a conceptual toolbox generated by this team, encapsulated in a computational modelling environment. This work has provided important insights into the process of morphogenesis, which remains a largely enigmatic area of developmental biology because of the difficulty of thinking coherently in 4D. The addition of the tissue conflicts idea seems to me very promising, and this is successfully illustrated by the insights it provides into petal deformation, as regulated by the DIV gene.*

We thank the reviewer for raising this valuable point about mechanism versus framework. It is understandable as we did not explicitly mention the alternative mechanism, which is that external forces drive tissue rotation. To clarify this point, we have clarified tissue conflict resolution and the alternative mechanism in the Introduction:

“A key feature of out of plane deformations of a tissue sheet is that they involve generation of curvature (local rotations out of the plane). […] We refer to this second mechanism, in which heterogeneity of specified growth within the tissue leads to local rotations that reduce potential stresses, as tissue conflict resolution (for a more mathematical definition of tissue conflict resolution see Materials and methods).”

*The paper has the difficult task of presenting 4D results, both in general and in the specific case of the snapdragon corolla. The authors do a good job of leading the reader through these complexities. I have suggested some additions below that would have helped me further, but I recognize that different people will find different narratives/illustrations helpful.*

*In the Discussion, the authors consider how their framework for the analysis of morphogenesis compares to others in the field. They also consider the molecular-cellular basis for the properties they propose. These aspects are extremely helpful in allowing the reader to understand the relationship between the various heuristics currently in use.*

*Overall, I find this paper to be an excellent contribution to this field that should be of wide general interest because of the important conceptual advance in provides.*

*My specific suggestions for improvement are:*

*1) For me the OPT sections presented in Figure 3 provided a much easier entry point to understanding the out of plane deformations of interest than the illustrations in Figure 2. Could some sections of this sort be added to Figure 2?*

We thank the reviewer for bringing this problem to our attention, which indeed reflects the complexity of conveying shape change in 4D as the reviewer points out. Currently we give the mature flower phenotype first (Figure 2) and then its development (Figure 3). The OPT sections in Figure 3 are easier to understand because they are less complex than those of the mature flower (there are fewer overlaps and folds). For this reason, we would prefer to keep the current arrangement. Given that the description of Figure 3 follows on immediately from Figure 2 we hope this is acceptable.

*2) I would have liked a more precise definition of exactly how the red, yellow and orange inferred growth orientation lines were drawn. For example, in Figure 5 the growth orientation lines in the zoomed in box from B do not map well onto the lines in the zoomed out image. Presumably the authors generated some scale-based rules for exactly how to draw these lines. These could maybe be included in Figure 4 where the approach is introduced.*

We thank the reviewer for pointing out this issue. Confusion seems to have arisen in relating the zoomed-in and zoomed-out versions. This is because we chose to show only the clearest regions of oriented division in the zoomed-out version. Thus, only the left region of the box from B is shown as red lines in the zoomed-out version. We have clarified this point by giving a fuller explanation in the figure legends and by correcting the zoomed-in version of B with some extra yellow lines that had been left out by mistake.

*3) I would have found it helpful if Figure 9 and Figure 10 were more directly comparable, with equivalent illustrations presented in the same order even where there was no difference. DIV is required to work quite hard and it is important to be able to grasp quickly the extent of its activities.*

We thank the reviewer for this suggestion. We have revised Figure 10 and incorporated elements from Figure 9 to make it more comparable.

*4) It would be helpful if the Discussion included a section on how resultant growth might be generated at the cellular level.*

We thank the reviewer for raising this point. We did not include a more detailed discussion of cellular mechanisms to keep the paper short. We have now given a more detailed description of cellular level analysis in a recent review (Coen and Rebocho, 2016) and cite this in the paper at the end of the Discussion.

*[Please note I did not review the detailed computational model]*

*Reviewer #3:*

*This is an elegant and overall well-executed study on morphogenesis in snapdragon petals. In addition, the paper is written enthusiastically and pedagogically. I nevertheless have a few reservations.*

*1) "Growth conflict", as termed by the authors, is a concept that has a long history including in previous work of the Coen group. It may be called "embedding of surfaces" in mathematics (e.g. work of John Nash), "incompatible strain" in applied mathematics (e.g. Lewicka et al.) or "non-Euclidian metric" in physics (e.g. Sharon et al.), as well in developmental biology (e.g. Paul Green, referenced in the paper). The main contribution of the present study is to characterize a case were spatial differences in growth are directional (instead of differences in areal growth as in previous studies).*

As with our response to referee #1 we have clarified our definition of tissue conflict resolution in the Introduction and Materials and methods, to make it clearer that it is a mechanism for generating local rotations, whether or not residual strain is involved. We agree with the reviewer that a major contribution of our paper is to characterise a case where deformation involves conflicting orientations of growth, but we felt it important to place this type of interaction in relation to other forms of tissue conflict resolution.

*2) It is possible to achieve the same 3D shape by various patterns of 2D growth, notably a pattern with no conflict (where resultant and target growth are identical). What is the experimental evidence that leads the authors to exclude an explanation with no conflict?*

As we now mention in the Introduction, curvature can arise without tissue conflict resolution through external forces being applied to the tissue. We point out that the normal corolla shape of organ identity mutants argues against this type of mechanism playing a major role (see also response below). There can be cases of tissue conflict resolution in which no residual strain is generated (e.g. conformal maps mentioned in the mathematical definition of tissue conflict resolution). However, this does not apply to the buckling mechanisms we describe and give evidence for (i.e. cell file analysis and modelling), as residual strains are necessarily generated as the two surfaces of the tissue attain different resultant areas (even though they are specified to be the same).

*3) As stated by the authors, the 3D shape arises from residual stresses due to the spatial differences in growth rate or direction. However, they neglect mechanical stresses/forces due to other floral organs. Are other organs really negligible? For instance, is petal shape influenced by homeotic conversions of stamens?*

We thank the reviewer for raising this point. We have now pointed out in the Introduction that an alternative to tissue conflict resolution is that deformations arise through forces from other tissues. Evidence against this come from the observation that corolla shape in whorl 2 in *plena* mutants, where stamens are converted to petals, is similar to that in wild type, indicating that stresses from whorl 3 do not have a strong effect on petal shape.

*4) It is unclear what are the causal links between directional growth, cell divisions, and/or PIN1a polarity, and the putative underlying polarity field. How far do cell divisions show the main direction of growth? How independent is PIN1a localisation from the orientation of recent divisions?*

We have clarified the causal relationships between cellular behaviours and tissue growth in a recent review (Coen & Rebocho, 2016), now cited in our paper. For our simulations we make the plausible assumption that the plane of cell division does not directly direct growth but reflects the pattern of growth. We see a correlation between PIN1a localisation and orientations of division walls, the most notable being the correlation between diagonal walls and deflected PIN1a polarisation (noted in the sixth paragraph of the Discussion).